# Increased occurrences of consecutive La Niña events under global warming

Tao Geng[1,2], Fan Jia[3✉], Wenju Cai[2,4,5✉], Lixin Wu[1,2], Bolan Gan[2], Zhao Jing[1,2], Shujun Li[2] & Michael J. McPhaden[6]

Most El Niño events occur sporadically and peak in a single winter[1–3], whereas La Niña tends to develop after an El Niño and last for two years or longer[4–7]. Relative to single-year La Niña, consecutive La Niña features meridionally broader easterly winds and hence a slower heat recharge of the equatorial Pacific[6,7], enabling the cold anomalies to persist, exerting prolonged impacts on global climate, ecosystems and agriculture[8–13]. Future changes to multi-year-long La Niña events remain unknown. Here, using climate models under future greenhouse-gas forcings[14], we find an increased frequency of consecutive La Niña ranging from 19 ± 11% in a low-emission scenario to 33 ± 13% in a high-emission scenario, supported by an inter-model consensus stronger in higher-emission scenarios. Under greenhouse warming, a mean-state warming maximum in the subtropical northeastern Pacific enhances the regional thermodynamic response to perturbations, generating anomalous easterlies that are further northward than in the twentieth century in response to El Niño warm anomalies. The sensitivity of the northward-broadened anomaly pattern is further increased by a warming maximum in the equatorial eastern Pacific. The slower heat recharge associated with the northward-broadened easterly anomalies facilitates the cold anomalies of the first-year La Niña to persist into a second-year La Niña. Thus, climate extremes as seen during historical consecutive La Niña episodes probably occur more frequently in the twenty-first century.

The El Niño–Southern Oscillation (ENSO) is the strongest year-to-year climate fluctuation alternating irregularly between warm El Niño and cold La Niña events, severely disrupting global weather patterns, agriculture and ecosystems[15,16]. ENSO exhibits diversity in its temporal evolution. Specifically, most El Niño events terminate rapidly after maturing in boreal winter, whereas roughly half of La Niña events persist and re-intensify in the subsequent one or two years to become multi-year La Niña events[1,17–19]. Compared with a single-year La Niña event, the multi-year La Niña events such as 2020–2022 create higher or cumulative risk of extreme weather events worldwide[20], including, for example, droughts and wildfires in the southwestern United States[8,9], flooding over southeast Asia[21,22] and altered patterns of hurricanes, cyclones and monsoons across the Pacific and Atlantic oceans[20–23]. How the multi-year La Niña responds to greenhouse warming is an important issue, with widespread environmental and socioeconomic ramifications.

The persistence of La Niña events is often modulated by the amplitude of a preceding El Niño and influence from the subtropical North Pacific. Multi-year La Niña events tend to follow a strong El Niño[4–6]. For example, all three extreme El Niño events in the twentieth century (1972/73, 1982/83 and 1997/1998) were followed by multi-year La Niña events (Extended Data Fig. 1). The multi-year La Niña events occur because a large upper-ocean heat discharge of the equatorial Pacific induced by the strong El Niño requires more than one La Niña event to recharge to the climatological state[24], as the recharge process during La Niña is generally weaker than the discharge associated with El Niño[25].

Further, during the developing phase of the first La Niña in boreal spring, anomalous northeasterly winds in the subtropical North Pacific extend southwestward to the equator with covarying cold sea-surface temperature (SST) anomalies (Extended Data Fig. 2), resembling a negative phase of the North Pacific Meridional Mode (NPMM)[7,26,27]. The negative NPMM-like pattern features increased extratropical trade winds that result from a Gill-type atmospheric response to the strong El-Niño-related SST over the equatorial eastern Pacific[28,29] and mid-latitude atmospheric fluctuations induced by either the El Niño atmospheric teleconnection or internal stochastic variability[30–33]. The negative NPMM-like pattern grows and persists in boreal spring–summer under thermodynamic air–sea interactions, especially a wind–evaporation–SST (WES) feedback[34], favouring development of a meridionally broad La Niña[7]. The meridionally wider pattern of SST and easterly wind anomalies is accompanied by a weaker negative wind stress curl at more-extratropical latitudes, slowing the recharge of the equatorial Pacific[6,7]. Thus, the cold SST anomalies persist through the decaying phase of the first-year La Niña into spring and are strengthened by the seasonal positive Bjerknes feedback in late summer and autumn[35], and probably into another La Niña event.

[1]Laoshan Laboratory, Qingdao, China. [2]Frontier Science Center for Deep Ocean Multispheres and Earth System (FDOMES) and Physical Oceanography Laboratory, Ocean University of China, Qingdao, China. [3]CAS Key Laboratory of Ocean Circulation and Waves, Institute of Oceanology, Chinese Academy of Sciences, and Laoshan Laboratory, Qingdao, China. [4]Centre for Southern Hemisphere Oceans Research (CSHOR), CSIRO Oceans and Atmosphere, Hobart, Tasmania, Australia. [5]State Key Laboratory of Loess and Quaternary Geology, Institute of Earth Environment, Chinese Academy of Sciences, Xi'an, China. [6]NOAA Pacific Marine Environmental Laboratory, Seattle, WA, USA. ✉e-mail: jiafan@qdio.ac.cn; wenju.cai@csiro.au

How might greenhouse warming affect multi-year La Niña remains unknown. Here, using outputs from latest climate models participating in the Coupled Model Intercomparison Project Phase 6 (CMIP6) (ref. 14), we show a substantial increase in the frequency of multi-year La Niña far greater than that expected from a projected increase in strong El Niño[36,37].

## Observed features and model selection

We begin by evaluating simulation of multi-year La Niña events by CMIP6 models in the twentieth century climate (1900–1999) (see 'Observational and CMIP6 data' in Methods). A key aspect of ENSO asymmetry is that the cold SST anomalies during La Niña are weaker but last longer than warm SST anomalies during El Niño, manifesting as positively skewed SST anomalies in the central-eastern equatorial Pacific (the 'Niño3.4' region; 5° S–5° N, 170° W–120° W). The positive SST skewness is primarily governed by a nonlinear Bjerknes feedback, in which only after SST warming surpasses a threshold and local atmospheric deep convection is triggered do zonal winds respond nonlinearly to further warming in the eastern equatorial Pacific, leading to further SST warming[38–41]. Many models underestimate this feedback[37,42]. A total of 20 out of 37 models simulate a positive Niño3.4 SST skewness over the 1900–1999 period (Fig. 1a), with a multi-model ensemble (MME) mean of 0.27 that is close to the observed value (0.31). Better simulation of ENSO skewness is associated with more realistic simulation of ENSO dynamics[37,43,44]. Thus, we select these 20 models for further analysis.

We use 0.5 standard deviations (s.d.) of the October–February (ONDJF) averaged Niño3.4 SST index as a threshold to define ENSO events and identify a multi-year La Niña event when cold anomalies occur in ONDJF for two or more consecutive years (see 'Definition of multi-year La Niña events' in Methods). This yields eight multi-year La Niña events, accounting for 44.4% of all (single-year plus multi-year) La Niña events observed in the twentieth century (Extended Data Fig. 1a), consistent with previous studies using various methods to define La Niña[5,6,9,19]. Because of model selection, the observed occurrence ratio of multi-year La Niña over the twentieth century is reasonably reproduced in the 20 selected models with an MME mean of 42.9 ± 3.1%, but is underestimated in the 17 non-selected models, which simulate an MME mean of 28.4 ± 4.9%, consistent with what is expected from the simulated ENSO skewness (Supplementary Fig. 1).

The 20 selected models also simulate the key features of observed multi-year La Niña events reasonably well. First, a multi-year La Niña tends to follow a strong El Niño in D(0)JF(1) (December, January and February; '0' refers to the previous year and '1' for the first year) (Fig. 1b), which generates a large upper-ocean heat discharge that—in turn—induces a strong first-year La Niña condition in D(1)JF(2) (in which '2' refers to the second year; Extended Data Fig. 3). Second, the models reproduce the meridionally broad SST structure of multi-year La Niña in D(1)JF(2) that is connected to extratropical climate variations[6,7]; in contrast, the anomalies of a single-year La Niña is narrower and more equatorially confined (Fig. 1c,d). Such a tropical–subtropical connection during multi-year La Niña can be traced to MAM(1), when the precursory northeasterly and cold SST anomalies appear in the subtropical North Pacific and extend to the equator as a negative NPMM-like pattern (Extended Data Fig. 4).

Below we show that most of these models project an increased frequency of multi-year La Niña events under greenhouse warming. The increased frequency is far larger than that expected from the projected increase in frequency of strong El Niño.

## Increased frequency in a warming climate

We count the number of multi-year La Niña in the twentieth century (1900–1999) under historical forcing and compare with that in the twenty-first century (2000–2099) under a high-emission warming scenario (Shared Socioeconomic Pathway 5-8.5 (SSP585)) in each of the 20 selected models. We focus on the SSP585 scenario and discuss other scenarios as sensitivity tests. In aggregation, the frequency of multi-year La Niña increases from one event every 12.1 years in 1900–1999 (166 events in 2,000 years, which cover 20 100-year periods from 20 models) to one event every 9.1 years in 2000–2099 (220 events in 2,000 years). The multi-model mean increase of 33 ± 13% is statistically significant above the 95% confidence level according to a bootstrap test (see 'Statistical significance test' in Methods), underscored by a strong inter-model agreement with a total of 15 out of 20 models (75%) simulating more multi-year La Niña events in the warmer twenty-first century (Fig. 2a). The inter-model consensus mainly reflects an increased occurrence of 'double' La Niña (events that last two years), whereas there is no inter-model agreement on how 'triple' La Niña (events that last three years) may change under greenhouse warming (see 'Sensitivity of projected increase in multi-year La Niña occurrence' in Methods). On the other hand, the frequency of multi-year La Niña is underestimated and shows no marked change in the non-selected models (Fig. 2a), which possibly relates to the overly weak nonlinear ENSO dynamics (Fig. 1a).

The projected increase in multi-year La Niña frequency is in spite of a long-standing model bias of overly cold SSTs in the equatorial Pacific and a low detectability of a change directly inferred from observations, as the current observational record is short and subject to much uncertainty (see 'Model bias and recent multi-year La Niña change' in Methods). A sensitivity test finds that the increase occurs in other emission scenarios, including SSP126, SSP245 and SSP370, and the number of multi-year La Niña increases with intensity of greenhouse-gas emissions (Fig. 2a, last three columns). Specifically, the increased multi-year La Niña frequency in terms of a multi-model mean is 18.6 ± 11.1%, 25.8 ± 14.6% and 31.5 ± 17.3% for SSP126, SSP245 and SSP370, respectively, statistically significant above the 95% confidence level according to a bootstrap test, and supported by 59.1%, 65.0% and 65.2% of models (Supplementary Fig. 2). Using an alternative method to identify multi-year La Niña by taking into account the different amplitudes between its first and second peaks yields essentially the same results (see 'Sensitivity of projected increase in multi-year La Niña occurrence' in Methods; Supplementary Fig. 3).

The increased frequency of multi-year La Niña is out of the range of natural variability that is rather substantial[45,46]. To illustrate, we use the Niño3.4 SST anomalies from three experiments in each of the 20 selected models: a multi-century (more than 500 years) pre-industrial control (piControl) simulation, a historical run starting from piControl in the mid-nineteenth century and forced with historical forcings till 2014 and a future warming experiment thereafter under the SSP585 scenario that ends at 2100 (see 'Observational and CMIP6 data' in Methods). The number of multi-year La Niña is calculated first in each model and then averaged across the models, in a 60-year running window that moves separately in piControl and from 1850 (the start of historical run) to the end of the twenty-first century. The evolution time series shows a rapid increase of multi-year La Niña occurrence commencing from the second half of the twentieth century, emerging out of its natural variations in piControl (Fig. 2b, dashed line) and plateauing around the middle of the twenty-first century (Fig. 2b).

The occurrence ratio of multi-year La Niña over all (single-year plus multi-year) La Niña events increases in the future climate and the increase is statistically significant above the 95% confidence level (Fig. 2c, 'A'). Using a 1.5 s.d. threshold of the ONDJF Niño3.4 index to define strong ENSO events, we find that more than 83% (45 out of 54 events in aggregation) of the increase results from an increased percentage of multi-year La Niña preceded by a strong El Niño (Fig. 2c, 'B'), which is also statistically significant above the 95% confidence level according to a bootstrap test (see 'Statistical significance test' in Methods).

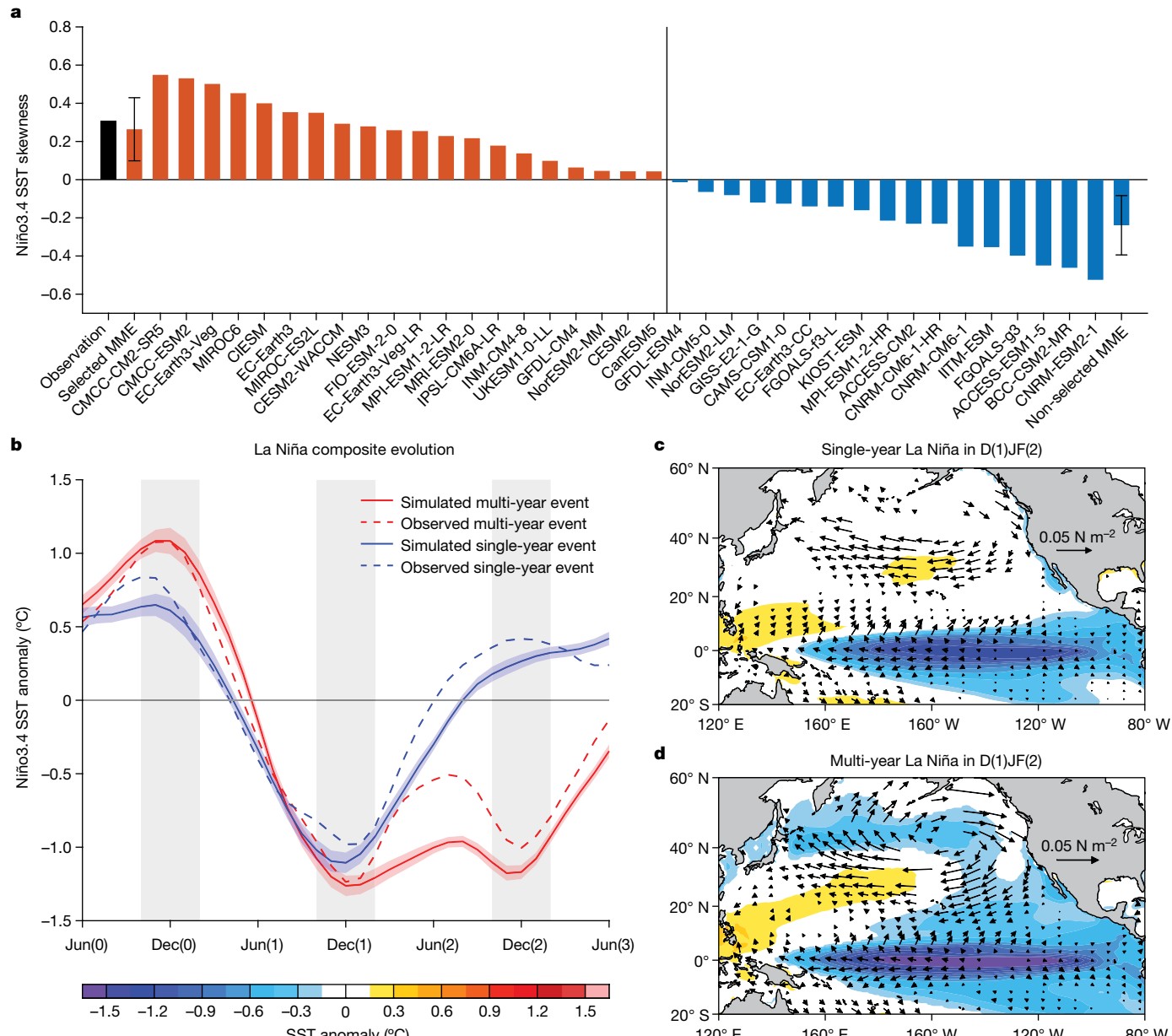

**Fig. 1 | Characteristics of simulated multi-year La Niña events. a**, Skewness of historical (1900–1999) Niño3.4 SST anomaly in observation (black bar) and CMIP6 models (coloured bars). The vertical line separates selected models with positive skewness (orange bars) from non-selected models with negative skewness (blue bars). The error bar denotes 1.0 s.d. of the inter-model spread in the selected (non-selected) MME. **b**, Temporal evolution of Niño3.4 SST anomaly composited for multi-year (red) and single-year (blue) La Niña events in the selected models over 1900–1999. Solid lines and shading indicate multi-model mean and 1.0 s.d. of a total of 10,000 inter-realizations based on a bootstrap method, respectively. Dashed lines indicate observations. The time series are smoothed with a three-month running-mean filter before analysis. The vertical grey shading denotes the time (October to February) when ENSO typically matures. **c,d**, Multi-model mean composite map of anomalous SST (°C; colouring) and surface wind stress (N m⁻²; vectors) for single-year (**c**) and multi-year (**d**) La Niña events during D(1)JF(2) in 1900–1999. Shown are values at which the ensemble mean exceeds 1.0 s.d. of the inter-model spread using a bootstrap method. Selected models simulate reasonably the observed evolution and pattern of multi-year La Niña.

Although strong El Niño events are projected to occur more frequently under greenhouse warming[37], the frequency of multi-year La Niña increases disproportionally more than that of strong El Niño, as seen in an approximately 65.5% increase (from 28.7 ± 4.0% in 1900–1999 to 47.5 ± 4.7% in 2000–2099) in the ratio of strong El Niño followed by multi-year La Niña over all strong El Niño events (Fig. 2c, 'C'). Furthermore, there is no inter-model consensus on the amplitude change of strong El Niño (Extended Data Fig. 5a; other emission scenarios in Supplementary Fig. 4) nor an inter-model relationship between changes in strong El Niño amplitude and changes in multi-year La Niña occurrences (Extended Data Fig. 5b,c). These results indicate that El Niño in

general becomes more efficient in triggering multi-year La Niña under a warming climate.

## Beyond changes in strong El Niño

A preceding El Niño induces an equatorial upper-ocean heat discharge[24] and a negative NPMM-like response in the subtropical North Pacific[27,29,47]. We calculate the discharge rate using anomalies of sea-surface height (SSH) in the equatorial Pacific (5° N–5° S, 120° E–80° W) during the mature-to-decaying phase of El Niño (December to May). Observational results indicate that such a rate of SSH change captures the ENSO-related

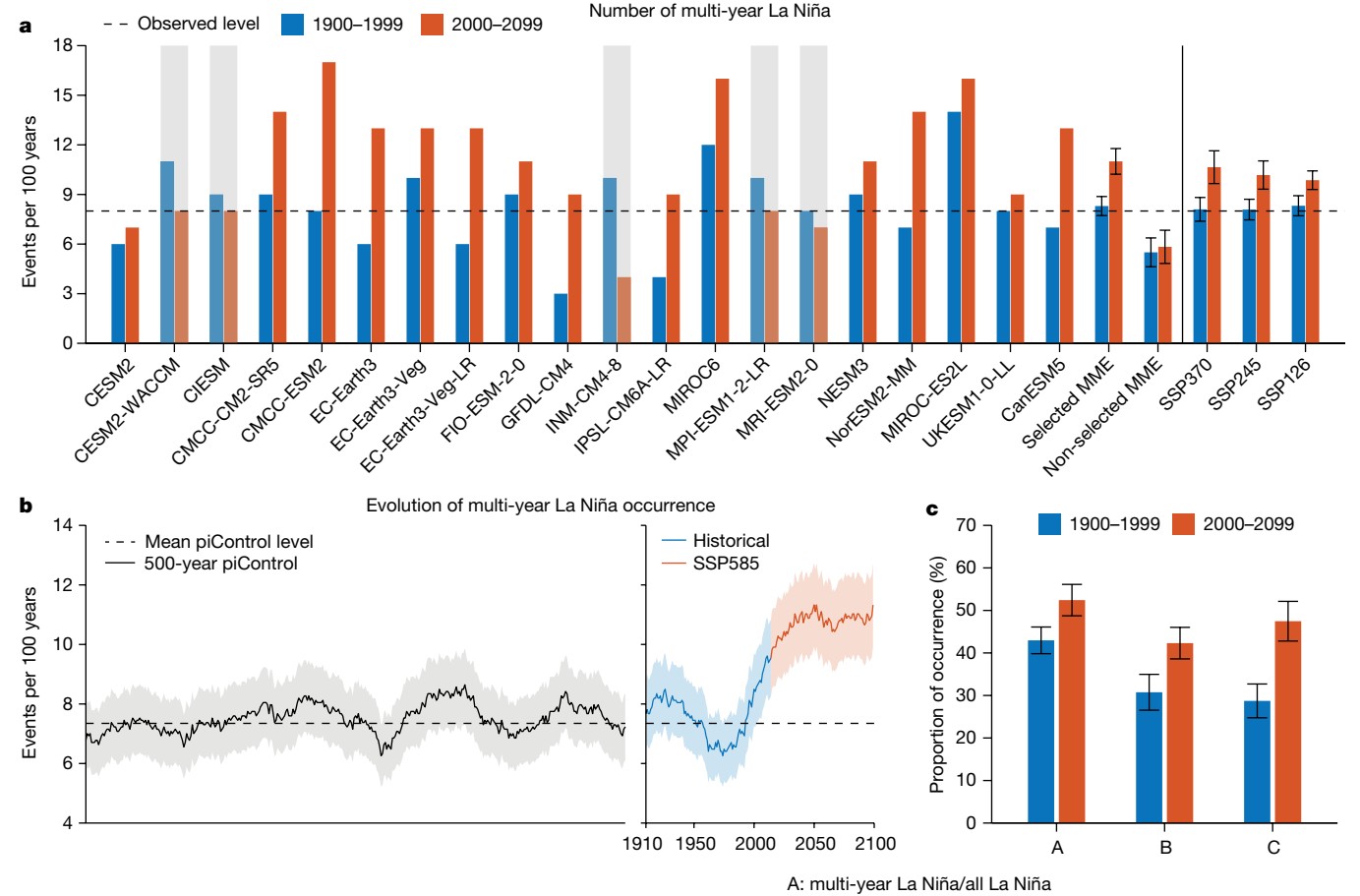

**Fig. 2 | Projected increase in frequency of multi-year La Niña events.**
**a**, Comparison of multi-year La Niña numbers (events per 100 years) over 1900–1999 (blue bars) and 2000–2099 (red bars) in the selected models under SSP585 (left of the vertical line). Multi-model mean results from other emission scenarios are also provided for the selected ensembles. Models that simulate a decrease are greyed out. Shown in the last four columns are the MME results of non-selected models under SSP585 and of selected models under low-emission scenarios. Note that not exactly the same set of models is used under different scenarios owing to data unavailability. The horizontal dashed line indicates observation. **b**, Evolution of multi-year La Niña occurrence (events per 100 years) diagnosed in a 60-year sliding window that moves separately in the past 500 years

of piControl (black) and from 1850 (the start of historical run; blue) to the end of the twenty-first century under SSP585 (red). Years on the x axis denote the end year of the sliding window. Solid lines and shading indicate multi-model mean and 95% confidence intervals based on a Poisson distribution, respectively. The dashed black line indicates the mean level of piControl. **c**, As in **a** but for proportions (as a percentage) of multi-year La Niña occurrences in different situations under SSP585 (see letters on the x axis and corresponding descriptions at the bottom). Error bars on the multi-model mean in **a** and **c** are calculated as 1.0 s.d. of 10,000 inter-realizations of a bootstrap method. Disproportionally more frequent multi-year La Niña events occur after a strong El Niño during the 2000–2099 period than during the 1900–1999 period.

discharge/recharge rate as described in the recharge oscillator theory[24], with a larger El Niño (La Niña) inducing a larger discharge (recharge) rate of the equatorial heat content (Extended Data Fig. 6a).

However, there is no inter-model consensus on changes in the equatorial discharge rate from strong El Niño under greenhouse warming (Extended Data Fig. 6b; other emission scenarios in Supplementary Fig. 5); further, inter-model differences in changes of the heat-discharge rate by strong El Niño are not responsible for inter-model differences in changes of multi-year La Niña numbers (Extended Data Fig. 6c). Occurrence of multi-year La Niña events can be facilitated by tropical inter-basin interactions[17], which seems to be the case for the 2020–2022 La Niña[48]; however, there is no inter-model relationship between changes in multi-year La Niña frequency and changes in the amplitude of climate-variability modes in the subtropical South Pacific and tropical Indian and Atlantic oceans during their respective peak seasons, suggesting no systematic changes in the impacts from these climate modes on the change of multi-year La Niña (Extended Data Fig. 7).

As we will show below, the increased frequency of multi-year La Niña is instead facilitated by an intensified extratropical response to tropical anomalies.

## Intensified extratropical response

In the twenty-first century, equatorial Pacific SST anomalies in D(0) JF(1) (that is, the winter season before the first-year La Niña) are associated with northward-broadened and stronger MAM(1) subtropical northeasterly anomalies (second and third columns of Extended Data Fig. 4), consistent with an atmospheric response that is meridionally broader in scale and stronger in strength. To understand the dynamics, we perform a singular value decomposition (SVD) analysis on D(0) JF(1) SST anomalies in the tropical Pacific (15° S–15° N, 120° E–80° W) and MAM(1) sea-level pressure (SLP) anomalies in the extratropical North Pacific (15° N–60° N, 120° E–80° W) for each of the 20 selected models over the 1900–2099 period. This yields principal patterns and

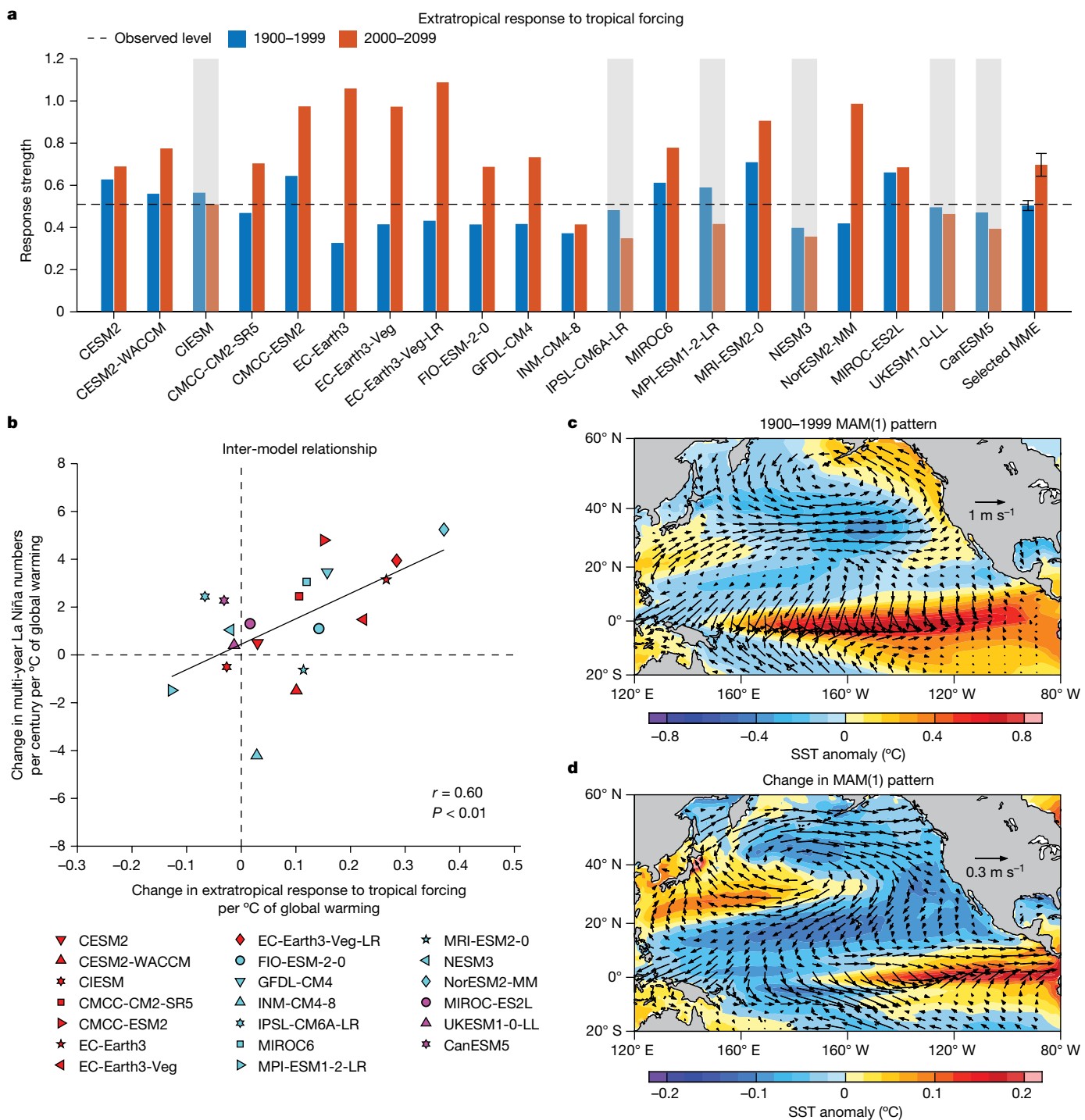

**Fig. 3 | Increased sensitivity of an extratropical response with a northward-broadened anomaly pattern. a**, Comparison of the extratropical response to tropical forcing over the 1900–1999 (blue bars) and 2000–2099 (red bars) periods. Models that simulate a decrease are greyed out. Error bars on the multi-model mean are calculated as 1.0 s.d. of 10,000 inter-realizations of a bootstrap method. The horizontal dashed line indicates observation. **b**, Inter-model relationship between changes (2000–2099 minus 1900–1999) in multi-year La Niña numbers and in the extratropical response to tropical forcing, both scaled by the increase in global mean SST of each model. Linear fit (solid black line) is shown together with correlation coefficient $r$ and corresponding $P$ value.

**c,d**, Multi-model mean pattern of the extratropical response to tropical forcing in MAM(1) during 1900–1999 (**c**) and its change during 2000–2099 (**d**). The response pattern is obtained in each model by first regressing grid-point MAM(1) SST (colouring) and 10-m wind (vectors) anomalies onto the normalized time series of SST expansion coefficient from SVD and then multiplying the regression coefficients by the s.d. of the SST expansion coefficient, separately for 1900–1999 and 2000–2099 (see text for details). Extratropical anomalies are more sensitive to tropical SST forcing and the first La Niña of a multi-year La Niña features further northward-broadened anomalies in the North Pacific during 2000–2099 than during 1900–1999.

associated expansion coefficients of SST and SLP. The principal pattern for the whole Pacific is obtained through regression onto the normalized SST expansion coefficient.

In D(0)JF(1), the leading SVD mode (SVD1) features a pattern with equatorial Pacific warm anomalies; the second SVD mode (SVD2) is characterized by an anomalous warming in the east but cooling in

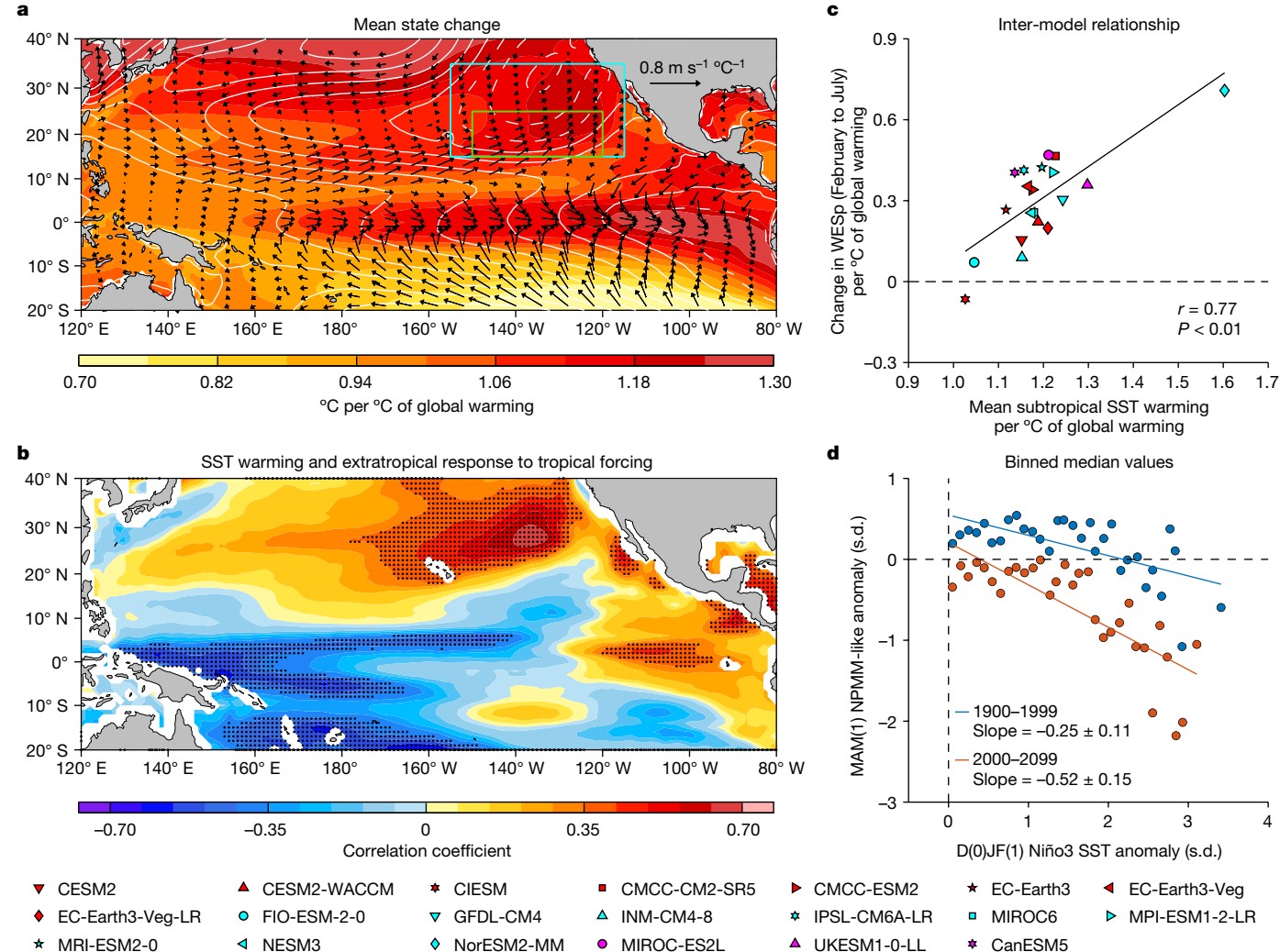

**a** Mean state change

0.8 m s⁻¹ °C⁻¹

0.70  0.82  0.94  1.06  1.18  1.30
°C per °C of global warming

**b** SST warming and extratropical response to tropical forcing

−0.70  −0.35  0  0.35  0.70
Correlation coefficient

**c** Inter-model relationship

*r* = 0.77
*P* < 0.01

Mean subtropical SST warming
per °C of global warming

**d** Binned median values

— 1900–1999
Slope = −0.25 ± 0.11
— 2000–2099
Slope = −0.52 ± 0.15

D(0)JF(1) Niño3 SST anomaly (s.d.)

▼ CESM2    ▲ CESM2-WACCM    ✳ CIESM    ■ CMCC-CM2-SR5    ▶ CMCC-ESM2    ★ EC-Earth3    ◀ EC-Earth3-Veg
◆ EC-Earth3-Veg-LR    ● FIO-ESM-2-0    ▽ GFDL-CM4    △ INM-CM4-8    ✳ IPSL-CM6A-LR    ▢ MIROC6    ▷ MPI-ESM1-2-LR
☆ MRI-ESM2-0    ◀ NESM3    ◆ NorESM2-MM    ● MIROC-ES2L    ▲ UKESM1-0-LL    ✳ CanESM5

**Fig. 4 | Mean state changes facilitate northward-broadened anomalies conducive to multi-year La Niña. a**, Multi-model mean changes of grid-point SST (colouring), SLP (contour; positive in solid lines and negative in dashed lines, with an interval of 5 Pa °C⁻¹) and 10-m wind (vectors) between 1900–1999 and 2000–2099, all scaled by the increase in global mean SST in each of the selected models. **b**, Inter-model correlation between changes (2000–2099 minus 1900–1999) in grid-point mean SST with changes in the strength of extratropical response to tropical forcing, both scaled by the increase in global mean SST of each model. Stippling indicates statistical significance above the 90% level based on a two-tailed Student's *t*-test. **c**, Inter-model relationship between mean SST warming (°C) and WESp change (W s m⁻³, averaged from February to July) in the subtropical northeastern Pacific (155° W–115° W, 15° N–35° N; blue box in **a**). A linear fit (solid black line) is shown, together with correlation coefficient *r* and corresponding *P* value. Note that, even after

excluding a seeming outlier model (NorESM2-MM), the inter-model correlation is still significant (*r* = 0.67, *P* < 0.01). **d**, Relationship between normalized NPMM-like SST anomalies in MAM(1) and normalized Niño3 SST anomalies (5° S–5° N, 150° W–90° W) in D(0)JF(1) for 1900–1999 (blue dots) and 2000–2099 (orange dots). The NPMM-like anomalies (normalized SST anomalies in 15° N–25° N, 150° W–120° W; green box in **a**) are binned in 0.1 s.d. intervals of Niño3 SST anomalies to obtain median values (circles) in each bin first. Only positive Niño3 SST anomalies are considered here. Also shown are respective slopes with 95% confidence intervals for the two periods. Warming maximum in the subtropical northeastern Pacific enhances the local thermodynamic response to convective anomalies from El Niño, generating anomalous easterlies at further northern latitudes than in 1900–1999, and warming maximum in the eastern equatorial Pacific enhances the convective response, both contributing to the increased frequency of multi-year La Niña.

the western equatorial Pacific (Extended Data Fig. 8a–f). The sum of these first two modes reflects eastern Pacific El Niño that tends to be strong[37–39]. Considering that a substantial portion (more than 83%) of the increased multi-year La Niña events occur after a strong El Niño, we use a combination of SVD1 + SVD2 and their normalized time series of SST and SLP expansion coefficients to depict the influence on multi-year La Niña events associated with a preceding strong El Niño. Two time series of combined expansion coefficients, one for D(0)JF(1) SST anomaly pattern and the other for MAM(1) SLP, are constructed. The months for the average of each field result in a relationship that SST leads SLP by three months. The total extratropical response to tropical forcing is calculated, separately over the twentieth and twenty-first centuries, as a regression coefficient of the SLP time series onto the combined

SST time series (Extended Data Fig. 8g,h) multiplied by the standard deviation. of the SST time series. The spatial pattern is obtained by lagged linear regression of the SST and 10-m wind anomalies onto the combined SST expansion coefficient multiplied by its standard deviation, separately for each century.

In the twenty-first century, the extratropical response to tropical forcing substantially increases with a strong inter-model consensus (70%; Fig. 3a). Models that simulate a greater increase in the extratropical response systematically produce a larger increase in multi-year La Niña frequency, with an inter-model correlation coefficient of 0.60 that is statistically significant above the 99% confidence level (*P* < 0.01) (Fig. 3b). The extratropical Pacific response to El Niño warm anomalies is a negative NPMM-like response in MAM(1) that features

northward-broadened northeasterly wind anomalies (Fig. 3c,d). It is this broadened meridional structure that is conducive to persistence of the cold anomalies of the first-year La Niña into the second year[6,7], contributing to increased frequency of multi-year La Niña events.

The importance of the northward-broadened negative NPMM-like anomaly in MAM(1) is highlighted by the fact that most (80%, 16 out of 20) of the models project an increase in multi-year La Niña events that co-occur with a negative NPMM-like event in MAM(1) (Extended Data Fig. 9a; other emission scenarios in Supplementary Fig. 6), defined as when the normalized SST anomalies in the subtropical North Pacific (15° N–25° N, 150° W–120° W) is greater than 0.5 s.d. in amplitude[31]. Among these concurrent events, there is a much larger proportion, from $31.7 \pm 4.3\%$ in 1900–1999 to $58.1 \pm 5.5\%$ in 2000–2099, of events in which the negative NPMM-like event is preceded by an El Niño (Extended Data Fig. 9b). The northward-broadened negative NPMM-like anomaly in MAM(1) is conducive to multi-year La Niña as it widens the meridional structure of the first-year La Niña and the associated weakening of the negative wind stress curl anomalies slows the upper-ocean heat recharge, creating a colder precondition easier for the cold SST anomalies to persist through the decay phase of the first La Niña (Extended Data Fig. 10).

Because ocean–atmosphere coupling in the equatorial Pacific intensifies under greenhouse warming[37], the persisting cold SST anomalies of the first La Niña are amplified by the stronger Bjerknes positive feedback in late boreal summer–autumn, more likely to develop into a second La Niña in the following winter (Supplementary Fig. 7). Below we show that the more sensitive extratropical response of the northward-broadened anomaly pattern is facilitated by the mean state change, particularly a faster warming in the equatorial eastern and the subtropical northeastern Pacific.

## Conducive background-warming pattern

Compared with the twentieth century, mean state changes in the twenty-first century feature two warming maxima separated by a local warming minimum around 10° N, one in the equatorial eastern Pacific and the other in the subtropical northeastern Pacific, accompanied by trends of equatorial westerlies and subtropical southwesterlies, respectively[49] (Fig. 4a). An inter-model correlation shows that the more sensitive extratropical response to the tropical forcing is systematically linked to the two warming maxima (Fig. 4b).

The faster warming in the subtropical northeastern Pacific, which is contributed by increased shortwave and longwave radiation into the ocean (Supplementary Fig. 8), enhances the WES feedback more to the north, such that the NPMM-like SST and surface wind anomalies induced by El Niño warm SST anomalies are broadened northward and are more sensitive. As a measure of the intensity of the WES feedback, we calculate change in latent heat flux per unit change in zonal wind speed[26,50,51] (see 'Diagnosis of the WES feedback' in Methods); under greenhouse warming, an increase in this parameter is seen in most models over the subtropical northeastern Pacific (15° N–35° N, 155° W–115° W), and models that simulate a greater subtropical mean SST warming systematically produce a stronger WES feedback from February to July ($r = 0.77$, $P < 0.01$; Fig. 4c).

The faster warming in the equatorial eastern Pacific promotes a stronger atmospheric convection response to an SST anomaly (Supplementary Fig. 9a, b), such that El-Niño-induced Gill-type atmospheric response and poleward atmospheric teleconnection are more sensitive even if the El-Niño-related SST variability remains unchanged[52], conductive to occurrences of the northward-broadened negative NPMM-like pattern in the following spring. Under greenhouse warming, the nonlinear response of atmosphere convection to SST anomalies initially intensifies, as the eastern equatorial Pacific warms faster than other tropical ocean regions[36]. However, after the background SST surpasses a threshold warming, the convective response

plateaus, limited by surface wind convergence that supports the convection[53]. The plateau in multi-year La Niña frequency despite continuing increase of radiative forcing in the second half of the twenty-first century (Fig. 2b) is a downstream consequence of the plateau in the convective response (Supplementary Fig. 9c).

The joint impact from the two warming maxima under greenhouse warming is a negative NPMM-like pattern in response to a positive eastern Pacific Niño3 SST anomaly in D(0)JF(1) that is not only northward broadened but also more sensitive to El Niño warm anomalies in the twenty-first century (Fig. 4d). As the pattern of northward-broadened easterly anomalies occurs more frequently owing to the increased sensitivity, the associated heat recharge of the equatorial Pacific slows, leading to a colder upper-ocean condition and more frequent multi-year La Niña events (Supplementary Fig. 10). That these mechanisms operate is underscored by an inter-model correlation after excluding the strong El Niño events, in which models that simulate the two stronger warming maxima similarly simulate a greater increase in multi-year La Niña events (Supplementary Fig. 11).

## Summary and discussion

Our finding of an increase in the occurrence of consecutive La Niña events under greenhouse warming is underpinned by northward-broadened easterly anomalies in the subtropical North Pacific in response to equatorial eastern Pacific warm anomalies. The northward broadening and its increased occurrences are—in turn—a consequence of a faster mean-state warming in the subtropical northeastern Pacific that induces a further northern and more sensitive response to El Niño convective anomalies, which are—per se—intensified by a faster warming in the equatorial eastern Pacific. The consequence of the northward-broadened easterlies is a slower heat recharge of the equatorial Pacific, leaving a colder upper-ocean condition after the first-year La Niña to persist into the second year. Our discovery of a two-way interaction between the tropics and subtropics that intensifies under greenhouse warming represents an advance beyond recent findings of a one-way warming-induced enhancement of the NPMM influence on ENSO[50,51]. Our result of a probable future increase in multi-year La Niña frequency strengthens calls for an urgent need to reduce greenhouse-gas emissions to alleviate the adverse impacts.

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

## Methods

### Observational and CMIP6 data

We use three SST reanalysis products to characterize observed multi-year La Niña events in 1900–2021, including HadISST v1.1 (Hadley Centre Sea Ice and Sea Surface Temperature dataset version 1.1)[54], ERSST v5 (Extended Reconstructed Sea Surface Temperature version 5)[55] and COBE-SST 2 (Centennial in situ Observation-Based Estimates of Sea Surface Temperature version 2)[56]. We also use surface wind stress and SLP from the National Centers for Environmental Prediction (NCEP)/National Center for Atmospheric Research (NCAR) reanalysis[57] and subsurface ocean temperature from IAP global ocean temperature gridded product[58] covering the period 1948–2021. SSH is taken from the European Centre for Medium-Range Weather Forecasts (ECMWF) Ocean Reanalysis System 5 (ORAS5)[59]. Monthly anomalies of all observational variables are constructed with reference to the monthly climatology of the full period in each dataset and quadratically detrended. Using different periods to compute the climatology does not alter our results.

We take outputs from 37 CMIP6 models over the 1900–2099 period, in which monthly data are available for ocean temperature, SST, surface winds, SLP, heat flux and SSH. These models are forced under historical anthropogenic and natural forcings up to 2014 and thereafter future greenhouse-gas forcing under the SSP585 emission scenario till 2100 (ref. 14). We compare changes in multi-year La Niña events between the twentieth (1900–1999) and twenty-first (2000–2099) centuries. Monthly anomalies of all variables from models are obtained with reference to the monthly climatology of 1900–1999 and quadratically detrended. As expected, the definition of events and the choice of period over which climatology is calculated could affect our results. As a test, we calculate anomalies separately for 1900–1999 and 2000–2099 with reference to the respective monthly climatology. Although changes in individual models occur, the MME results hold (Supplementary Fig. 12). Before the analysis, outputs of each model are re-gridded into a common 1° × 1° resolution.

As in previous studies[37,50,51], we use one experiment only from each model (the 'model democracy' approach) to avoid dominance by models with which many experiments are carried out such that each model is represented equally in the assessment of inter-model consensus and the ensemble mean change. Projected changes in ENSO might be subject to internal variability[46,60]. For a given model, the longer the time window used to diagnose ENSO variability change, the lower the noise level of unforced natural variability compared with a warming-induced ENSO SST variability change signal[42]. Using two 100-year periods for diagnosis could largely eliminate the influence from internal variability[42,61], although detailed time-varying features within each period may not be well captured[62]. We use a multi-century pre-industrial control (piControl) simulation of each model to examine the influence of internal variability in the ENSO cycle and the results show distinct increase in the frequency of multi-year La Niña emerging from background natural variability in the twenty-first century (Fig. 2b). We also use all available models under different emission scenarios (SSP126, SSP245 and SSP370; Supplementary Table 1) to test the sensitivity of our results.

### Definition of multi-year La Niña events

La Niña is commonly defined as when the Niño3.4 index (that is, SST anomalies in 5° S–5° N, 170° W–120° W) is below a certain threshold for several months, for instance, below −0.5 °C for a minimum of five consecutive months in observations. However, ENSO amplitude varies vastly across models and, therefore, we use a threshold of 0.5 s.d. to detect El Niño/La Niña in both observation and CMIP6 model outputs. Specifically, a La Niña (an El Niño) event is defined as when the Niño3.4 SST anomaly over ONDJF is below (above) the −0.5 (0.5) s.d. of the ONDJF Niño3.4 SST values in 1900–1999, which is 0.44 °C averaged over three observational reanalysis datasets and 0.54 ± 0.16 °C across the 20 selected CMIP6 models. A multi-year event is identified as when

the La Niña (El Niño) persists for two consecutive ONDJF seasons or more. As in previous studies[6,7,19], if a La Niña lasts for three years, the first two years are applied to our analysis. We denote the year that precedes a La Niña as year (0) and the following two years when La Niña develops and re-intensifies as year (1) and year (2), respectively. According to this criterion, we extract eight multi-year La Niña and ten single-year La Niña events in the twentieth century (1900–1999; Extended Data Fig. 1a), which are consistent with previous studies[5,6,9,19] and coincide with those obtained from the Oceanic Niño Index (ONI) provided by the National Oceanic and Atmospheric Administration (NOAA; https://origin.cpc.ncep.noaa.gov/products/analysis_monitoring/ensostuff/ONI_v5.php).

### Model bias and recent multi-year La Niña change

We have assessed potential impacts from a common SST bias in the equatorial Pacific. Although most models still simulate a too-cold climatological cold tongue in the equatorial Pacific, there is no significant ($P = 0.34$) inter-model relationship between the intensity of the cold-tongue bias and projected change in multi-year La Niña frequency under greenhouse warming, suggesting that the mean SST bias does not systematically affect our results.

On recent observed changes, the last 40 years seem to have witnessed more frequent multi-year La Niña events under a stronger mean-state warming in the western than the eastern tropical Pacific[63]. However, the current observational record is subject to much uncertainty, especially in the first half of the twentieth century[64], meaning that the observed change itself may not be relied on without any further corroborating evidence. Although models tend to underestimate the recent SST warming trend between the western and eastern equatorial Pacific[63], there are models that are able to simulate both an increase in multi-year La Niña frequency and a faster warming in the western than the eastern equatorial Pacific in 1981–2020 than 1941–1980 (Supplementary Fig. 13a), suggesting an influence from natural variability on recent multi-year La Niña change.

Further, our sensitivity test indicates that, even if data quality was not an issue, detecting observed change may depend on the time period in which the frequency of multi-year La Niña is diagnosed. For example, when comparing 1945–1979 with 1980–2014, there is no change in the frequency of multi-year La Niña events in observations (Extended Data Fig. 1a), although there is still a strengthened west-minus-east SST gradient in the equatorial Pacific (Supplementary Fig. 13b). Therefore, the observed multi-year La Niña change is subject to uncertainty and may be influenced by natural variability.

### Sensitivity of projected increase in multi-year La Niña occurrence

We have tested the sensitivity of identifying multi-year La Niña events to different thresholds or methods. For example, using a more stringent threshold of 0.7 s.d., which is 0.62 °C in the reanalysis and 0.75 ± 0.23 °C across the models, the number of multi-year La Niña events increases in 14 of the 20 (70%) selected models from 1900–1999 to 2000–2099, with an MME increase of about 60% that is statistically significant above the 95% confidence level according to a bootstrap test[65]. The occurrence ratio of multi-year La Niña relative to all La Niña events also increases notably, with an MME change from 33.3 ± 2.5% in 1900–1999 to 44.4 ± 3.9% in 2000–2099. If a multi-year La Niña event is alternatively defined as when the Niño3.4 index falls below −0.75 s.d. in any month during October (0) to February (1) and remains below −0.5 s.d. in any month during October (1) to February (2) (the s.d. is calculated separately for each month)[9], we obtain an essentially identical result, that is, multi-year La Niña events occur more frequently in the twenty-first century, supported by a statistically significant MME increase and strong inter-model consensus (Supplementary Fig. 3).

A sensitivity test finds that, even if all the models are considered, there is still a substantial multi-model mean increase (24.3 ± 13.2%) of multi-year La Niña events, with 24 out of 37 (65%) models showing an increase. Sensitivity to warming scenarios suggests that the

increase in multi-year La Niña numbers is more evident under higher greenhouse-gas forcings, as seen in a larger number of models simulating a marked increase under SSP585, SSP370 and SSP245 compared with SSP126 (Supplementary Figs. 2 and 3). On the one hand, the increase with intensity of greenhouse-gas emissions further highlights the underpinning role of greenhouse warming. On the other hand, this implies that a continued reduction of future greenhouse-gas emissions may help mitigate the adverse climatic influences associated with multi-year La Niña that would otherwise increase progressively into the future.

Further, our analysis indicates that there is no inter-model consensus on the change of triple La Niña events from 1900–1999 to 2000–2099, even under a high-emission warming scenario SSP585. Only a total of 8 out of 20 (40%) models show an increase in triple La Niña events with an MME increase of 12.2 ± 28.1%. Triple La Niña is relatively rare in the historical record and its mechanism is still uncertain. A preliminary analysis indicates that there is no systematic change in the meridional structure of the second-year La Niña between 1900–1999 and 2000–2099, which may provide a possible explanation for why there is no inter-model consensus on the change in triple La Niña frequency under greenhouse warming.

### Diagnosis of the WES feedback

The WES feedback refers to a coupled air–sea feedback loop, in which a weakened surface wind, a reduction of evaporation and associated latent heat flux cooling, and warm SST anomalies reinforce each other, and vice versa[34]. The WES feedback relies on thermodynamic processes and its intensity is usually estimated by a WES parameter (WESp), defined as a change in latent heat flux per unit change in zonal wind speed[26,50,51],

$$\text{WESp} = -\frac{\partial \text{LH}}{\partial u} = -\text{LH}\frac{u}{W^2} \tag{1}$$

in which LH denotes latent heat flux, $u$ is the 10-m zonal wind and $W$ is the total wind speed. The intensity of WESp depends on the mean state. A stronger mean zonal wind and a warmer background mean SST corresponds to a more intense WES feedback (that is, a larger WESp)[50].

### Statistical significance test

Given the limited number of CMIP6 models, a bootstrap method[65] is used to examine whether the multi-model mean increase in multi-year La Niña occurrences is statistically significant. Specifically, the 20 values of multi-year La Niña numbers in 1900–1999 (blue bars in Fig. 2a) are resampled randomly to construct 10,000 realizations of a multi-model mean over the 20 models. In this random resampling process, any value of the number is allowed to be selected again. The same is carried out for the 2000–2099 period (red bars in Fig. 2a). We calculate the s.d of the 10,000 realizations of mean value for the two periods, and if the difference of the multi-model mean value between the two periods is greater than the sum of the two separate 10,000-realization s.d. values, the difference is considered statistically significant above the 95% confidence level. This bootstrap test is also used to check the occurrence ratio of multi-year La Niña, the strength of extratropical response to tropical forcing and multi-model mean differences in the composite analyses. In terms of the evolution of multi-year La Niña frequency (Fig. 2b), we choose to test its significance using a Poisson distribution, which is suitable for a discrete probability distribution that expresses the probability of a given number of events occurring in a fixed interval of time. Using a bootstrap test instead does not alter the result.

### Data availability

Data related to the paper can be downloaded from the following websites: HadISST v1.1, https://www.metoffice.gov.uk/hadobs/hadisst/; ERSST v5, https://psl.noaa.gov/data/gridded/data.noaa.ersst.v5.html; COBE-SST 2, https://psl.noaa.gov/data/gridded/data.cobe2.html; NCEP-NCAR Reanalysis 1, https://psl.noaa.gov/data/gridded/data.ncep.reanalysis.html; IAP data, http://www.ocean.iap.ac.cn/pages/dataService/dataService.html?navAnchor=dataService; ORAS5, https://www.ecmwf.int/en/research/climate-reanalysis/ocean-reanalysis; CMIP6 datasets, https://esgf-node.llnl.gov/projects/cmip6/.

### Code availability

Codes for the main results are available on Zenodo at https://doi.org/10.5281/zenodo.7885442.

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

**Acknowledgements** This study is supported by the Science and Technology Innovation Project of Laoshan Laboratory (LSKJ202203300) and the Strategic Priority Research Program of Chinese Academy of Sciences (XDB 40030000). T.G. is supported by National Natural Science Foundation of China (NSFC) projects (42206209, 42276006) and China National Postdoctoral Program for Innovative Talents (BX20220279). F.J. is supported by the National Key Research and Development Program of China (2020YFA0608801), LSKJ202202402, NSFC (41876008) and Youth Innovation Promotion Association of Chinese Academy of Sciences (2021205). B.G. is supported by LSKJ202202602 and NSFC (42276016). S.L. is supported by NSFC (42006173 and 92058203). Pacific Marine Environmental Laboratory (PMEL) contribution number 5457.

**Author contributions** T.G., F.J. and W.C. designed the study and wrote the initial manuscript, in discussion with L.W. T.G. performed analysis and generated all figures with F.J. All authors contributed to interpreting the results and improving the paper.

**Funding** Open access funding provided by CSIRO Library Services.

**Competing interests** The authors declare no competing interests.

**Additional information**
**Correspondence and requests for materials** should be addressed to Fan Jia or Wenju Cai.

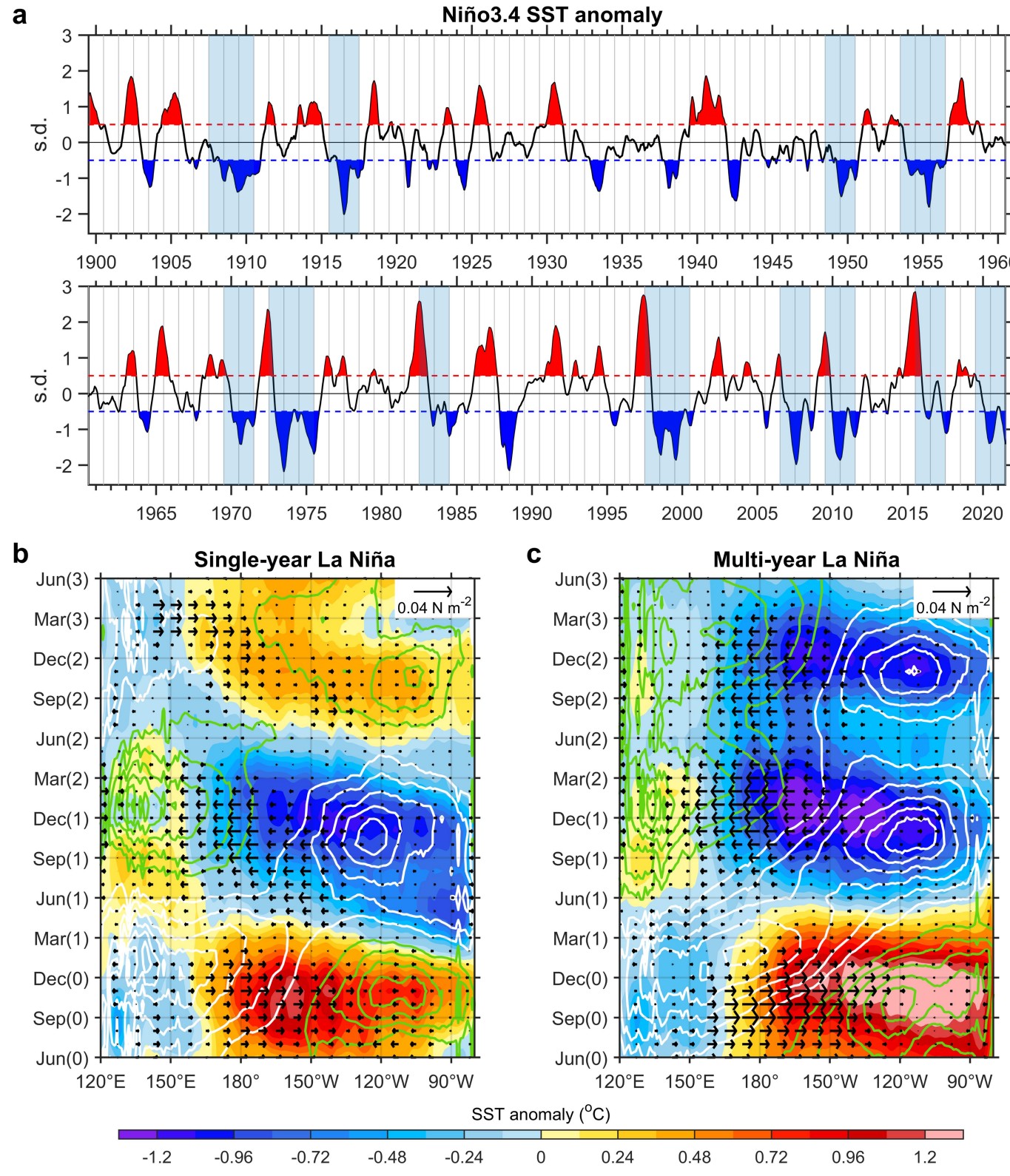

**Extended Data Fig. 1 | Observed multi-year La Niña events. a**, Time series of Niño3.4 SST anomaly in 1900–2021 averaged from several reanalysis datasets[54–56]. The time series is scaled by ONDJF s.d. of Niño3.4 SST in 1900–1999 and smoothed with a three-month running-mean filter. The red and blue dashed lines indicate the 0.5 s.d. and −0.5 s.d. thresholds for identifying El Niño (red) and La Niña (blue) events, respectively. The vertical blue shadings mark the years of multi-year La Niña events. **b,c**, Time–longitude evolution of equatorial (5° N–5° S average) SST (°C; colouring), depth of 20 °C isotherm (m; contours; positive in green and negative in white, with an interval of 3 m) and surface zonal wind stress (N m$^{-2}$; vectors) anomalies, composited for single-year (**b**) and multi-year (**c**) La Niña events, respectively. Ocean temperature and surface wind data are from IAP[58] and NCEP/NCAR reanalysis[57], respectively, in 1948–2021.

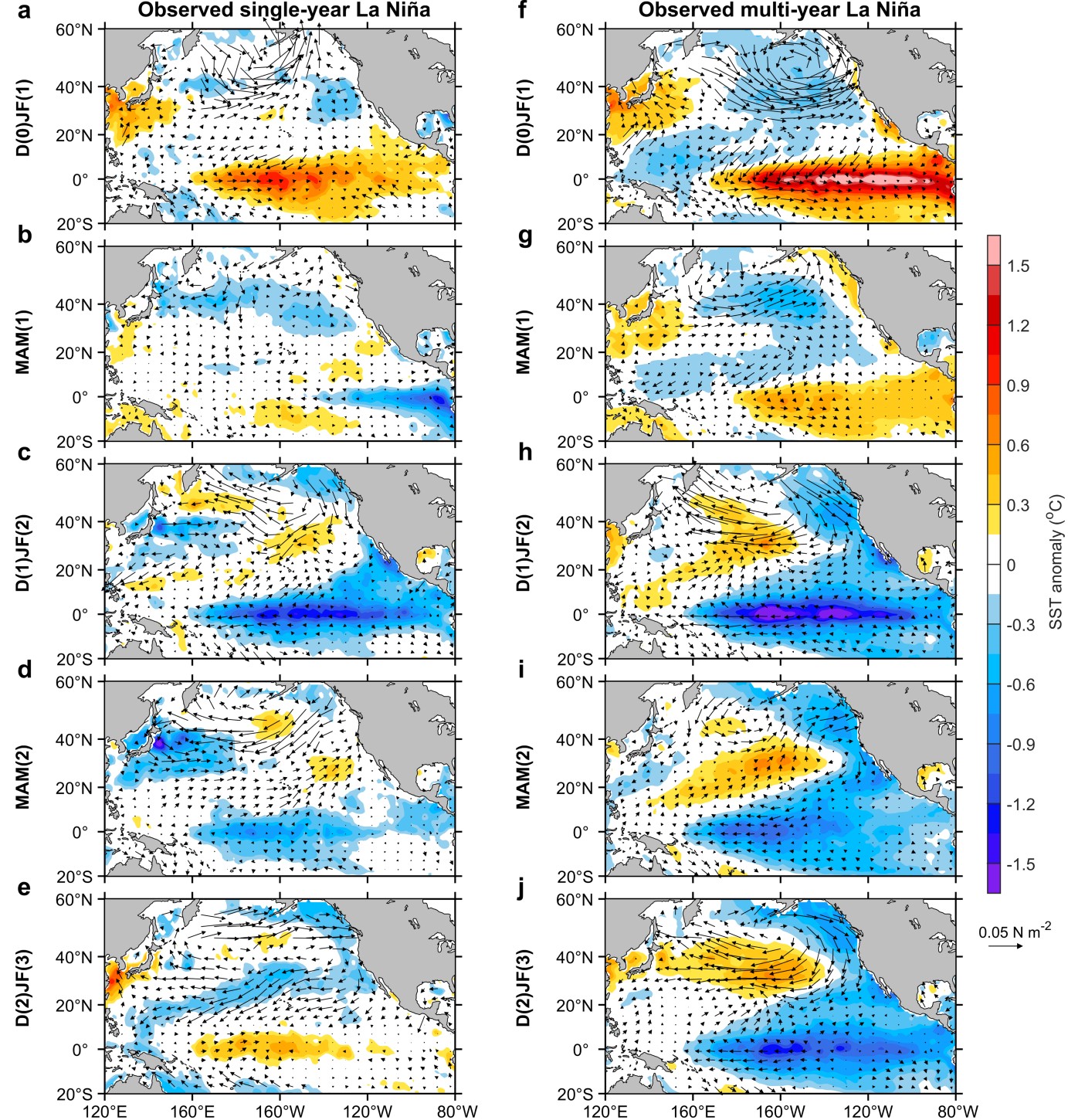

**Extended Data Fig. 2 | Spatial evolution of single-year and multi-year La Niña events in observation. a–e**, Composite maps of anomalous SST (°C; colouring) and surface wind stress (N m$^{-2}$; vectors) for single-year La Niña events during D(0)JF(1) (**a**), MAM(1) (**b**), D(1)JF(2) (**c**), MAM(2) (**d**) and D(2)JF(3) (**e**).

**f–j**, Same as **a–e** but for multi-year La Niña events. SST and surface wind data are from HadISST[54] and NCEP/NCAR reanalysis[57], respectively, in 1948–2021. The first La Niña of a multi-year La Niña event features a northward-broadened anomaly pattern.

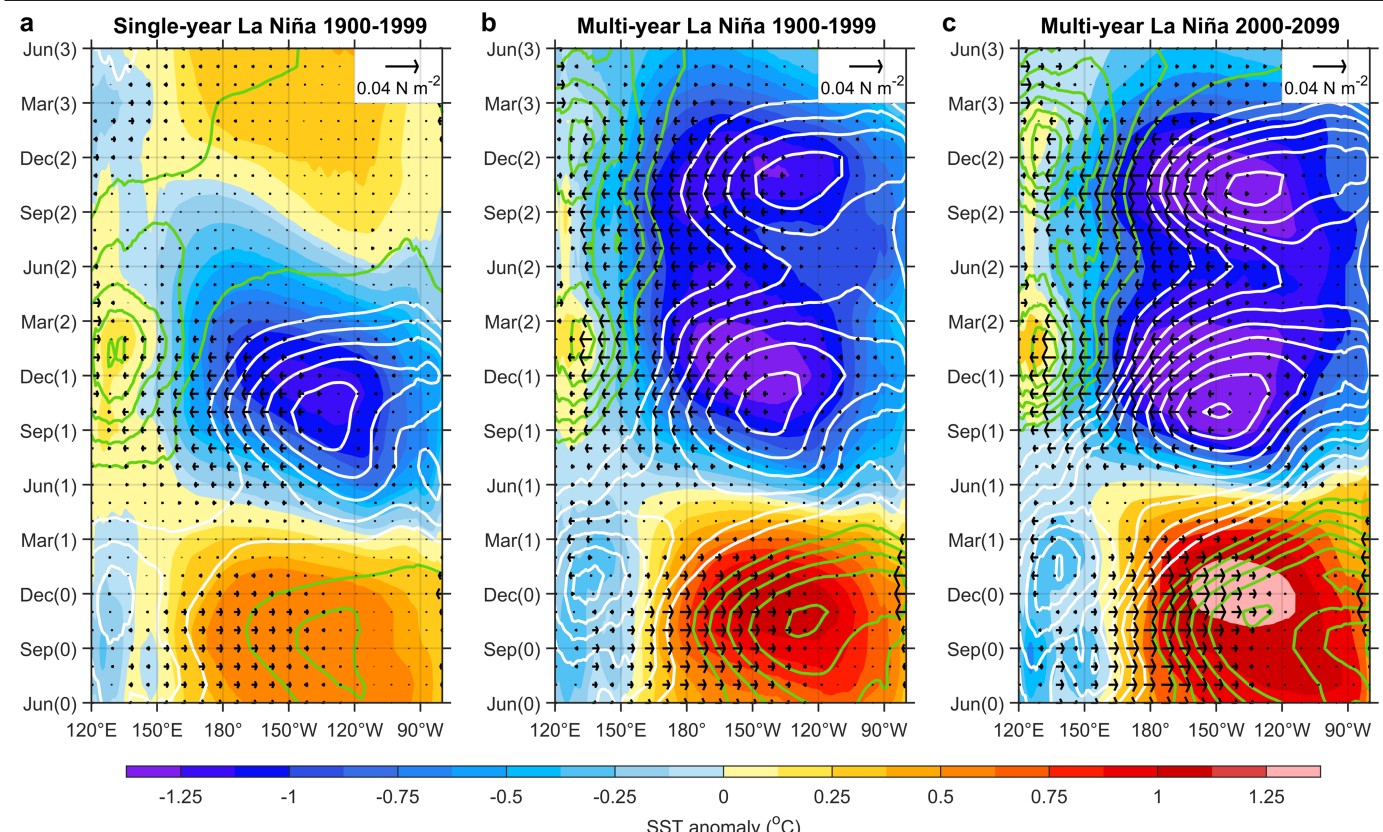

**Extended Data Fig. 3 | Temporal evolution of single-year and multi-year La Niña events in models.** Time–longitude evolutions of equatorial (5° N–5° S average) SST (°C; colouring), SSH (m; contours; positive in green and negative in white, with an interval of 0.01 m) and surface zonal wind stress (N m⁻²; vectors) anomalies, composited for single-year La Niña events in 1900–1999 (**a**), multi-year La Niña events in 1900–1999 (**b**) and in 2000–2099 (**c**), respectively, in the selected models. Models simulate reasonably well the observed multi-year La Niña evolution.

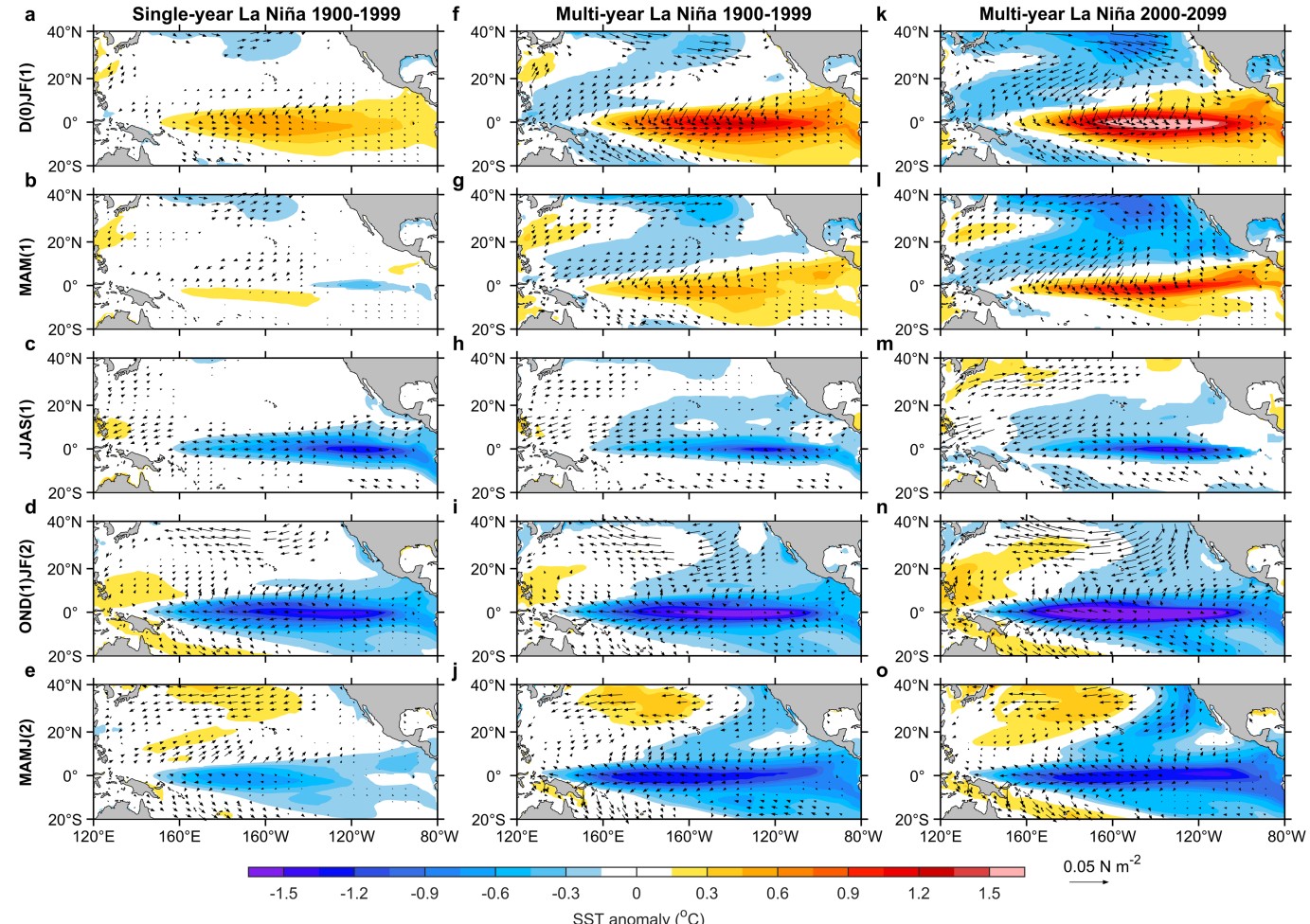

**Extended Data Fig. 4 | Spatial evolution of single-year and multi-year La Niña events in models.** Composite maps of SST (°C; colouring) and surface wind stress (N m⁻²; vectors) anomalies for simulated single-year La Niña events in 1900–1999 during D(0)JF(1) (**a**), MAM(1) (**b**), JJAS(1) (**c**), OND(1)JF(2) (**d**) and MAMJJ(2) (**e**) in the selected models. **f–j**,**k–o**, Same as **a–g** but for multi-year La Niña events in 1900–1999 and 2000–2099, respectively. Shown are values at which the ensemble mean exceeds 1.0 s.d. according to a bootstrap method. The first La Niña of a multi-year La Niña event during 2000–2099 features further northward-broadened anomalies in the North Pacific than during 1900–1999.

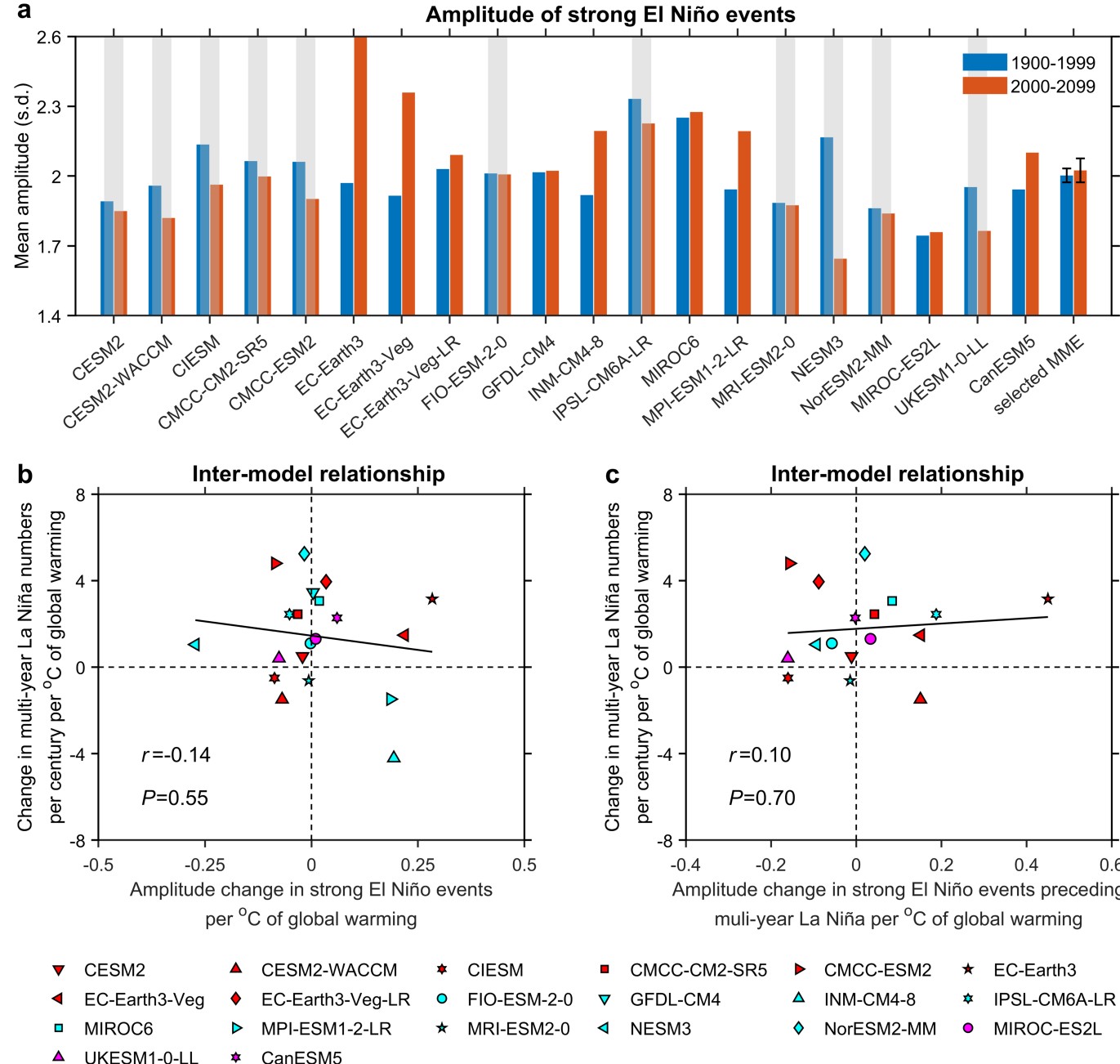

**Extended Data Fig. 5 | No impact from changing amplitude of strong El Niño. a**, Comparison of mean amplitude (s.d.) of strong El Niño events over 1900–1999 (blue bars) and 2000–2099 (red bars) in the selected models. Models that simulate a decrease are greyed out. Error bars on the multi-model mean are calculated as 1.0 s.d. of 10,000 inter-realizations of a bootstrap method. **b**, Inter-model relationship between changes (2000–2099 minus 1900–1999) in multi-year La Niña numbers and in the mean amplitude of all strong El Niño events, both scaled by the increase in global mean SST of each model. **c**, As in **b** but for strong El Niño events that precede multi-year La Niña. Linear fits (solid black line) are shown together with correlation coefficient *r* and corresponding *P*-value in **b**,**c**. Note that GFDL-CM4 and MPI-ESM1-2-LR do not simulate a strong El Niño preceding multi-year La Niña in 1900–1999 (that is, no data for the two models in the *x* axis of **c**). There is no inter-model consistent change in amplitude of strong El Niño events, indicating that it is not responsible for the increased frequency of multi-year La Niña.

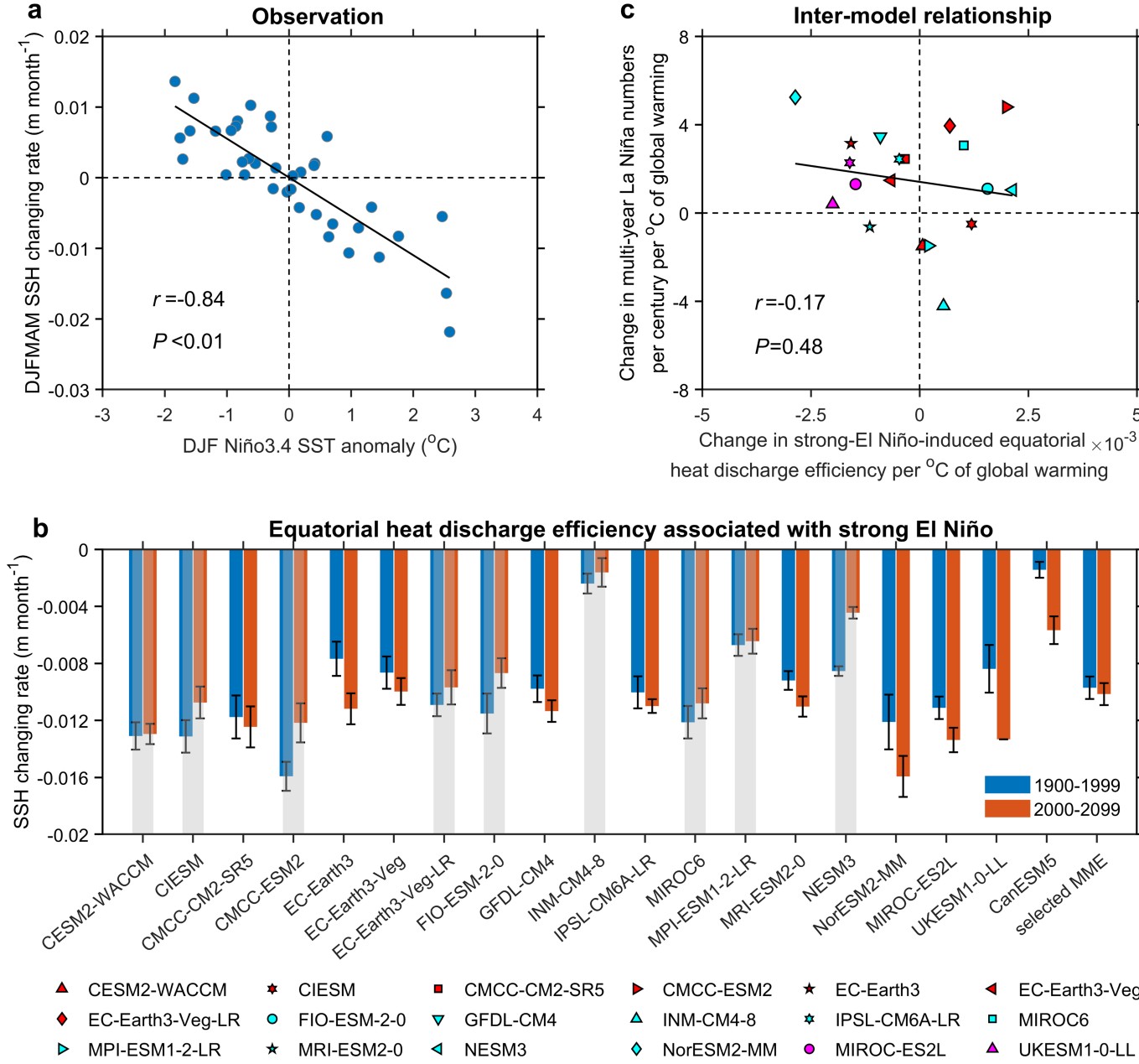

**Extended Data Fig. 6 | No impact from changing equatorial heat-discharge rate associated with strong El Niño. a**, Observational relationship between DJF Niño3.4 SST anomalies (°C) and DJFMAM changing rate of SSH (a surrogate for thermocline depth) anomalies (m month$^{-1}$) in the equatorial Pacific (120° E–80° W, 5° N–5° S), based on ORAS5 in 1979–2017 (ref. 59). **b**, Comparison of mean DJFMAM changing rate of SSH in strong El Niño years, which measures the equatorial heat-discharge rate associated with strong El Niño events, over 1900–1999 (blue bars) and 2000–2099 (red bars) in the selected models. Models that simulate a decrease are greyed out. Error bars denote 1.0 s.d. of 10,000 inter-realizations using a bootstrap method of the samples for each

model simulation and the MME mean. **c**, Inter-model relationship between changes (2000–2099 minus 1900–1999) in multi-year La Niña numbers and in the mean equatorial heat discharge (DJFMAM SSH changing rate) associated with strong El Niño events, both scaled by the increase in global mean SST of each model. Linear fits (solid black line) are shown together with correlation coefficient *r* and corresponding *P*-value in **a** and **c**. Only models with available SSH data are shown here. Equatorial Pacific heat-discharge rate associated with strong El Niño shows no consistent changes across models and is not responsible for the increased frequency of multi-year La Niña.

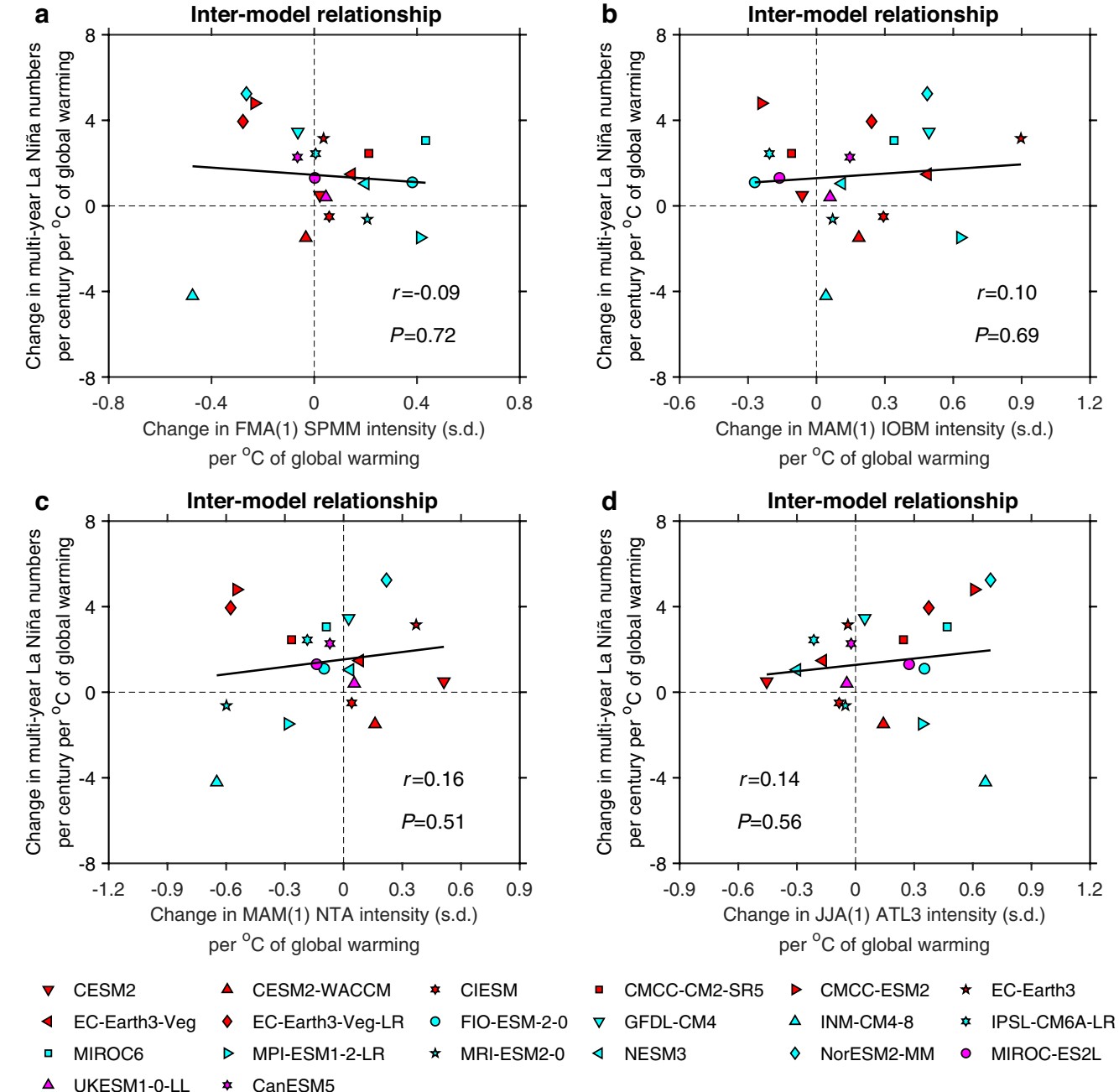

**Extended Data Fig. 7 | No impact from changing amplitude of climate-variability modes in the subtropical South Pacific and tropical Indian and Atlantic oceans.** Inter-model relationship between changes (2000–2099 minus 1900–1999) in multi-year La Niña numbers and in the mean amplitude of the South Pacific Meridional mode (SPMM) in FMA(1) (**a**), the Indian Ocean Basin Mode (IOBM) in MAM(1) (**b**), the North Tropical Atlantic (NTA) SST anomalies in MAM(1) (**c**) and the Atlantic Niño/Niña (measured by the ATL3 index) in JJA(1) (**d**). Linear fits (solid black lines) are shown together with correlation coefficient *r* and corresponding *P*-value. Here the SPMM index is defined as normalized SST anomalies over the subtropical southeastern Pacific (15° S–25° S and 110° W–90° W), the IOBM is defined as the first empirical orthogonal function mode of SST anomalies in the tropical Indian Ocean (20° S–20° N and 40° E–110° E), the NTA index is taken as normalized SST anomalies in the north tropical Atlantic (10° N–20° N and 60° W–20° W) and the ATL3 index is taken as normalized SST anomalies in the equatorial Atlantic cold-tongue region (3° S–3° N and 20° W–0° E). There is no inter-model relationship between changes in multi-year La Niña frequency and changes in the mean amplitude of climate-variability modes in the subtropical South Pacific and tropical Indian and Atlantic Oceans during their respective peak seasons, suggesting no systematic changes in the impact from these climate modes on the change of multi-year La Niña frequency.

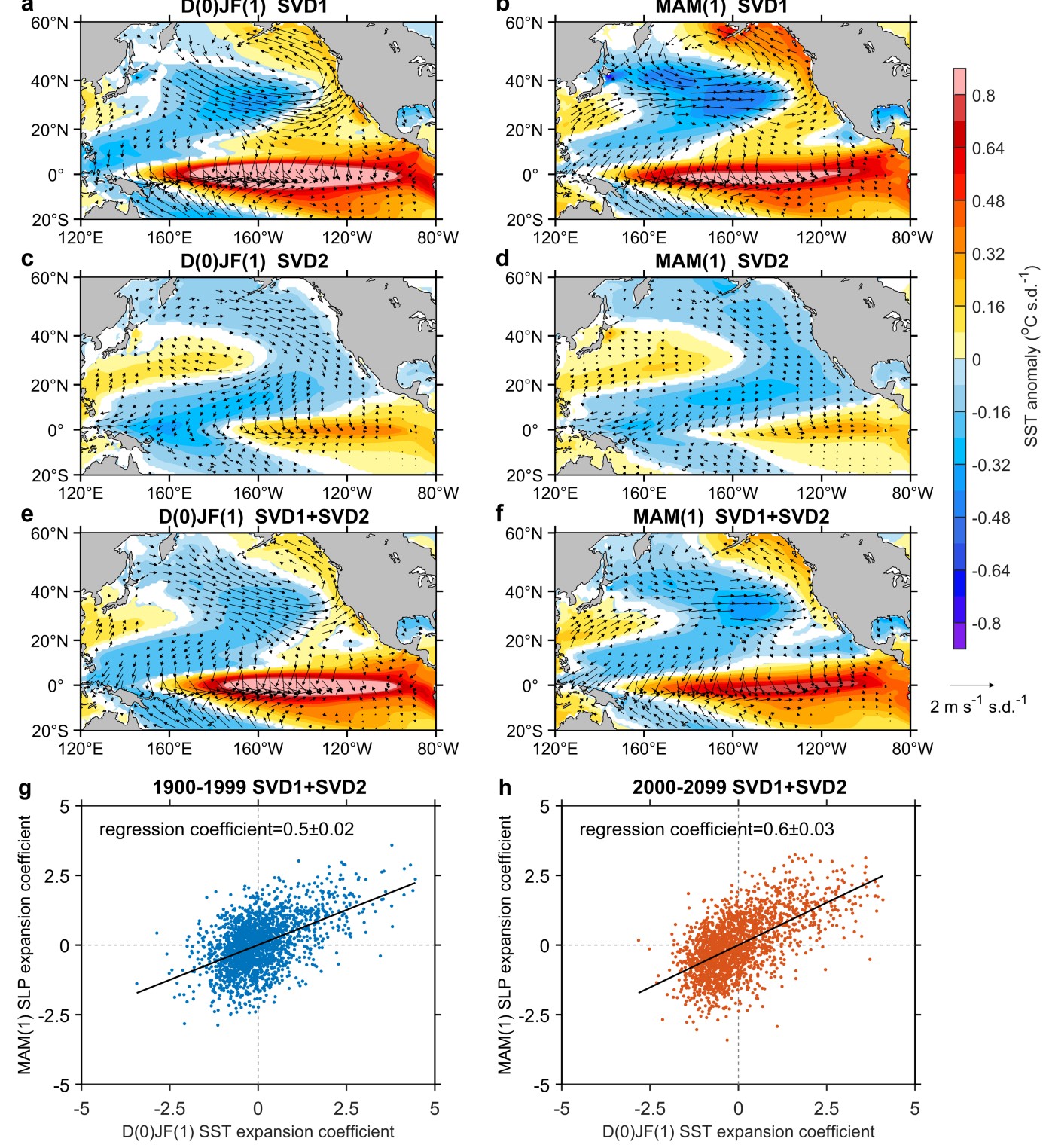

**Extended Data Fig. 8 | Principal modes of simulated Pacific extratropical response to tropical forcing. a**, Multi-model mean regression pattern of grid-point D(0)JF(1) SST (colouring; °C s.d.⁻¹) and 10-m wind (vectors; m s⁻¹ s.d.⁻¹) anomalies onto the normalized SST expansion coefficient of the first SVD mode (SVD1) in 1900–2099 (see text for details). **b**, Same as **a** but for MAM(1). **c,d**, As in **a,b**, respectively, but for the second SVD mode (SVD2). **e,f**, As in **a,b**, respectively, but for the sum of SVD1 and SVD2. **g**, Relationship of the combined (SVD1 + SVD2)

expansion coefficients between D(0)JF(1) SST (*x* axis) and MAM(1) SLP (*y* axis) in 1900–1999 for the selected models. A regression line is shown together with an MME regression coefficient and 1.0 s.d. uncertainty range estimated from a bootstrap method. **h**, Same as **g** but for 2000–2099. Shown in **a**–**f** are values at which the ensemble mean exceeds 1.0 s.d. from a bootstrap test. Reconstructed evolution of multi-year La Niña captures the simulated evolution.

**a**

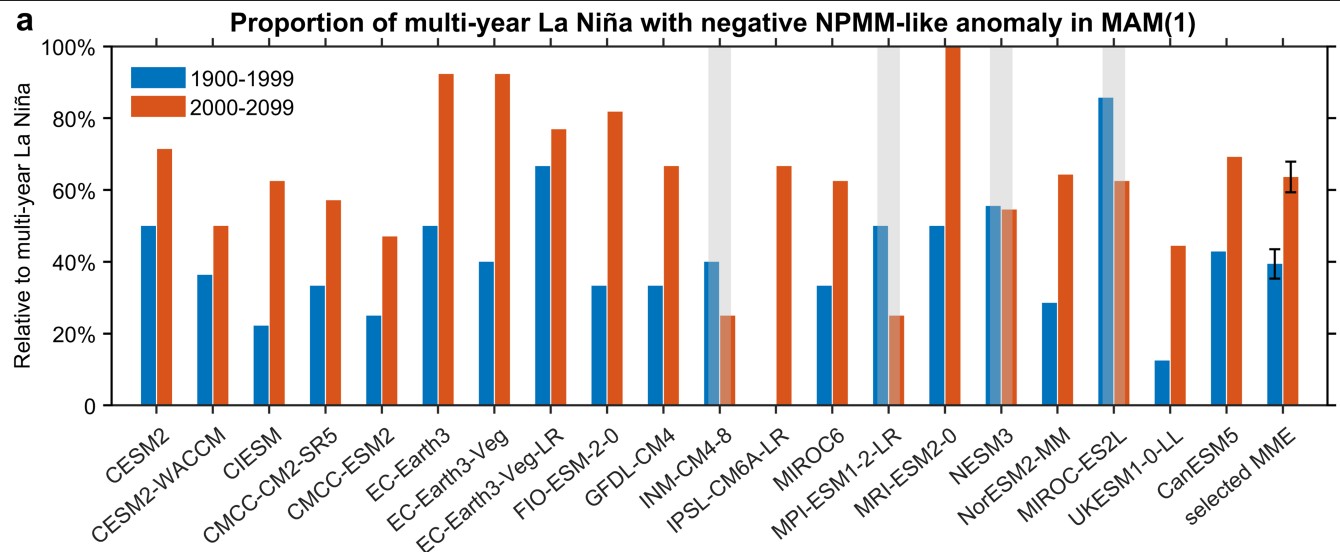

**b**

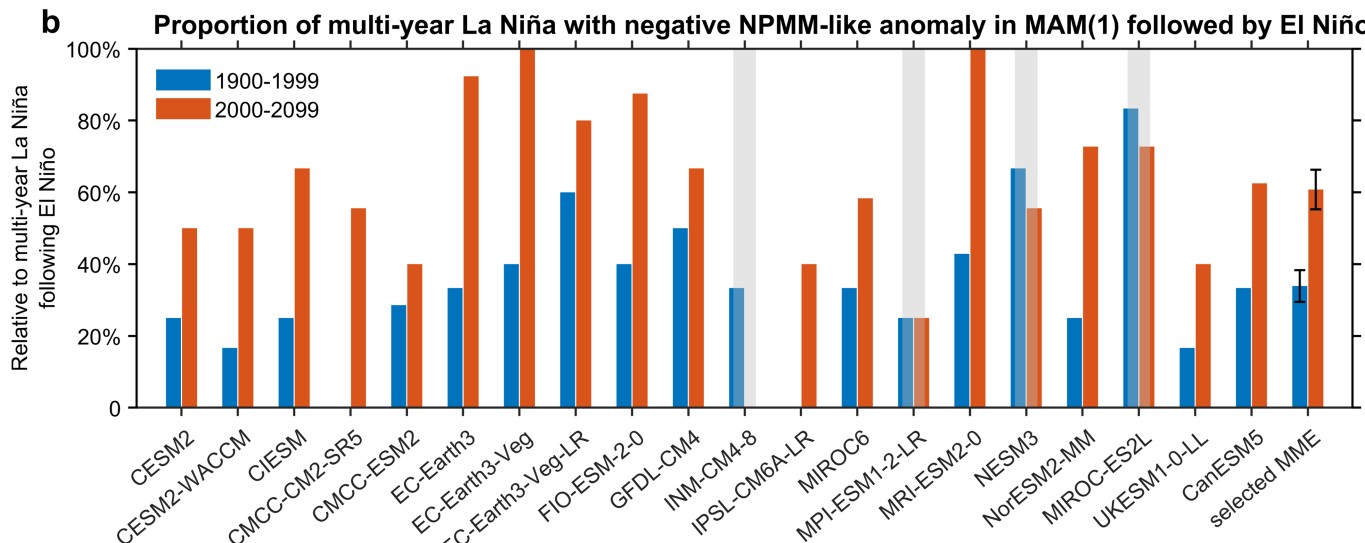

**Extended Data Fig. 9 | Projected increase in co-occurrences of multi-year La Niña with negative NPMM-like events in MAM(1). a**, Comparison of the proportion (in percentage) of multi-year La Niña with negative NPMM-like events in MAM(1) relative to all multi-year La Niña events over 1900–1999 (blue bars) and 2000–2099 (red bars) in the selected models. A negative NPMM-like event is defined as when the NPMM index (normalized SST anomalies in 15° N–25° N, 150° W–120° E) is less than −0.5 s.d. in MAM. Models that simulate a decrease are greyed out. Error bars on the multi-model mean are calculated as 1.0 s.d. of 10,000 inter-realizations of a bootstrap method. **b**, As in **a** but for the proportion (in percentage) of multi-year La Niña with negative NPMM-like events in MAM(1) following an El Niño in the previous boreal winter relative to all multi-year La Niña events that are preceded by an El Niño. Both strong and weak El Niño events are considered here. There is a greater involvement of a negative NPMM in the generation of multi-year La Niña during 2000–2099 than during 1900–1999.

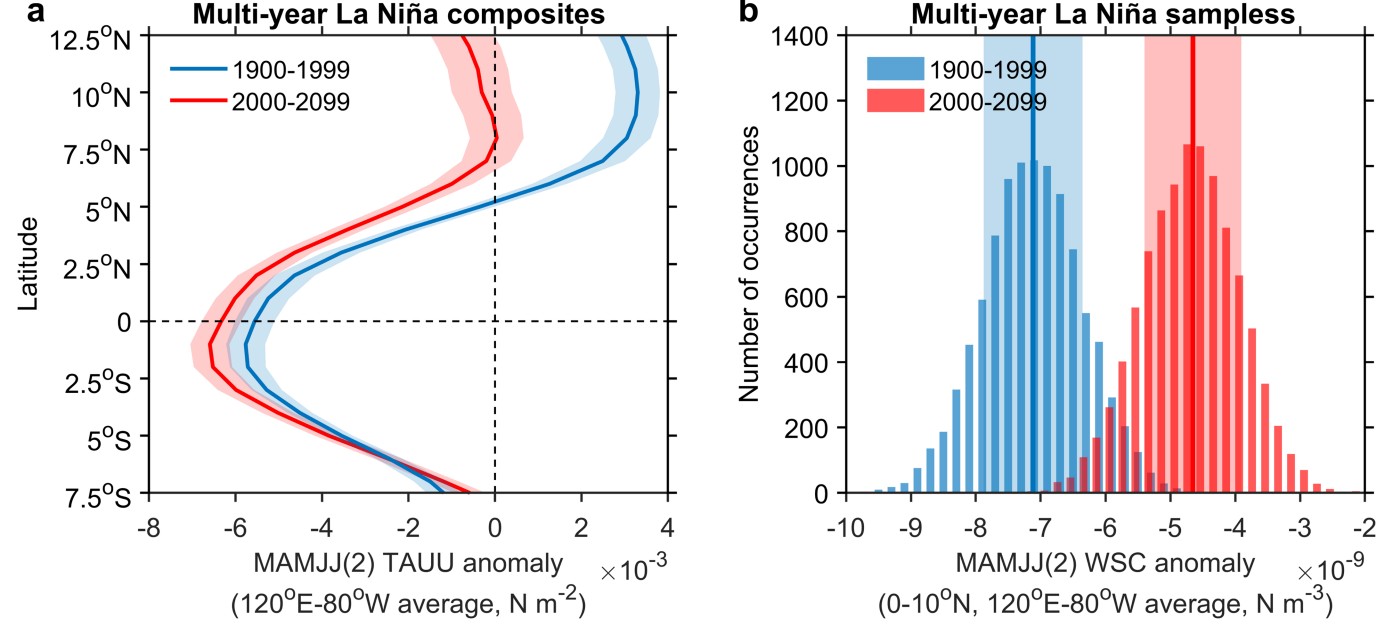

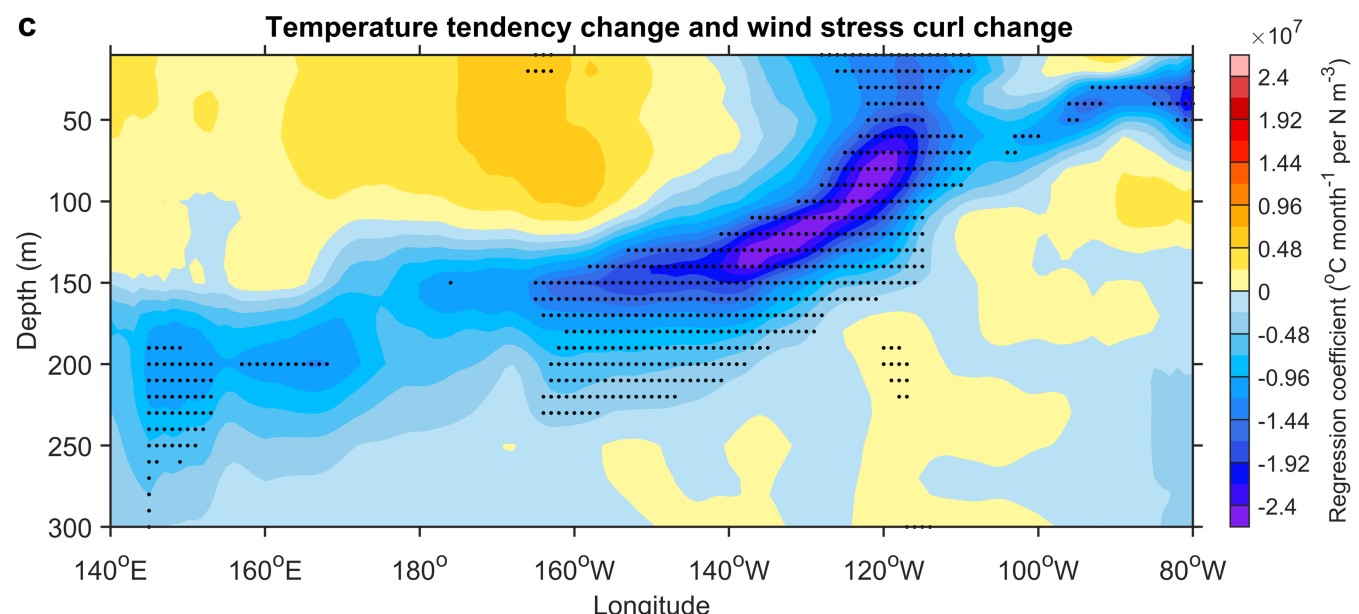

**Extended Data Fig. 10 | Northward-broadened easterly anomalies and associated weakening of upper-ocean heat recharge. a**, Latitudinal distribution of zonal wind stress (TAUU) anomalies (N m$^{-2}$) in the Pacific (120° E–80° W) composited for multi-year La Niña events during MAMJJ(2) in the selected models. **b**, Histograms of 10,000 realizations of a bootstrap method for wind stress curl anomalies (averaged in 0–10° N, 120° E–80° W; N m$^{-3}$) during MAMJJ(2) for all multi-year La Niña samples in 1900–1999 (blue bars) and 2000–2099 (red bars). Solid lines and shading in **a** and **b** indicate multi-model mean and 1.0 s.d. of a total of 10,000 inter-realizations based on a bootstrap method, respectively. **c**, Inter-model regression of changes (2000–2099 minus 1900–1999) in D(1)JFMAMJJAS(2) equatorial (5° S–5° N

average) temperature tendency onto changes in mean intensity of MAMJJ(2) wind stress curl anomalies. Both temperature tendency and wind stress curl anomalies are a composite of the first La Niña of multi-year La Niña events in each model before performing the inter-model regression. Stippling indicates statistical significance above the 90% level based on a two-tailed Student's *t*-test. Changes are scaled by the increase in global mean SST of each model to facilitate inter-model comparison. During 2000–2099, easterly anomalies of the first-year La Niña are meridionally broadened and the associated negative wind stress curl anomalies are weaker, slowing the heat recharge of the equatorial Pacific and providing a colder ocean state conducive to development of the second-year La Niña.