## [Peer Review File · Nature]

Manuscript Title: Increased occurrences of consecutive La Niña events under global warming

Reviewer Comments & Author Rebuttals

Reviewer Reports on the Initial Version:

Referees' comments:

Referee #1 (Remarks to the Author):

Review for "Increased occurrences of multi-year La Niña events under greenhouse warming" by Geng et al.

The manuscript investigates the future projection of multiyear La Nina events using the CMIP6 historical and future simulations. The frequency of multiyear La Nina is projected to increase in the 21st century than in the 20th century, and this increase is more pronounced under higher emission scenarios. Interaction between tropical Pacific subtropical North Pacific gets stronger due to global warming, favoring the development of negative NPMM mode following El Nino and thus the La Nina development and persistence. The paper is well organized and written, and the figures are well presented. The model projection of the more frequent multiyear La Nina events in the 21st century is a new result and critical due to prolonged climate impacts caused by multiyear La Nina events. The mechanism related to NPMM-ENSO interaction or more particularly NPMM-multiyear La Nina interaction has been well identified and studied in previous studies as reviewed in the Introduction and thus is not a newly proposed physical mechanism, but the authors find this process is key to explaining the future changes in multiyear La Nina events in the CMIP6 models. I have some major comments regarding the model fidelity in simulating tropical climate change and the NPMM mechanisms. Please see my specific major and minor comments listed below.

Major comments:

1. Model fidelity in simulating and projecting the mean-state and ENSO dynamics is not well considered and discussed in this manuscript. One uncertainty is the mean-state SST response in the eastern Pacific cold tongue to anthropogenic warming. Almost all current models overestimate the warming trend in the eastern Pacific in the last ~40 years (e.g., Seager et al. 2019; Wills). In observation, the frequency of multiyear La Nina increased under stronger mean-state warming in the western than the eastern tropical Pacific in the last 40 years, which is opposite to the pattern in Fig. 4a. It is unknown if different mechanisms act to influence multiyear La Nina in model and observation or observed La Nina changes are more driven by natural variability than forced changes. The authors may discuss the potential influence of potential model biases on the results presented in this paper, and its implication to explain the observed changes in multiyear La Nina events in the recent decades.

Seager, R. et al. Strengthening tropical Pacific zonal sea surface temperature gradient consistent with rising greenhouse gases. *Nat Clim Change* 9, 517–522 (2019).

Wills, R. C. J., Dong, Y., Proistosescu, C., Armour, K. C., & Battisti, D. S. (2022).

Systematic climate model biases in the large-scale patterns of recent sea-surface temperature and sea-level pressure change. *Geophysical Research Letters*, 49, e2022GL100011. <https://doi.org/10.1029/2022GL100011>

2. Line 166-170: I am not very convinced that the changes in the preceding El Nino precursor are not important to changes in the frequency of multiyear La Nina. From Extended Fig. 3, the

preceding El Nino, on average, gets stronger in 2000-2099 than in 1900-1999. Based on Fig. 2c, there is an increase in the frequency of multiyear La Nina preceded by strong El Nino, although increases in strong El Nino occurrence do not guarantee increases in occurrence of multiyear La Nina. Does it suggest that the changes in the preceding El Nino precursor still play a role? In extended Fig. 7, Why use the amplitude change in strong El Nino events in the x-axis rather than the frequency change in strong El Nino events or the amplitude of all El Nino events? Also, I wonder if most models are able to simulate the relationship between strong El Nino and multiyear La Nina.

3. It remains unclear to me why the authors think the NPMM mode is the key mechanism and how it acts to persist the La Nina events. Extended Data Fig. 10: The authors argue that the negative NPMM in MAM(1) is a response to the preceding El Nino D(0)JF(1) warm anomalies, favoring the transition of El Nino to La Nina. However, why does a negative NPMM still reemerge in MAM(2)? Should we expect to have a positive NPMM in MAM(2) following La Nina in D(0)JF(2)?

Line 185-187: There are significant changes over the tropical Indian and Atlantic Oceans (Fig. 9). Can you discuss in more detail why these Indian and Atlantic changes are not important to the changes in multiyear La Nina?

The subtropical South Pacific is cut off in all figures, but I wonder if there are significant mean-state changes in the South Pacific and contributions from the SPMM to ENSO. The SPMM seasonally intensifies in boreal summer and autumn. Fig. 9c shows significant southeasterly wind anomalies over 0°-20°S in the western Pacific in JJA(1), symmetric to the counterpart in the northwestern Pacific.

Minor comments:

1. Extended Data Fig. 1, SST datasets have large disagreement before the 1950s. Will the identification of multiyear La Nina events based on HadISST remain similar when other SST datasets are used? The authors mentioned they used three different datasets in the Methods but did not show the results.

2. Line 22: A recent paper on the impact of multiyear La Nina events in the Horn of Africa: W. Anderson, B. I. Cook, K. Slinski, K. Schwarzwald, A. McNally, C. Funk, Multi-year La Niña events and multi-season drought in the Horn of Africa. *J Hydrometeorol* (2022), doi:10.1175/jhm-d-22-0043.1.

3. Line 27 and Line 285, 287: 'A faster warming...' I think it might be necessary to indicate if it refers to mean-state warming. Is 'faster' here indicating comparisons among relative regions or models?

4. Line 82-93 & Fig. 1a: The authors select 22 CMIP6 models based on their ability to simulate the asymmetric amplitude between El Nino and La Nina (amplitude skewness), but I think it is more straightforward to select models based on the performance of duration asymmetry as the paper focuses on the duration. For example, Line 100-102: Like Fig. 1a, can you include the statistics of the frequency of multiyear La Nina for individual models in the supplemental?

Referee #2 (Remarks to the Author):

January 2023,

A review of the manuscript (2022-12-19333A) entitled "Increased occurrences of multi-year La

Niña events under greenhouse warming” by Geng et al.

The authors first showed that the multi-year La Niña may occur more frequently under a warmer climate based on CMIP6 climate model simulations. They confirmed that climate models having positive skewness of the Niño-3.4 SST anomalies can simulate single-year and multi-year La Niña reasonably. Then, they showed that the occurrence frequency of multi-year La Niña is increased in the 21st century under various emission scenarios. To explain the reason for multi-year La Niña occurrence changes, they degraded two potential mechanisms and interpreted that a wind-evaporation-SST (WES) feedback change due to global warming results in a higher impact of the North Pacific SST anomalies, resembling Pacific Meridional Mode (PMM), that increase the La Niña longevity by reducing the wind curl (i.e., heat recharge). The Pacific mean-state warming pattern is explained as a key for increasing the multi-year La Niña occurrences.

The presented results about the multi-year La Niña events are interesting to not only climate researchers but also the other fields affected by climate variability. However, I have three major concerns about originality, presentation, and accuracy.

First, a recent previous article by Fan et al. (2022, J Climate <https://doi.org/10.1175/JCLI-D-21-0683.1>), which is not cited in the manuscript, already showed the potential future change in the PMM and its impact on ENSO in association with changes in a WES parameter using CMIP6 climate models. Although the previous work did not focus on the multi-year La Niña events, it is highly relevant to the present manuscript. At least, the authors should discuss the consistency of their results with the previous work and highlight their originality.

Second, the manuscript largely relies on the extended data figures. Even though the first paragraph (abstract) includes the results from a low-emission scenario to compare with those in a high-emission scenario, the main figures only provide results from the high-emission scenario. Because confirming their finding in various emission scenarios is critical for concluding the future change of multi-year La Niña occurrences, key results from the four emission scenarios examined in the present work must appear in the main figures.

Third, I have many detailed questions about their data analysis.

- Some key results are not tested using multiple emission scenario simulations. [Specific comment #1, #9, #12, #19, #28]
- Some results are not tested statistically. [#13, #16, #20, #22, #24]
- The authors indicated that two potential mechanisms could be denied for resulting in the increasing frequency of multi-year La Niña events, but their analysis is not reasonable to conclude so yet. I think the enhanced tropics-extratropics interaction under a warmer climate, which is also recently examined by Fan et al. (2022, J Climate) is reliable and interesting, but the authors should not conclude that the other two mechanisms could be less important. [#13, #14]
- Also, it is not explained why the suggested mechanism does not contribute to increasing the occurrence frequency of “triple-dip” La Niña and multi-year El Niño (Lines 128-130 and 563-567). [#27]
- In addition, the developing phase of La Niña (boreal summer and autumn) is not carefully examined as the provided results are mostly about the seasons of boreal winter and spring. [#15]
- Furthermore, I am not sure if the definition of climatology based on 1900-1999 could be used for 2000-2099 (Lines 514-515). Given that the background seasonality in, for instance, storm track and equatorial Pacific cold tongue, could be altered from the past to the warmer climate, using the past climatology for calculating anomalies in the warmer climate can modulate the ENSO index and thus La Niña frequency and pattern. This issue cannot be solved by removing the long-term trend quadratically if the annual cycle is amplified.
- Selecting climate models based on the skewness of the Niño-3.4 SST anomalies using a single realization for each model remains a large uncertainty as a large ensemble is required for precisely estimating the simulated ENSO skewness. [#7]

- I also realized that the model set needs to be revised. [#25, #26]

More detailed comments follow below, of which relevant numbers were shown at each point of the third concern. Provided that the authors would address my detailed questions about their data analysis and make the figure presentation better, I would like to suggest a major revision if the authors can clearly answer my first concern to highlight the novelty of this study.

Specific comments:

1. Lines 25-26: As the results from a low-emission scenario are provided in the first paragraph, add key results from various low-emission scenarios in the main figures.
2. Lines 65-67: The connection from the WES feedback to "favoring development of a meridionally broad La Nina" is unclear in this sentence. A brief explanation is needed with relevant references.
3. Lines 70-71: In this sentence, the authors indicate that the air-sea coupling is stronger in boreal summer and autumn based on reference 38, Mitchell and Wallace 1992 JCLI [https://doi.org/10.1175/1520-0442\(1992\)005<1140:TACIEC>2.0.CO;2](https://doi.org/10.1175/1520-0442(1992)005<1140:TACIEC>2.0.CO;2). However, this previous work might have discussed the mean state seasonality in equatorial convection and SST (the cold tongue-ITCZ complex). A better explanation needs to be written, or more relevant previous works should be cited here.
4. Line 73: Reference 18 (Okumura et al. 2017 GRL) does not conclude the recent increase of the multi-year La Nina occurrence frequency.
5. Line 77: Reference is required for "a projected increase in strong El Nino". Probably, Cai et al. (2022 and 2014 in Nat Clim Change) are relevant.
6. Line 90: The underestimation of ENSO feedback is not first indicated by reference 14 (Cai et al. 2021). Add more relevant original papers here instead of only citing the review paper.
7. Lines 90-92: A single realization for each model cannot precisely estimate the skewness of ENSO SST (Lee et al. 2021, GRL). A reasonable reason for using a single realization for each model needs to be added or provide relevant supporting information.
8. Lines 99-102: This is partly shown in Fig. 1a but the figure reference is missing. Also, Fig. 1a could show the results of the non-selected models and all models to indicate that the selected models better reproduce the multi-year La Nina occurrences.
9. Line 120-121, Fig. 1a: Even though the authors want to focus on SSP585, the results in the other SSP scenarios should be shown in the main figures at least their ensemble mean with uncertainty, as they are critical as the first paragraph includes them.
10. Lines 144-147, Fig. 1b: Concatenating the piControl and historical simulations is misleading. Why the piControl could be continuously connected to the historical simulation? Since connecting these two simulations may lead to a misinterpretation as if the authors have data prior to the historical era with the external forcing changing with time, the piControl time series should be regarded as an independent one.
11. Lines 153-155 and 266-270: This plateauing in the mid-21st century might be caused by the 60-year forward sliding method. The data after 2040 is overlapped, which may create a plateau.
12. Lines 166-169: This result should be confirmed in the other SSP simulations because the El Nino change is seemingly inconsistent with previous works (e.g., Cai et al. 2018=ref. 43).
13. Lines 180-181: I wonder if the discharge rates shown in Extended Data Fig. 8b have high uncertainty. In panel a (observed relation), strong El Nino events are accompanied by a wide range of the discharge rate, which is much higher than the future changes shown in panel b. Therefore, to conclude as in Lines 180-183, the number of ensemble members for each model needs to be increased to detect the signal. Also, in Extended Data Figure 8b, each bar could have the uncertainty range (e.g., 1 s.d. of the samples for each model simulation).
14. Lines 185-187: This is not concluded in Extended Data Fig. 9 as there are significant changes in the tropical Indian and Atlantic Oceans as well.
15. Lines 192-193: It is not clear if the northward broadened and stronger subtropical northeasterly anomalies in the 21st century continue from MAM(1) to JJA(1) and SON(1) because the authors do not show the results in boreal summer and autumn (these seasons are indeed favorable for dynamic feedback, according to the introduction). For example, Extended Data Figure

9 does not include the results in SON(1).

16. Lines 210-213: Show an example for the regression line and scatter plots of the time series in the extended data, for instance, as this analysis is key but complicated and the potential uncertainty residing in the regression coefficient is not described.

17. Lines 217-220: Why the x-axis and y-axis are scaled by the global SST change (GW) in Fig. 3b? Since the figure wants to indicate the relationship between the tropics-extratropics connection intensity and the multi-year La Nina number, such scaling seems to be not necessary.

18. Line 222: This comparison should be done in the main figure.

19. Lines 226-228: Have the authors confirmed this result in the other SSP scenario simulations? Because this result is key, inter-scenario uncertainty should be considered.

20. Lines 249-250: It is not clear if the majority of the models project the increasing trend of the shortwave and longwave radiation in the subtropical northern Pacific because the Extended Data Fig. 13 does not include information about the inter-model consensus of the mean state changes.

21. Lines 259-260 and 272: Why the Nino3 SST is used instead of the Nino3.4 SST, which is used in the other analysis of this manuscript?

22. Figure 1b: Does the CMIP6 models overestimate the multi-year La Nina amplitude between Dec(1) and Dec(2)? How many models could capture the observed amplitude realistically?

23. Figures 3a and 4d: The observed values in 1900-1999 could be provided.

24. Figure 4d: The inter-model difference of these slopes could be analyzed to show that the future change in this relationship is in consensus among the models.

25. Methods: CanESM5 and CanESM5-CanOE are identical to each other in the historical and SSP simulations (Table 2 of Swart et al. 2019 <https://doi.org/10.5194/gmd-12-4823-2019>). Even though the CanESM5 r1i1p1f1 realization could be analyzed, it includes a serious bug (See Appendix E of Swart et al. 2019: 'Notable in the differences are the "cold" spots in Antarctica, which arise from a mis-specified land fraction in p1, and were resolved in p2'). To avoid the double count of a single model, I would suggest removing the CanESM5 r1i1p1f1 realization. Then, the CanESM5-CanOE r1i1p2f1 results could be replaced with the CanESM5 r1i1p2f1 results as they are identical.

26. Methods: The MCM-UA-1-0 model (The Manabe Climate Model v1.0 - University of Arizona climate model, released in 1991) is not appropriate for examining the air-sea interaction mode as flux adjustments are implemented. <https://www.wdc-climate.de/ui/cmip6?input=CMIP6.CMIP.UA.MCM-UA-1-0> <https://explore.es-doc.org/cmip6/models/ua/mcm-ua-1-0>

27. Lines 563-567: Why the triple La Nina or multi-year El Nino events do not increase in the future climate? The same mechanism for increasing the multi-year La Nina event occurrence frequency would work for these events.

28. Extended Data Figures 7a and 8b: The results in the other emission scenarios should be shown.

29. Extended Data Figure 7c: Why the number of models shown in Extended Data Fig. 7c is much smaller than 22?

Typos etc.:

1. Line 426: b instead of B

2. Line 561: influences instead of Influences

3. Line 603: I am not sure if "Codes are available from the corresponding author on request" is acceptable.

4. Extended Data Figure 9: The MCM-UA-1-0 and CESM2 models are missing in Extended Data Fig. 8 although the legend includes them.

5. Reference 46: IPCC AR6 has been published, so update the reference information.

Referee #3 (Remarks to the Author):

Comments on "Increased occurrences of multi-year La Niña events under greenhouse warming" by Geng et al.

Overall comment:

In this manuscript, the authors analyzed CMIP6 models to quantify and explain the projected increased number of multi-year La Niña under global warming. They stated that larger mean state warming in the subtropical northeastern Pacific could lead to both a faster warming in the eastern Pacific (i.e., more strong El Niños) and a weaker recharge process during La Niña, leading to more numbers of multi-year La Niña under global warming.

The topic is crucial and interesting, especially since there was a three-year La Niña event last year. I appreciate the authors did a lot of effort to analyze those results. And indeed, the authors propose a possible explanation for "parts of" the increased number of multi-year La Niña under greenhouse warming. That said, the scientific story of why multi-year La Niña is projected to increase is not solid and comprehensive enough. I thus cannot recommend the publication of the current manuscript in Nature. My detailed comments are as follows.

Major comments:

1. The citations are not accurate enough

The multi-year La Niña is a hot topic in the ENSO community for more than a decade, and relative research from different perspectives has been made. However, the authors do not provide sufficient citations for their discussions that reflect the community contributions over the past decades. For example, in Lines 20-24 and 43-44, the authors simply provide citations for El Niño or ENSO impacts but not specifically for multi-year La Niña impacts, such as Anderson et al. (2017), Jong et al. (2020), and Lopes et al. (2022), even though the sentences are discussing the multi-year La Niña impacts. Also, for the dynamics of multi-year La Niña in Lines 18-22 and 56-72, the authors focus on the precondition of the first-year La Niña, but not the transition from the first-year La Niña to the second-year La Niña. And other citations that are not accurate enough are in detailed comments.

Anderson, W., Seager, R., Baethgen, W., & Cane, M. (2017). Life cycles of agriculturally relevant ENSO teleconnections in North and South America. *International Journal of Climatology*, 37(8), 3297-3318.

Jong, B. T., Ting, M., Seager, R., & Anderson, W. B. (2020). ENSO teleconnections and impacts on US summertime temperature during a multiyear La Niña life cycle. *Journal of Climate*, 33(14), 6009-6024.

Lopes, A. B., Andreoli, R. V., Souza, R. A., Cerón, W. L., Kayano, M. T., Canchala, T., & de Moraes, D. S. (2022). Multiyear La Niña effects on the precipitation in South America. *International Journal of Climatology*.

2. Lacking discussions on the transition from the first- to the second-year (or more) La Niña

The authors focus on the pre-condition of the first-year La Niña, including the strong El Niño and the larger trade winds (or widen meridional structure of La Niña). However, there is limited discussion on the condition from the first- to second-year La Niña, as well as second to third or third to fourth or more. The pre-condition of the first-year La Niña may lead to a colder ocean that favors the development of the second-year La Niña, but, as the authors have mentioned and shown, the increased number of the multi-year La Niña is larger than the increase of strong El Niño and no relation to the discharge rate. This indicates that there is something missing in the authors' argument, which may come directly from the transition between the first- and second-year La Niña.

Or oppositely, most of the results are tightly related to the projected increase of strong El Niño, such as the increase of svd time series, the negative PMM, and the weakening of the negative wind stress curl. However, does the precondition of La Niña really have a colder ocean state? And can this colder ocean state remain and impact the onset of the second-year La Niña? The authors have limited discussion on the following parts, which is the main component that directly influence the increase of multi-year La Niña.

3. The broadening of La Niña should lead to a weaker discharge, which is not seen in extended figure 9.

A major argument is that the northward trades lead to the broadening of La Niña, as well as the weaker discharge. However, the two proxies of discharge, the SSH and wind stress curl are not matched, one showing no relation with La Niña but the other shows an impact. The authors should better justify this connection either using oceanic variables or other methods that more intuitively indicate the colder first-year La Niña (which is also not shown) is related to the broadening of PMM and also the northward change of trades.

4. Not enough ocean analyses

The authors use SSH and wind stress curl as the proxies for ocean heat change. However, limited discussion on the pre-condition of the first- and second-year La Niña from ocean variables. As their argument is that the ocean has a colder precondition of the first-year La Niña that leads to the multi-year La Niña. The analyses directly related to the precondition should also be provided, especially with more direct oceanic variables, such as upper 300m potential temperature or 20-degree C isothermal depth, or other oceanic variables.

Detailed comments:

1. The title should be narrowed/weakened as the manuscript is mainly about the strong El Niño to two-year La Niña, but not the entire multi-year La Niña.

2. Lines 17-18, the strong El Niño to Multi-year La Niña transition has been noticed, but the 2020-2023 three-year La Niña event is not preceded by a strong El Niño. The authors should not phrase as the multi-year La Niña is always from a strong El Niño. Also, only Wu et al. (2013) within the four cited publications discuss the preceding strong El Niño, which the authors need to revise with more accurate wording and citations.

3. Lines 20-24, citations 10-14 are mostly about El Niño impact, but not multi-year La Niña impacts.

4. Lines 43-44, citations 17, 19, 20, 21, and 22 are not really for multi-year La Niña impacts, even though the sentence is for comparing single to multi-year La Niña impacts.

5. Lines 55, citation 26 is about the onset of two strong El Niño events, but not the recharge process of La Niña.

6. Lines 56-59, citations 28-30 are only for PMM but not directly for the La Niña development, please also cite some recent papers about negative PMM to La Niña.

7. Line 93, the authors should also provide results from the non-chosen simulations and briefly discuss their difference compared to the chosen simulations, regarding their changes in future projection (not all models as in Line 132 but only the excluded). This is because the classification from skewness already excludes the possibility of multi-year La Niña not from strong El Niño.

8. Lines 96-97, the authors have excluded the three-year (or more) multi-year La Niña from classification and emphasized the importance of the precondition from extreme El Niño; however, as shown in Lines 128-130, the triple-year La Niña seems to have a different change, meaning the authors should indicate the limitation of the mechanism from a strong El Niño to two-year La Niña and better quantify them, as described in comment 12.
9. Lines 99-102, what is the increase when including multi-year La Niña for more than two years?
10. Lines 114-116, the increase of two-year La Niña is more than the increase of strong El Niño. Does it mean that how the La Niña event continues is more important than the preceding El Niño? Or is the preceding El Niño not even the main reason?
11. Lines 129-131, where does this disagreement come from? Is this change significant?
12. Lines 157-161, as the authors have excluded the triple-year La Niña, meaning the two-year La Niña can only come from a neutral state or an El Niño state and less events are included. As a result, the "more than 75%" should be smaller. Also, do the 66 events refer to all multi-year events in the models that have an increase of multi-year La Niña?
13. Lines 173-176, when influence from extratropic is weak? But the PMM impacts tropical regions mainly during this period.
14. Lines 184-185, "seems to be the case for 2020-2022 La Niña"? Is there a reference or figure? The citation 10 is Okumura's paper in 2010 that proposes the inter-basin interactions.
15. Lines 185-189, this result seems to contradict the explanation of why the increased number of multi-year La Niña is larger than the increase of strong El Niño. The authors stated that the faster warming in the equatorial eastern and subtropical northeastern Pacific leads to more strong El Niños and meridionally broader first-year La Niñas; however, in the end, these strong El Niños and broader first-year La Niñas should lead to a colder ocean state to onset the second La Niña, meaning the discharge process should also have relation with the increased number of the multi-year La Niña, i.e., the mean discharge rate should be proportional/related to the increased ratio of the broader first-year La Niña after all strong El Niños.
16. Lines 195-198, the SVD tries to capture the relation between ENSO peak phase (DJF SST in the tropical Pacific) and the following atmospheric responses (MAM SLP in the north Pacific). However, this cannot infer to the influence on multi-year La Niña
17. Lines 206-207, "most of"? is there a figure showing this? Extended Data Fig.1a do not support this statement, as well as when we consider the model La Niña. If it is not 'most of', what is the ratio of multi-year La Niña that can be explained by this SVD?
18. Lines 216-217, more strong El Niño leads to an increase of the north subtropical atmospheric responses (an increase of svd time-series), but cannot directly connect to the precondition of the first-year La Niña, or even the onset of the second-year La Niña.
19. Lines 226-230, this 77.2% does not come from the review paper from Amaya (2019). Where does it from?
20. Lines 232-235, why is there a weakening of negative wind stress curl but not the discharge rate? Are they tightly related? Is there really a colder precondition for the first-year La Niña?
21. Lines 238-239, "re-intensify"? where is this from? Other papers or result from any figure?

22. Lines 236-241, how does this ocean-atmosphere coupling change impact the transition from the first- to the second-year La Niña?
23. Lines 249-252, how does the broadening of first-year La Niña (at roughly 20N 140W) relate to the northward extension of PMM (at roughly 20N 180E)?
24. Lines 256-258, the correlation in 4c seems to be dominated by the outliers (One in the very top-right and also the bottom-left). How does the correlation changes when excluding those outliers?
25. Lines 267-270, does it mean other processes may contribute to the increase of multi-year La Niña but not only the increase of WES feedback?
26. Lines 271-279, the Strengthened extratropical response to tropical forcing excluding strong El Niño simply means El Niño, in general, can induce the circulation change in the northern extratropical Pacific, but not negative PMM or even the possible following La Niña. Also, the correlation should be high "mathematically" in extended figure 15b since the svd is calculated from the SST in the equatorial Pacific, and the first two principal components have a strong signature in the equatorial eastern Pacific.
27. Lines 281-292, how the northward extension of trade winds can lead to a weaker discharge of El Niño, then a colder precondition for the first-year La Niña, and then the onset of the second-year La Niña is not clearly justified, even though the authors show the weakening of negative wind stress curl.

Reviewed by Shih-Wei Fang

Author Rebuttals to Initial Comments:

Response to Referee #1

The manuscript investigates the future projection of multiyear La Nina events using the CMIP6 historical and future simulations. The frequency of multiyear La Nina is projected to increase in the 21st century than in the 20th century, and this increase is more pronounced under higher emission scenarios. Interaction between tropical Pacific subtropical North Pacific gets stronger due to global warming, favoring the development of negative NPMM mode following El Nino and thus the La Nina development and persistence. The paper is well organized and written, and the figures are well presented. The model projection of the more frequent multiyear La Nina events in the 21st century is a new result and critical due to prolonged climate impacts caused by multiyear La Nina events. The mechanism related to NPMM-ENSO interaction or more particularly NPMM-multiyear La Nina interaction has been well identified and studied in previous studies as reviewed in the Introduction and thus is not a newly proposed physical mechanism, but the authors find this process is key to explaining the future changes in multiyear La Nina events in the CMIP6 models. I have some major comments regarding the model fidelity in simulating tropical climate change and the NPMM mechanisms. Please see my specific major and minor comments listed below.

Thank you for your positive and helpful comments!

Major comments:

1. Model fidelity in simulating and projecting the mean-state and ENSO dynamics is not well considered and discussed in this manuscript. One uncertainty is the mean-state SST response in the eastern Pacific cold tongue to anthropogenic warming. Almost all current models overestimate the warming trend in the eastern Pacific in the last ~40 years (e.g., Seager et al. 2019; Wills). In observation, the frequency of multiyear La Nina increased under stronger mean-state warming in the western than the eastern tropical Pacific in the last 40 years, which is opposite to the pattern in Fig. 4a. It is unknown if different mechanisms act to influence multiyear La Nina in model and observation or observed La Nina changes are more driven by natural variability than forced changes. The authors may discuss the potential influence of potential model biases on the results presented in this paper, and its implication to explain the observed changes in multiyear La Nina events in the recent decades.

Seager, R. et al. Strengthening tropical Pacific zonal sea surface temperature gradient consistent with rising greenhouse gases. *Nat Clim Change* 9, 517–522 (2019).

Wills, R. C. J., Dong, Y., Proistosescu, C., Armour, K. C., & Battisti, D. S. (2022).

Systematic climate model biases in the large-scale patterns of recent sea-surface temperature and sea-level pressure change. *Geophysical Research Letters*, 49,e2022GL100011. <https://doi.org/10.1029/2022GL100011>

Great suggestion. We have assessed the potential impact from a common SST bias in the equatorial Pacific. Although most models still simulate a too-cold climatological cold tongue in the equatorial Pacific, there is **no significant** ($p=0.34$) inter-model relationship between the intensity of the cold tongue bias and projected change in multi-year La Niña frequency under greenhouse warming (**Fig. R1-1**), suggesting that the mean SST bias does not systematically affect our results.

Regarding the observed changes in multi-year La Niña frequency and mean-state equatorial Pacific warming, the current observational record is subject to much uncertainty in the first half of the 20th century. Therefore, the observed change itself may not be relied upon without any additional corroborating evidence. Although models tend to underestimate the observed SST warming trend between the western and eastern equatorial Pacific (**Fig. R1-2a**; please also note large inter-product differences in reanalysis datasets), there are models that are able to simulate both an increase in multi-year La Niña frequency and a faster warming in the western than eastern equatorial Pacific in the last 40 years (1981-2020) than 40 years before (1941-1980) (**Fig. R1-2a, the upper right quadrant**), suggesting an influence from natural variability on recent multi-year La Niña change.

Further, a sensitivity test indicates that, even if data quality were not an issue, detecting observed change may depend on the time period in which the frequency of multi-year La Niña is diagnosed. For example, when comparing 1945-1979 with 1980-2014, there is **no change** in the frequency of multi-year La Niña events in observations (black symbols, **Fig. R1-2b**), though there is still a strengthened west-minus-east SST gradient in the equatorial Pacific. Therefore, our analyses above indicate that the observed multi-year La Niña change is subject to uncertainty and may be influenced by natural variability.

In the revised version, we have added a new section ‘Model bias and recent multi-year La Niña change’ in Methods to discuss about model bias and observed multi-year La Niña changes (Lines 136-139).

Fig. R1-1 | Influence of the cold tongue bias on projected change in multi-year La Niña frequency under greenhouse warming. Inter-model relationship between the cold tongue bias ($^{\circ}\text{C}$), measured by model-observation difference in climatological SST over the equatorial Pacific region (160°E - 100°W , 2°S - 2°N) in 1900-1999, and change (2000-2099 minus 1900-1999, scaled by per $^{\circ}\text{C}$ of global warming) in multi-year La Niña frequency in the selected models. Linear fit (solid black line) is displayed together with correlation coefficient R and corresponding p value.

Fig. R1-2 | Multi-year La Niña frequency and mean equatorial Pacific zonal SST gradient during recent decades. **a**, Relationship between change (1981-2020 minus 1941-1980) in multi-year La Niña numbers and trend in mean equatorial Pacific zonal SST gradient ($^{\circ}\text{C}$ per decade) during 1941-2020. Shown are results from the selected CMIP6 models and three observational reanalysis datasets. The zonal SST gradient is defined the SST difference between the western Pacific (150°E - 180° , 5°N - 5°S) and the central-eastern Pacific (150°W - 120°W , 5°N - 5°S), so that a positive trend in the gradient represents a faster warming in the western than eastern equatorial Pacific. **b**, As in **a**, but for the period of 1945-2014.

2. Line 166-170: I am not very convinced that the changes in the preceding El Nino precursor are not important to changes in the frequency of multiyear La Nina. From Extended Fig. 3, the preceding El Nino, on average, gets stronger in 2000-2099 than in 1900-1999. Based on Fig. 2c, there is an increase in the frequency of multiyear La Nina preceded by strong El Nino, although increases in strong El Nino occurrence do not guarantee increases in occurrence of multiyear La Nina. Does it suggest that the changes in the preceding El Nino precursor still play a role?

Yes. The preceding El Nino precursor still plays an important role under greenhouse warming.

The increase in multi-year La Niña frequency is **despite** that there is no inter-model consensus on the amplitude change in strong El Niño, meaning that an El Niño becomes **more efficient** in triggering an ensuing multi-year La Niña, and we find that it does so through an enhanced tropical-subtropical interaction, which is our main point.

In extended Fig. 7, Why use the amplitude change in strong El Niño events in the x-axis rather than the frequency change in strong El Niño events or the amplitude of all El Niño events?

We meant to gauge the direct influence associated with the intensity change of strong El Niño since a substantial portion (more than 83%, 45 out of 54 events in aggregation) of the increase in multi-year La Niña frequency stems from an increased percentage of multi-year La Niña preceded by a strong El Niño (Line165).

Obviously, there should be a strong link between frequency change in multi-year La Niña and frequency change in strong El Niño (an inter-model correlation is $r=0.74$, $p<0.01$). However, the frequency of multi-year La Niña increases **disproportionally** with that of strong El Niño (Fig. 2c) and there is **no significant** inter-model relationship between the change in multi-year La Niña frequency with the change in amplitude of all preceding El Niño events (**Fig. R1-3**). The result reinforces the point that the change in multi-year La Niña is due to a more efficient preceding El Niño but not necessarily a stronger El Niño.

Fig. R1-3 | No impact from changing amplitude of preceding El Niño. Inter-model relationship between changes (2000-2099 minus 1900-1999) in multi-year La Niña numbers and in the mean amplitude of all El Niño events that precede multi-year La Niña, both scaled by the increase in global mean SST increase (i.e., global warming, GW) in each model. Linear fit (solid black line) is displayed together with correlation coefficient R and corresponding p value.

Also, I wonder if most models are able to simulate the relationship between strong El Niño and multiyear La Niña.

Yes, 90% (18 out of 20) of models are able to simulate a multi-year La Niña following a strong El Niño in 1900-1999, with an MME percentage of $32.4 \pm 4.3\%$ comparable to the observed 34.15% (Fig. R1-4, blue bars). During 2000-2099, all models can simulate the relationship (Fig. R1-4, red bars) and 70% (14 out of 20) of models produce an increase in the proportion of a multi-year La Niña preceded by a strong El Niño relative to all multi-year La Niña from 1900-1999 to 2000-2099 (Fig. R1-4, compare blue and red bars).

Fig. R1-4 | Proportion of multi-year La Niña preceded by strong El Niño. Comparison of proportion (in percentage) of a multi-year La Niña preceded by a strong El Niño relative to all multi-year La Niña over 1900-1999 (blue bars) and 2000-2099 (red bars) in the selected models. Models that simulate a decrease are greyed out. Error bars on the multi-model mean are calculated as 1.0 standard deviation of 10,000 inter-realizations of a Bootstrap method. The horizontal dashed line indicates observation.

3. It remains unclear to me why the authors think the NPMM mode is the key mechanism and how it acts to persist the La Nina events. Extended Data Fig. 10: The authors argue that the negative NPMM in MAM(1) is a response to the preceding El Nino D(0)JF(1) warm anomalies, favoring the transition of El Nino to La Nina. However, why does a negative NPMM still remerge in MAM(2)? Should we expect to have a positive NPMM in MAM(2) following La Nina in D(0)JF(2)?

The direct trigger of a negative NPMM is enhanced trade winds over the subtropical northeastern Pacific, which can be induced by either a strong El Niño or a La Niña but via different processes. During the peak and decaying phase (boreal winter and spring) of a strong El Niño, the northern edge of the anomalous cyclone associated with a Gill-type atmospheric response over the **equatorial eastern** Pacific enhances the trade winds, initiating a negative NPMM (i.e., NPMM in MAM1) and favoring the transition from El Niño to La Niña. During the peak and decaying phase of a La Niña, the cold SST anomalies in the **equatorial western and central** Pacific effectively weaken the permanent deep convection, inducing an anomalous anticyclone over the subtropical northeastern Pacific through either Rossby wave trains (Fang

& Yu 2020) or Aleutian Low response (Stuecker 2018). The southern edge of the anomalous anticyclone also enhances the trade winds and initiates a negative NPMM (i.e., NPMM in MAM2), favoring the persistence of La Niña.

Besides, the negative NPMM in MAM(2) is a possible effect of the meridionally broad first-year La Niña which decays slowly and leaves a residual negative NPMM-like cold anomaly in the subtropical northeastern Pacific.

References

Stuecker, M. F. Revisiting the Pacific Meridional Mode. *Sci. Rep.*, **8**, 3216 (2018).

Fang, S. W. & Yu, J. Y. A control of ENSO transition complexity by tropical Pacific mean SSTs through tropical-subtropical interaction. *Geophys. Res. Lett.* **47**, e2020GL087933 (2020).

Line 185–187: There are significant changes over the tropical Indian and Atlantic Oceans (Fig. 9). Can you discuss in more detail why these Indian and Atlantic changes are not important to the changes in multiyear La Nina?

We should have made this clearer. As shown in **Fig. R1-5** below, there is no inter-model relationship between changes in multi-year La Niña frequency and changes in the mean amplitude of modes of climate variability in the subtropical South Pacific, tropical Indian and Atlantic oceans during their respective peaking seasons, suggesting no systematic changes in the impact from these climate modes on the change of multi-year La Niña frequency.

We have included **Fig. R1-5** as new Extended Data Fig. 7 (in replacement of the original Extended Data Fig. 9) and added more detail on this regard in the revision (Lines 190-196).

Fig. R1-5 (new Extended Data 7) | Negligible impact from changing modes of climate variability in the subtropical South Pacific, tropical Indian and Atlantic oceans. Inter-model relationship between changes (2000-2099 minus 1900-1999) in multi-year La Niña numbers and in the mean amplitude of (a) the South Pacific Meridional mode (SPMM) in FMA(1), (b) the Indian Ocean Basin Mode (IOBM) in MAM(1), (c) the North Tropical Atlantic (NTA) SST anomalies in MAM(1), and (d) the Atlantic Niño/Niña (measured by ATL3 index) in JJA(1). Linear fits (solid black line) are displayed together with correlation coefficient R and corresponding p value. Here the SPMM index is defined as normalized SST anomalies over the subtropical southeastern Pacific (15°S – 25°S and 110°W – 90°W), the IOBM is defined as the first EOF mode of SST anomalies in the tropical Indian Ocean (20°S – 20°N and 40°E – 110°E), the NTA index is taken as normalized SST anomalies in (10°N – 20°N and 60°W – 20°W), and the ATL3 index is normalized SST anomalies averaged over the equatorial Atlantic cold tongue region (3°S – 3°N and 20°W – 0°E).

The subtropical South Pacific is cut off in all figures, but I wonder if there are significant mean-state changes in the South Pacific and contributions from the SPMM to ENSO. The SPMM

seasonally intensifies in boreal summer and autumn. Fig. 9c shows significant southeasterly wind anomalies over 0°-20°S in the western Pacific in JJA(1), symmetric to the counterpart in the northwestern Pacific.

There are significant mean-state changes in the South Pacific, with a local warming minimum in the subtropical southeastern Pacific accompanied by a trend of southeasterlies (Fig. R1-6).

However, there is no significant inter-model relationship between changes of the SPMM amplitude and changes of multi-year La Niña numbers (Fig. R1-5a). Further, there is no inter-model consensus on changes in the occurrence ratio of multi-year La Niña events with a negative SPMM event in its peaking season FMA(1) relative to all multi-year La Niña events (Fig. R1-7). Using JJA(1) yields an essentially similar result. These indicate that the impact from changing SPMM on multi-year La Niña frequency is negligible.

Fig. R1-6 | Mean state changes. Multi-model mean changes of grid-point SST (shading) and 10 m wind (vectors) between 1900-1999 and 2000-2099, scaled by the increase in global mean SST (i.e., global warming, GW) in each of the selected models. Only the mean that exceeds 1.0 s.d. from a bootstrap method is shown.

Fig. R1-7 | Role of SPMM. Comparison of proportion (in percentage) of multi-year La Niña with a negative SPMM events in its peaking season FMA(1) relative to all multi-year La Niña events, over 1900-1999 (blue bars) and 2000-2099 (red bars) in the selected models. Models that simulate a decrease are greyed out. Error bars on the multi-model mean are calculated as 1.0 s.d. of 10,000 inter-realizations of a Bootstrap method.

Minor comments:

1. Extended Data Fig. 1, SST datasets have large disagreement before the 1950s. Will the identification of multiyear La Nina events based on HadISST remain similar when other SST datasets are used? The authors mentioned they used three different datasets in the Methods but did not show the results.

Yes, the identification of multi-year La Niña events remains the same in other SST datasets (**Fig. R1-8**). We have updated Extended Data Fig. 1, now using the average from multiple reanalysis datasets.

Fig. R1-8 | Identification of multi-year La Niña events in two reanalysis datasets. As in Extended Fig. 1a. but for results in (a) ERSSTv5 and (b) COBE-SST2.

2. Line 22: A recent paper on the impact of multiyear La Nina events in the Horn of Africa:

W. Anderson, B. I. Cook, K. Slinski, K. Schwarzwald, A. McNally, C. Funk, Multi-year La Niña events and multi-season drought in the Horn of Africa. J Hydrometeorol (2022), doi:10.1175/jhm-d-22-0043.1.

We have added this reference.

3. Line 27 and Line 285, 287: ‘A faster warming...’ I think it might be necessary to indicate if it refers to mean-state warming. Is ‘faster’ here indicating comparisons among relative regions or models?

It refers to the mean-state that warms at a faster rate than adjacent regions. We have made it clear throughout the revision.

4. Line 82-93 & Fig. 1a: The authors select 22 CMIP6 models based on their ability to simulate the asymmetric amplitude between El Nino and La Nina (amplitude skewness), but I think it is more straightforward to select models based on the performance of duration asymmetry as the paper focuses on the duration. For example, Line 100-102: Like Fig. 1a, can you include the statistics of the frequency of multiyear La Nina for individual models in the supplemental?

The asymmetry in the duration of warm and cold phases of ENSO is also reflected in a positive ENSO SST skewness when the SST climatology is defined as a time mean state.

We have included statistics of the frequency of multi-year La Niña for each individual model in Supplemental Fig. 2.

We thank you again for your helpful and thorough comments.

Response Referee #2

The authors first showed that the multi-year La Niña may occur more frequently under a warmer climate based on CMIP6 climate model simulations. They confirmed that climate models having positive skewness of the Niño-3.4 SST anomalies can simulate single-year and multi-year La Niña reasonably. Then, they showed that the occurrence frequency of multi-year La Niña is increased in the 21st century under various emission scenarios. To explain the reason for multi-year La Niña occurrence changes, they degraded two potential mechanisms and interpreted that a wind-evaporation-SST (WES) feedback change due to global warming results in a higher impact of the North Pacific SST anomalies, resembling Pacific Meridional Mode (PMM), that increase the La Niña longevity by reducing the wind curl (i.e., heat recharge). The Pacific mean-state warming pattern is explained as a key for increasing the multi-year La Niña occurrences. The presented results about the multi-year La Niña events are interesting to not only climate researchers but also the other fields affected by climate variability. However, I have three major concerns about originality, presentation, and accuracy.

Thank you for your positive and helpful comments!

First, a recent previous article by Fan et al. (2022, J Climate <https://doi.org/10.1175/JCLI-D-21-0683.1>), which is not cited in the manuscript, already showed the potential future change in the PMM and its impact on ENSO in association with changes in a WES parameter using CMIP6 climate models. Although the previous work did not focus on the multi-year La Niña events, it is highly relevant to the present manuscript. At least, the authors should discuss the consistency of their results with the previous work and highlight their originality.

Thanks for the suggestion. Our results are consistent with Fan et al. (2022) in that we both identify a strengthened NPMM impact on ENSO under greenhouse warming.

Our study is novel because (1) we found for the first time (at least to our knowledge) an increase in the frequency of multi-year La Niña under greenhouse warming (as also acknowledged by the reviewer), which is new and important given the devastating climatic impacts caused by multi-year La Niña events; (2) we identified the mechanism, particularly an intensified **two-way** interaction between the tropics and subtropics that plays a key role in the multi-year persistence of La Niña under greenhouse warming, which is also a clearly novel aspect that no previous studies have discovered. Fan et al. (2022) focused only on the one-way subtropical-to-tropical pathway.

In the revision, we have discussed the consistency of our results with Fan et al. (2022) and highlighted our novelty, citing the great paper you provided. Please see Lines 301-303.

Second, the manuscript largely relies on the extended data figures. Even though the first paragraph (abstract) includes the results from a low-emission scenario to compare with those in a high-emission scenario, the main figures only provide results from the high-emission scenario. Because confirming their finding in various emission scenarios is critical for concluding the future change of multi-year La Nina occurrences, key results from the four emission scenarios examined in the present work must appear in the main figures.

Great suggestion. We have now included key results from other low emissions scenarios in the main figures.

Third, I have many detailed questions about their data analysis.

- Some key results are not tested using multiple emission scenario simulations. [Specific comment #1, #9, #12, #19, #28]

We have tested the key results in other emission scenario simulations. Please see our response to your specific comments [#1, #9, #12, #19, #28].

- Some results are not tested statistically. [#13, #16, #20, #22, #24]

We have tested the statistical significance of our results. Please see our response to your specific comments [#13, #16, #20, #22, #24].

- The authors indicated that two potential mechanisms could be denied for resulting in the increasing frequency of multi-year La Nina events, but their analysis is not reasonable to conclude so yet. I think the enhanced tropics-extratropics interaction under a warmer climate, which is also recently examined by Fan et al. (2022, J Climate) is reliable and interesting, but the authors should not conclude that the other two mechanisms could be less important. [#13, #14]

We should have made this clearer. We meant there is no significant change and impact of the amplitude of preceding strong El Niño and other modes of climate variability in the tropical Indian and Atlantic oceans. Our point is that riding on warming-induced mean-state changes an El Niño is generally more efficient in inducing a northward broadened negative NPMM-like easterly wind response, which slows the heat recharge of the equatorial Pacific during the decaying phase of first-year La Niña (because of a less negative wind stress curl), facilitating persistence of the cold anomalies of first year La Niña into the second year.

We have modified the text and updated figures in the revision. Please see our response to your specific comments [#13, #14].

- Also, it is not explained why the suggested mechanism does not contribute to increasing the occurrence frequency of “triple-dip” La Niña and multi-year El Niño (Lines 128-130 and 563-567). [#27]

We have provided explanations in the revision. Please see our response to your specific comment [#27].

- In addition, the developing phase of La Niña (boreal summer and autumn) is not carefully examined as the provided results are mostly about the seasons of boreal winter and spring. [#15]

We have provided results during these seasons in the revision. Please see our response to your specific comment [#15].

- Furthermore, I am not sure if the definition of climatology based on 1900-1999 could be used for 2000-2099 (Lines 514-515). Given that the background seasonality in, for instance, storm track and equatorial Pacific cold tongue, could be altered from the past to the warmer climate, using the past climatology for calculating anomalies in the warmer climate can modulate the ENSO index and thus La Nina frequency and pattern. This issue cannot be solved by removing the long-term trend quadratically if the annual cycle is amplified.

We have tested and found that using different climatologies to compute ENSO SST anomalies does not alter our results (**Fig. R2-1**). We have added information about this sensitivity in Methods (Lines 541-543). Using historical climatology to calculate anomalies is an approach commonly adopted by the community for studying projected ENSO changes (e.g., Cai et al. 2018; Ding et al. 2022).

References

- Cai, W. et al, Increased variability of eastern Pacific El Niño under greenhouse warming. *Nature*. **564**, 201-206 (2018).
- Ding, R. et al, Multi-year El Niño events tied to the North Pacific Oscillation. *Nature communications*, **13** (1), 3871 (2022).

Fig. R2-1 | Sensitivity of identification of multi-year La Niña events to climatology used for computing SST anomalies. Comparison of multi-year La Niña numbers (events per 100 years) over 1900-1999 (blue bars) and 2000-2099 (red bars) in the selected models under SSP585. Here the SST anomalies are calculated separately for 1900-1999 and 2000-2099, with reference to the monthly climatology of 1900-1999 and 2000-2099, respectively, and detrended. Models that simulate a decrease are greyed out. Error bars on the multi-model mean are calculated as 1.0 standard deviation of 10,000 inter-realizations of a Bootstrap method. The horizontal dashed line indicates observation.

- Selecting climate models based on the skewness of the Niño-3.4 SST anomalies using a single realization for each model remains a large uncertainty as a large ensemble is required for precisely estimating the simulated ENSO skewness. [#7]

Since not all models have the same number of ensemble experiments, using one experiment each model is to give each model the same weight thereby facilitating inter-model comparison. This is also an approach commonly used by the community (e.g., Fan et al. 2022).

We have provided a reason for choosing this “one experiment each model’ approach in the revision. Please also see our response to your specific comment [#7].

- I also realized that the model set needs to be revised. [#25, #26]

We have revised model sets following your suggestions [#25, #26].

More detailed comments follow below, of which relevant numbers were shown at each point of the third concern. Provided that the authors would address my detailed questions about their data analysis and make the figure presentation better, I would like to suggest a major revision if the authors can clearly answer my first concern to highlight the novelty of this study.

Thanks again for the detailed and helpful comments. We have now clearly highlighted the novelty of our study and revised the paper following each of your suggestion. Please see our point-to-point response below to your respective comments.

Specific comments:

1. Lines 25-26: As the results from a low-emission scenario are provided in the first paragraph, add key results from various low-emission scenarios in the main figures.

Done. We have added key results from low-emission scenarios in main Fig.2a and updated others in the supplementary.

2. Lines 65-67: The connection from the WES feedback to "favoring development of a meridionally broad La Nina" is unclear in this sentence. A brief explanation is needed with relevant references.

Done. We have added a brief explanation with a reference study. Please see Lines 65-68.

3. Lines 70-71: In this sentence, the authors indicate that the air-sea coupling is stronger in boreal summer and autumn based on reference 38, Mitchell and Wallace 1992 JCLI [https://doi.org/10.1175/1520-0442\(1992\)005<1140:TACIEC>2.0.CO;2](https://doi.org/10.1175/1520-0442(1992)005<1140:TACIEC>2.0.CO;2). However, this previous work might have discussed the mean state seasonality in equatorial convection and SST (the cold tongue-ITCZ complex). A better explanation needs to be written, or more relevant previous works should be cited here.

Done. We have added a brief explanation. Please see Lines 71-73.

4. Line 73: Reference 18 (Okumura et al. 2017 GRL) does not conclude the recent increase of the multi-year La Nina occurrence frequency.

Done. We have deleted the reference revised the text.

5. Line 77: Reference is required for "a projected increase in strong El Nino". Probably, Cai et al. (2022 and 2014 in Nat Clim Change) are relevant.

Done. We have added references.

6. Line 90: The underestimation of ENSO feedback is not first indicated by reference 14 (Cai et al. 2021). Add more relevant original papers here instead of only citing the review paper.

Done. References have been updated.

7. Lines 90-92: A single realization for each model cannot precisely estimate the skewness of ENSO SST (Lee et al. 2021, GRL). A reasonable reason for using a single realization for each model needs to be added or provide relevant supporting information.

Using one experiment only from each model avoids dominance by models with which many experiments are carried out such that each model is represented equally in the assessment of inter-model consensus and the ensemble mean change.

In the revision, we have provided a reason for choosing this 'model democracy' approach in the 'Observational and CMIP6 data' section in Methods (Lines 549-553).

8. Lines 99-102: This is partly shown in Fig. 1a but the figure reference is missing. Also, Fig. 1a could show the results of the non-selected models and all models to indicate that the selected models better reproduce the multi-year La Nina occurrences.

Thanks for the suggestion. We have revised the text and updated Fig. 1a.

9. Line 120-121, Fig. 1a: Even though the authors want to focus on SSP585, the results in the other SSP scenarios should be shown in the main figures at least their ensemble mean with uncertainty, as they are critical as the first paragraph includes them.

Done. We have added MME results and associated uncertainty range from all SSP scenarios in Fig. 2a.

10. Lines 144-147, Fig. 1b: Concatenating the piControl and historical simulations is misleading. Why the piControl could be continuously connected to the historical simulation? Since connecting these two simulations may lead to a misinterpretation as if the authors have data prior to the historical era with the external forcing changing with time, the piControl time series should be regarded as an independent one.

The piControl is used to gauge the range of natural variations of multi-year La Niña under a constant (1850) level of external forcing. We did treat the piControl time series as an independent one and have made it clear in the Methods of the revision. Please see Lines 545-548.

11. Lines 153-155 and 266-270: This plateauing in the mid-21st century might be caused by the 60-year forward sliding method. The data after 2040 is overlapped, which may create a plateau.

Year number on the X-axis in Fig.2b denotes the end year of the running window, therefore there is no overlap in data after 2040.

12. Lines 166-169: This result should be confirmed in the other SSP simulations because the El Niño change is seemingly inconsistent with previous works (e.g., Cai et al. 2018=ref. 43).

We found no significant change in the mean amplitude of strong El Niño events. But strong El Niño events do occur more frequently and the increased frequency will result in an increased ENSO SST variability which measures the amplitude of all El Niño events. Therefore, there is no contradiction between our results and Cai et al. (2018).

We have confirmed the results in other SSP simulations (Supplementary Fig. 5).

13. Lines 180-181: I wonder if the discharge rates shown in Extended Data Fig. 8b have high uncertainty. In panel a (observed relation), strong El Niño events are accompanied by a wide range of the discharge rate, which is much higher than the future changes shown in panel b. Therefore, to conclude as in Lines 180-183, the number of ensemble members for each model needs to be increased to detect the signal. Also, in Extended Data Figure 8b, each bar could have the uncertainty range (e.g., 1 s.d of the samples for each model simulation).

Done. We have updated Extended Data Fig. 6 (original Extended Data Fig. 8) with uncertainty estimate.

14. Lines 185-187: This is not concluded in Extended Data Fig. 9 as there are significant changes in the tropical Indian and Atlantic Oceans as well.

We should have made this clearer. As shown in **Fig. R2-2** below, there is no inter-model relationship between changes in multi-year La Niña frequency and changes in the amplitude of climate variability modes in the subtropical South Pacific, tropical Indian and Atlantic oceans during their respective peaking seasons, suggesting no significant changes in the impacts from these climate modes on the change of multi-year La Niña.

We have included **Fig. R1-5** as new Extended Data Fig. 7 (in replacement of the original Extended Data Fig. 9) and added more detail on this regard in the revision (Lines 190-196).

Fig. R2-2 (new Extended Data 7) | Negligible impact from changing amplitude of climate variability modes in the subtropical South Pacific, tropical Indian and Atlantic oceans. Inter-model relationship between changes (2000-2099 minus 1900-1999) in multi-year La Niña numbers and in the mean amplitude of (a) the South Pacific Meridional mode (SPMM) in FMA(1), (b) the Indian Ocean Basin Mode (IOBM) in MAM(1), (c) the North Tropical Atlantic (NTA) SST anomalies in MAM(1), and (d) the Atlantic Niño/Niña (measured by ATL3 index) in JJA(1). Linear fits (solid black line) are displayed together with correlation coefficient R and corresponding p value. Here the SPM index is defined as normalized SST anomalies over the subtropical southeastern Pacific (15°S–25°S and 110°W–90°W), the IOBM is defined as the first EOF mode of SST anomalies in the tropical Indian Ocean (20°S–20°N and 40°E–110°E), the NTA index is taken as normalized SST anomalies in (10°N–20°N and 60°W–20°W), and the ATL3 index is normalized SST anomalies averaged over the equatorial Atlantic cold tongue region (3°S–3°N and 20°W–0°E).

15. Lines 192-193: It is not clear if the northward broadened and stronger subtropical northeasterly anomalies in the 21st century continue from MAM(1) to JJA(1) and SON(1)

because the authors do not show the results in boreal summer and autumn (these seasons are indeed favorable for dynamic feedback, according to the introduction). For example, Extended Data Figure 9 does not include the results in SON(1).

We have now covered all seasons in Extended Data Fig. 4 to depict the evolution of the first-year La Niña.

16. Lines 210-213: Show an example for the regression line and scatter plots of the time series in the extended data, for instance, as this analysis is key but complicated and the potential uncertainty residing in the regression coefficient is not described.

Done. This has been added in Extended Data Fig. 8.

17. Lines 217-220: Why the x-axis and y-axis are scaled by the global SST change (GW) in Fig. 3b? Since the figure wants to indicate the relationship between the tropics-extratropics connection intensity and the multi-year La Nina number, such scaling seems to be not necessary.

We use such scaling to facilitate inter-model comparison and to make it consistent with other figures. Using raw data without scaling does not qualitatively alter our results.

18. Line 222: This comparison should be done in the main figure.

Done. We have added a panel in main Fig. 3 and revised the text.

19. Lines 226-228: Have the authors confirmed this result in the other SSP scenario simulations? Because this result is key, inter-scenario uncertainty should be considered.

Yes. We have confirmed the results in other SSP simulations (Supplementary Fig. 7).

20. Lines 249-250: It is not clear if the majority of the models project the increasing trend of the shortwave and longwave radiation in the subtropical northern Pacific because the Extended Data Fig. 13 does not include information about the inter-model consensus of the mean state changes.

We have confirmed that more than 80% of the models agree on the sign of the projected mean-state changes. Such information about inter-mode consensus have been included in supplementary Fig. 8 (in replacement of original Extended Data Fig. 13).

21. Lines 259-260 and 272: Why the Nino3 SST is used instead of the Nino3.4 SST, which is used in the other analysis of this manuscript?

Because strong El Niño tends to peak in the eastern equatorial Pacific (roughly the Niño3 region). We have tested that using Niño3.4 SST instead does not qualitatively alter the results.

22. Figure 1b: Does the CMIP6 models overestimate the multi-year La Nina amplitude between Dec(1) and Dec(2)? How many models could capture the observed amplitude realistically?

It is a long-standing issue that most climate models simulate an overly large ENSO amplitude

(e.g., Bellenger et al. 2014; Planton et al., 2021). In terms of the multi-year La Niña amplitude between Dec(1) and Dec(2), 4 out of 20 models underestimate the observed mean amplitude of -0.82°C while 16 out of 20 models overestimate it, with an MME of -1.13°C . However, we have tested and found that the simulated amplitude of multi-year La Niña between Dec(1) and Dec(2) does not affect the projected change in multi-year La Niña frequency under greenhouse warming.

References

Bellenger, H., Guilyardi, É., Leloup, J., Lengaigne, M. & Vialard, J. ENSO representation in climate models: from CMIP3 to CMIP5. *Clim. Dyn.* **42**, 1999–2018 (2014)

Planton, Y. Y. et al. Evaluating climate models with the CLIVAR 2020 ENSO metrics package. *Bull. Am. Meteorol. Soc.* **102**, E193–E217 (2021).

23. Figures 3a and 4d: The observed values in 1900-1999 could be provided.

Done. We have added the observed value (horizontal dashed line) in Fig. 3a and provided the observed value in the caption of Fig. 4d considering that the current Fig.4b is already too busy.

24. Figure 4d: The inter-model difference of these slopes could be analyzed to show that the future change in this relationship is in consensus among the models.

Done. We have added information about the inter-model consensus in figure caption.

25. Methods: CanESM5 and CanESM5-CanOE are identical to each other in the historical and SSP simulations (Table 2 of Swart et al. 2019 <https://doi.org/10.5194/gmd-12-4823-2019>). Even though the CanESM5 r1i1p1f1 realization could be analyzed, it includes a serious bug (See Appendix E of Swart et al. 2019: 'Notable in the differences are the “cold” spots in Antarctica, which arise from a mis-specified land fraction in p1, and were resolved in p2'). To avoid the double count of a single model, I would suggest removing the CanESM5 r1i1p1f1 realization. Then, the CanESM5-CanOE r1i1p2f1 results could be replaced with the CanESM5 r1i1p2f1 results as they are identical.

Done. Model sets have now been updated.

26. Methods: The MCM-UA-1-0 model (The Manabe Climate Model v1.0 - University of Arizona climate model, released in 1991) is not appropriate for examining the air-sea interaction mode as flux adjustments are implemented. <https://www.wdc-climate.de/ui/cmip6?input=CMIP6.CMIP.UA.MCM-UA-1-0> <https://explore.es-doc.org/cmip6/models/ua/mcm-ua-1-0>

The MCM-UA-1-0 model has been removed.

27. Lines 563-567: Why the triple La Nina or multi-year El Nino events do not increase in the future climate? The same mechanism for increasing the multi-year La Nina event occurrence frequency would work for these events.

Our mechanism centers on (strong) El Niño transitioning into double-year La Niña. Triple La Niña is rare in the historical record and the mechanism controlling its persistence especially from the second year to the third year is still uncertain. As for model simulations, our preliminary analysis indicates that there is no significant change in the meridional structure of the second-year La Niña between 1900-1999 and 2000-2099 (**Fig. R2-3**), which may explain why there is no inter-model consensus on the change in triple La Niña frequency under greenhouse warming. We have added this information in the Methods (Lines 620-627).

Multi-year El Niño is different in that it is usually not preceded by a La Niña. Therefore, different dynamics and mechanisms may apply to the change of multi-year El Niño, which is not the focus of our study.

Fig. R2-3 | Composite maps of SST ($^{\circ}\text{C}$; shadings) and surface wind stress (N m^{-2} ; vectors) anomalies for simulated multi-year La Niña events during D(2)JF(2) in (a) 1900-1999, (b) 2000-2099, and (c) their difference (2000-2099 minus 1900-1999) in the selected models. Only values that are statistically significant above the 95% confidence level based on a two-tailed Student's t-test are shown in c.

28. Extended Data Figures 7a and 8b: The results in the other emission scenarios should be shown.

Done. We have provided the results in other SSP simulations in Supplementary Figs. 5 and 6.

29. Extended Data Figure 7c: Why the number of models shown in Extended Data Fig. 7c is much smaller than 22?

Because two models (GFDL-CM4 and MPI-ESM1-2-LR) do not simulate a strong El Niño preceding multi-year La Niña in 1900-1999 (i.e., no data for the two models in x-axis). This information has been added in the figure caption.

Typos etc.:

1. Line 426: b instead of B

Corrected.

2. Line 561: influences instead of Influences

Corrected.

3. Line 603: I am not sure if “Codes are available from the corresponding author on request” is acceptable.

We have uploaded our code to a public database and provided an accessible link (Lines 663-665).

4. Extended Data Figure 9: The MCM-UA-1-0 and CESM2 models are missing in Extended Data Fig. 8 although the legend includes them.

Because these two models do not have SSH data. We have added this information in the caption and revised the legend.

5. Reference 46: IPCC AR6 has been published, so update the reference information.

Updated.

Response Referee #3

We thank the editor for facilitating clarification of the reviewer's comments, which we also list in red.

In this manuscript, the authors analyzed CMIP6 models to quantify and explain the projected increased number of multi-year La Niña under global warming. They stated that larger mean state warming in the subtropical northeastern Pacific could lead to both a faster warming in the eastern Pacific (i.e., more strong El Niños) and a weaker recharge process during La Niña, leading to more numbers of multi-year La Niña under global warming.

Thank you for your time and effort in evaluating our paper.

There seems to be a misunderstanding as we have not made the statement "...larger mean state warming in the subtropical northeastern Pacific could lead to both a faster warming in the eastern Pacific (i.e., more strong El Niños) ...".

Our main point is that a faster warming in the subtropical northeastern Pacific favors a northward broadened easterly wind anomaly by boosting a stronger and more sensitive negative NPM-like response to an equatorial eastern Pacific warm anomaly (through intensified WES feedback), which is further strengthened by a faster warming in the equatorial eastern Pacific. The consequence of the northward broadened easterlies is a slower heat recharge of the equatorial Pacific during the decay phase of first La Niña (because of a less negative wind stress curl), facilitating persistence of the cold anomalies of first year La Niña into the second year.

Major comments:

1. The citations are not accurate enough

The multi-year La Niña is a hot topic in the ENSO community for more than a decade, and relative research from different perspectives has been made. However, the authors do not provide sufficient citations for their discussions that reflect the community contributions over the past decades. For example, in Lines 20-24 and 43-44, the authors simply provide citations for El Niño or ENSO impacts but not specifically for multi-year La Niña impacts, such as Anderson et al. (2017), Jong et al. (2020), and Lopes et al. (2022), even though the sentences are discussing the multi-year La Niña impacts. Also, for the dynamics of multi-year La Niña in Lines 18-22 and 56-72, the authors focus on the precondition of the first-year La Niña, but not the transition from the first-year La Niña to the second-year La Niña. And other citations that are not accurate enough are in detailed comments.

Anderson, W., Seager, R., Baethgen, W., & Cane, M. (2017). Life cycles of agriculturally relevant ENSO teleconnections in North and South America. *International Journal of Climatology*, 37(8), 3297-3318.

Jong, B. T., Ting, M., Seager, R., & Anderson, W. B. (2020). ENSO teleconnections and impacts on US summertime temperature during a multiyear La Niña life cycle. *Journal of*

Climate, 33(14), 6009-6024.

Lopes, A. B., Andreoli, R. V., Souza, R. A., Cerón, W. L., Kayano, M. T., Canchala, T., & de Moraes, D. S. (2022). Multiyear La Niña effects on the precipitation in South America. *International Journal of Climatology*.

In the revised version, we have updated references to more accurately reflect community efforts over the past decades.

2. Lacking discussions on the transition from the first- to the second-year (or more) La Niña

The authors focus on the pre-condition of the first-year La Niña, including the strong El Niño and the larger trade winds (or wider meridional structure of La Niña). However, there is limited discussion on the condition from the first- to second-year La Niña, as well as second to third or third to fourth or more. The pre-condition of the first-year La Niña may lead to a colder ocean that favors the development of the second-year La Niña, but, as the authors have mentioned and shown, the increased number of the multi-year La Niña is larger than the increase of strong El Niño and no relation to the discharge rate. This indicates that there is something missing in the authors' argument, which may come directly from the transition between the first- and second-year La Niña.

Or oppositely, most of the results are tightly related to the projected increase of strong El Niño, such as the increase of svd time series, the negative PMM, and the weakening of the negative wind stress curl. However, does the precondition of La Niña really have a colder ocean state? And can this colder ocean state remain and impact the onset of the second-year La Niña? The authors have limited discussion on the following parts, which is the main component that directly influence the increase of multi-year La Niña.

Sorry for the confusion. First, yes, I think the transition between the first- and second-year La Niña should be discussed more.

For the sentences above, I messed up x-axis of the Extended Figure 8c with the pre-condition of the equatorial ocean state for the first-year La Niña, which is more direct for me, compared to the discharge rate of El Niño. If the authors want to argue that the pre-condition of equatorial heat content from El Niño is not as important as the pre-condition of meridional extension of La Niña, the pre-condition of equatorial ocean state for the first-year La Niña after strong El Niño may be better.

Yes, composite analysis shows that compared to 1900-1999 the equatorial upper ocean is significantly colder in 2000-2099 after the peak of first-year La Niña during March-July (**Fig. R3-1**). Such a colder upper ocean state will provide a favorable precondition for the second-year La Niña to develop in the following winter. The colder ocean state preconditioned by the first-year La Niña is consistent with SST and SSH fields shown in Extended Data Fig. 3b, c.

(**The following relates to your major comment #3 and #4**). During 2000-2099, Pacific easterly wind anomalies of the first-year La Niña in MAMJJ(2) are meridionally broadened,

and the associated negative wind stress curl anomalies are weaker (**Fig. R3-2 a, b, compare red and blue**). The consequence of the northward broadened easterlies is a slower upper ocean heat recharge (measured by temperature tendency) after the peak of the first-year La Niña, leading to a colder ocean state favorable for the development of second-year La Niña. An inter-model regression suggests that models with a larger weakening of the negative wind stress curl systematically simulate an anomalously colder upper ocean temperature tendency field (**Fig. R3-2 c**), which favors the persistence of cold anomalies of the first-year La Niña into the second year and ultimately contributes to an increase in multi-year La Niña frequency under greenhouse warming.

We have included Fig.R3-2 as new Extended Data Fig. 10 and added more discussion about the transition between the first- and second-year La Niña in the revised manuscript (Lines 241-246).

Fig. R3-1 | Colder equatorial upper ocean state preconditioned by the first-year La Niña. **a**, Multi-model mean equatorial (5°S-5°N average) upper ocean temperature anomalies (°C; shading) composited for simulated multi-year La Niña events in 1900-1999 during MAMJJ(2). **b**, As in **a**, but for 2000-2099. Stippling indicates where the ensemble mean temperature difference between 2000-2099 and 1900-1999 exceeds 1.0 s.d. of inter-model spread according to a Bootstrap method.

Fig. R3-2 (new Extended Data Fig. 10) | Northward broadened easterly anomalies and associated weakening of upper ocean heat recharge. **a**, Latitudinal distribution of zonal wind stress (TAUU) anomalies ($N m^{-2}$) in the Pacific (120°E-80°W) composited for multi-year La Niña events during MAMJJ(2) in the selected models. **b**, Histograms of 10,000 realizations of a Bootstrap method for wind stress curl anomalies (averaged in 0-10°N, 120°E-80°W; $N m^{-3}$) during MAMJJ(2) for all multi-year La Niña samples in 1900-1999 (blue bars) and 2000-2099 (red bars). Solid lines and shadings in **a**, **b** indicate multi-model mean and 1.0 standard deviation (s.d.) of a total of 10,000 inter-realizations based on a Bootstrap method, respectively. **c**, Inter-model regression of future change in equatorial (5°S-5°N average) temperature tendency during December(1) to September(2) onto the change in mean intensity of MAMJJ(2) wind stress curl anomalies. Stippling indicates statistical significance above the 90% level based on a two-tailed Student's t -test. All changes in **c** are calculated as 2000-2099 minus 1900-1999 and scaled by per °C of global warming to facilitate inter-model comparison. **During 2000-2099, easterly anomalies of the first-year La Niña are meridionally broadened, and the associated negative wind stress curl anomalies are weaker, slowing the heat recharge of the equatorial Pacific and providing a colder ocean state conducive to the development of second-year La Niña.**

3. The broadening of La Niña should lead to a weaker discharge, which is not seen in extended

figure 9.

A major argument is that the northward trades lead to the broadening of La Niña, as well as the weaker discharge. However, the two proxies of discharge, the SSH and wind stress curl are not matched, one showing no relation with La Niña but the other shows an impact. The authors should better justify this connection either using oceanic variables or other methods that more intuitively indicate the colder first-year La Niña (which is also not shown) is related to the broadening of PMM and also the northward change of trades.

Sorry for the typo. I mean recharge, but there is no such figure in the paper. I think I meant to refer to the Extended Fig. 8c, which is the one I misunderstood. The main concern from this part is whether the meridional extension of the first-year La Niña can really lead to less recharge and leading to a colder state to onset the second-year La Niña.

The meridional extension of the first-year La Niña leads to a less efficient upper ocean heat recharge thereby generating a colder state to onset the second-year La Niña.

Please see our response above to your major comment #2.

4. Not enough ocean analyses

The authors use SSH and wind stress curl as the proxies for ocean heat change. However, limited discussion on the pre-condition of the first- and second-year La Niña from ocean variables. As their argument is that the ocean has a colder precondition of the first-year La Niña that leads to the multi-year La Niña. The analyses directly related to the precondition should also be provided, especially with more direct oceanic variables, such as upper 300m potential temperature or 20-degree C isothermal depth, or other oceanic variables.

Yes. The authors have shown the possible weaker recharge of the first-year La Niña by the change of wind stress curl, but whether this wind stress curl change can really lead to a condition easier for the onset of the second-year La Niña is missing.

Our analysis of upper ocean temperatures clearly show that a less negative wind stress curl associated with the meridionally broadened easterlies lead to a condition easier for the onset of the second-year La Niña.

Please see our response above to your major comment #2.

Detailed comments:

1. The title should be narrowed/weakened as the manuscript is mainly about the strong El Niño to two-year La Niña, but not the entire multi-year La Niña.

The title has been modified to “Increased occurrences of consecutive La Niña events under global warming”.

2. Lines 17-18, the strong El Niño to Multi-year La Niña transition has been noticed, but the

2020-2023 three-year La Niña event is not preceded by a strong El Niño. The authors should not phrase as the multi-year La Niña is always from a strong El Niño. Also, only Wu et al. (2013) within the four cited publications discuss the preceding strong El Niño, which the authors need to revise with more accurate wording and citations.

We have rephrased the sentence as "...whereas La Niña tends to, **though not always**, develop after an El Niño...". Corresponding references have been updated.

3. Lines 20-24, citations 10-14 are mostly about El Niño impact, but not multi-year La Niña impacts.

Citations have been updated.

4. Lines 43-44, citations 17, 19, 20, 21, and 22 are not really for multi-year La Niña impacts, even though the sentence is for comparing single to multi-year La Niña impacts.

Citations have been updated.

5. Lines 55, citation 26 is about the onset of two strong El Niño events, but not the recharge process of La Niña.

The citation has been removed.

6. Lines 56-59, citations 28-30 are only for PMM but not directly for the La Niña development, please also cite some recent papers about negative PMM to La Niña.

Two relevant references have been added.

7. Line 93, the authors should also provide results from the non-chosen simulations and briefly discuss their difference compared to the chosen simulations, regarding their changes in future projection (not all models as in Line 132 but only the excluded). This is because the classification from skewness already excludes the possibility of multi-year La Niña not from strong El Niño.

We have included results from the non-selected models in Fig. 1a and Fig. 2a and discussed their difference from the selected models regarding projected multi-year La Niña changes (Lines 132-135).

8. Lines 96-97, the authors have excluded the three-year (or more) multi-year La Niña from classification and emphasized the importance of the precondition from extreme El Niño; however, as shown in Lines 128-130, the triple-year La Niña seems to have a different change, meaning the authors should indicate the limitation of the mechanism from a strong El Niño to two-year La Niña and better quantify them, as described in comment 12.

There seems to be a misunderstanding as we did not exclude three-year (or more) multi-year La Niña events in our results. As explicitly stated in Lines 96-97 and 'Definition of multi-year La Niña events' in Methods, a multi-year La Niña event is identified as when the La Niña

persists for **two or more consecutive years**. If a La Niña lasts for three years, the first two years were applied to our analysis, which is a conventional way commonly used to study multi-year La Niña events.

9. Lines 99-102, what is the increase when including multi-year La Niña for more than two years?

Our results already included multi-year La Niña events for more than two years.

10. Lines 114-116, the increase of two-year La Niña is more than the increase of strong El Niño. Does it mean that how the La Niña event continues is more important than the preceding El Niño? Or is the preceding El Niño not even the main reason?

No. Our finding is that under greenhouse warming a preceding El Niño becomes more efficient in inducing a meridionally broadened La Niña, which in turn boosts the probability that the first-year La Niña evolves into the second year. Therefore, both factors (the preceding El Niño and how La Niña continues) conspire to facilitate the increased frequency of multi-year La Niña.

11. Lines 129-131, where does this disagreement come from? Is this change significant?

Our analysis indicates that there is no inter-model consensus on the change of triple La Niña from 1900-1999 to 2000-2099 even under a high-emission warming scenario SSP585. Only a total of 8 out of 20 (40%) models show an increase in triple La Niña events with a non-significant MME increase of $12.2 \pm 28.1\%$. Please see lines 620-623.

12. Lines 157-161, as the authors have excluded the triple-year La Niña, meaning the two-year La Niña can only come from a neutral state or an El Niño state and less events are included. As a result, the “more than 75%” should be smaller. Also, do the 66 events refer to all multi-year events in the models that have an increase of multi-year La Niña?

We did not exclude the triple-year La Niña in our results (please see our response above to your detailed points#8 and #9).

The 66 events (now 54 events as we have revised the model set following Reviewer2's suggestion) refer to all multi-year La Niña events that last two consecutive years or more in the selected models.

13. Lines 173-176, when influence from extratropic is weak? But the PMM impacts tropical regions mainly during this period.

Using ONDJF to calculate the equatorial heat discharge rate does not alter our results (**Fig. R3-3**). We have revised the text.

Fig. R3-3 | No impact from changing equatorial heat discharge rate associated with strong El Niño. Same as Extended Data Fig.8b, but in ONDJF. Comparison of mean ONDJF changing rate of SSH in strong El Niño years, which measures the equatorial heat discharge rate associated with strong El Niño events, over 1900-1999 (blue bars) and 2000-2099 (red bars) in the selected models. Models that simulate a decrease are greyed out. Error bars denote 1.0 s.d. of 10,000 inter-realizations using a Bootstrap method of the samples for each model simulation and the multi-model ensemble mean.

14. Liens 184-185, “seems to be the case for 2020-2022 La Niña”? Is there a reference or figure? The citation 10 is Okumura’s paper in 2010 that proposes the inter-basin interactions.

We have added a reference.

15. Lines 185-189, this result seems to contradict the explanation of why the increased number of multi-year La Niña is larger than the increase of strong El Niño. The authors stated that the faster warming in the equatorial eastern and subtropical northeastern Pacific leads to more strong El Niños and meridionally broader first-year La Niñas; however, in the end, these strong El Niños and broader first-year La Niñas should lead to a colder ocean state to onset the second La Niña, meaning the discharge process should also have relation with the increased number of the multi-year La Niña, i.e., the mean discharge rate should be proportional/related to the increased ratio of the broader first-year La Niña after all strong El Niños.

Sorry. Same issue as before for the extended Fig. 8. I think I messed up the Extended Fig 8 and 9 during the review process. The main comment here is similar to the above for not discussing the transition between the first- and second-year La Niña.

There is no contradiction. We have added discussion about the transition between the first- and second-year La Niña using upper ocean temperature fields.

Please see our response above to your major point#2.

16. Lines 195-198, the SVD tries to capture the relation between ENSO peak phase (DJF SST in the tropical Pacific) and the following atmospheric responses (MAM SLP in the north

Pacific). However, this cannot infer to the influence on multi-year La Niña

Misunderstanding. We did not use this text to infer an influence on multi-year La Niña. The text describes the SVD analysis.

17. Lines 206-207, “most of”? is there a figure showing this? Extended Data Fig.1a do not support this statement, as well as when we consider the model La Niña. If it is not ‘most of’, what is the ratio of multi-year La Niña that can be explained by this SVD?

We meant a substantial portion (more than 83%, 45 out of 54 events in aggregation) of the increased multi-year La Niña events occur after a strong El Niño. We have made it clear in the revision.

18. Lines 216-217, more strong El Niño leads to an increase of the north subtropical atmospheric responses (an increase of svd time-series), but cannot directly connect to the precondition of the first-year La Niña, or even the onset of the second-year La Niña.

Sorry for not stating it clearly. My main question is whether the pre-condition of **“equatorial ocean heat content”** for the first-year La Niña (i.e. coming from the discharge of El Niño or development of the meridional extended La Niña).

It seems to be out of the context as original Lines 216-217 simply describe an increase in the extratropical response to the tropical forcing through the SVD.

We assume the reviewer’s question is about the relation between the SVD and the equatorial upper ocean state preconditioned by the first-year La Niña. To this end, we perform an inter-model regression of equatorial upper ocean temperature change after the first-year La Niña in MAMJJ(2) onto the strength change of extratropical response to tropical forcing calculated from the SVD (**Fig. R3-4**). Models with a greater increase in the extratropical response systematically generates a colder upper ocean state in the central-eastern equatorial Pacific (**Fig. R3-4a**). The colder ocean state serves as a favorable precondition for the occurrence of second-year La Niña, contributing to the increased frequency of multi-year La Niña. A similar relationship is also found when strong El Niño events are excluded (**Fig. R3-4b**), further underscoring the robustness of our mechanism and the utility of the SVD analysis.

Fig. R3-4 | Impact of enhanced extratropical response to tropical forcing on equatorial upper ocean temperature preconditioned by the first-year La Niña. **A**, Inter-model regression of change (2000–2099 minus 1900–1999) in MAMJJ(2) equatorial (5°S–5°N average) upper ocean temperature anomalies onto the change in the extratropical response to tropical forcing, both scaled by per °C of global warming in each model. Stippling indicates statistical significance above the 90% level based on a two-tailed Student’s t-test. **B**, As in **a**, but after excluding strong El Niño years.

19. Lines 226-230, this 77.2% does not come from the review paper from Amaya (2019). Where does it from?

This was obtained from Extended Data 9a (in replacement of original Extended Data 11a). We have made it clear in the revision.

20. Lines 232-235, why is there a weakening of negative wind stress curl but not the discharge rate? Are they tightly related? Is there really a colder precondition for the first-year La Niña?

Sorry for the typo. Yes, I meant recharge. I think the precondition of the **equatorial** ocean state for the first-year and second-year La Niña should be discussed.

Yes, there is a colder ocean state preconditioned by the first-year La Niña that is associated with the weakening of negative wind stress curl. Please see our response above to major point#2.

21. Lines 238-239, “re-intensify”? where is this from? Other papers or result from any figure?

This has been rephrased to “develop”.

22. Lines 236-241, how does this ocean-atmosphere coupling change impact the transition from the first- to the second-year La Niña?

The intensified ocean-atmosphere coupling, that is, a strengthened positive Bjerknes feedback in the equatorial Pacific, will boost any remaining cold(er) temperature anomalies of the first-year La Niña to grow and persist, thus facilitating the transition from the first- to the second-year La Niña.

23. Lines 249-252, how does the broadening of first-year La Niña (at roughly 20N 140W) relate to the northward extension of PMM (at roughly 20N 180E)?

Lines 49-52, “enhances the WES feedback more to the north, such that the NPMM-like SST and surface wind anomalies induced by El Niño warm SST anomalies are **broadened northward and are more sensitive**”. To be succinct, how does the northward extended WES feedback contribute to the meridional broadened La Niña and be more sensitive? Can the authors point to the figure that explains this causation?

As the WES feedback is enhanced more to the north due to a faster background mean-state warming in the subtropical northeastern Pacific (Fig.4a, b), the NPMM-like anomalies are more easily to develop (i.e., more sensitive to small atmospheric or oceanic disturbances) from a more-northern location, inducing easterly anomalies all the way towards the equator that are also more northward broadened. These processes are reflected in a significant inter-model relationship between WESp and a meridional TAUU gradient in MAMJJ(2), a surrogate for the meridional width of the first-year La Niña (**Fig. R3-5**); models with a stronger increase in WESp systematically generate a larger decrease in the easterly meridional gradient during MAMJJ(2), therefore contributing to a more meridionally broadened first-year La Niña.

Fig. R3-5 | Enhanced WES feedback and meridionally broadened easterly wind anomalies. Inter-model relationship between change in the mean north-minus-south (120°E-80°W, 10°N-15°N minus 0°N-5°N) meridional gradient of zonal wind stress (TAUU; N m^{-2} per °C of GW) anomalies during MAMJJ(2) and change in WESp (W s m^{-3} per °C of GW) from February to July over the subtropical northeastern Pacific (155°W-115°W, 15°N-35°N; blue box in Fig. 4a). Linear fit (solid black line) is displayed together with correlation coefficient R and corresponding p value. A decrease (negative change) in the north-minus-south TAUU gradient represents a weakened easterly wind meridional gradient and therefore a meridionally broadened structure for the first-year La Niña.

24. Lines 256-258, the correlation in 4c seems to be dominated by the outliers (One in the very top-right and also the bottom-left). How does the correlation changes when excluding those outliers?

The correlation is still significant ($r=0.67$, $p<0.01$).

25. Lines 267-270, does it mean other processes may contribute to the increase of multi-year La Niña but not only the increase of WES feedback?

The authors evidence the stopped multi-year La Niña increase by in Figure 14c; however, if considering the historical period when the slope between winter OLR and Niño3 steadily decreases, why the multi-year La Niña occurrence in Fig. 2b does not have the same steady increase of occurrence?

Because in the pre-1960s when greenhouse warming impact is weak, the occurrence of multi-year La Niña is largely controlled by natural variability.

26. Lines 271-279, the Strengthened extratropical response to tropical forcing excluding strong El Niño simply means El Niño, in general, can induce the circulation change in the northern extratropical Pacific, but not negative PMM or even the possible following La Niña. Also, the correlation should be high “mathematically” in extended figure 15b since the svd is calculated from the SST in the equatorial Pacific, and the first two principal components have a strong signature in the equatorial eastern Pacific.

The main point here is that the authors should show how effective can this change lead to the onset of the second-year La Niña.

Please see our response above to your detailed point#18.

27. Lines 281-292, how the northward extension of trade winds can lead to a weaker discharge of El Niño, then a colder precondition for the first-year La Niña, and then the onset of the second-year La Niña is not clearly justified, even though the authors show the weakening of negative wind stress curl.

Yes. Again. Related to the misunderstanding of that figure.

Please see our response above to your major point#2.

Additional comments:

1. The authors argue with evidence that the mean state change -> more strong El Niño in D(0)FJ(1) -> even more (sensitivity) negative PMM MAM(1) -> more broaden first-year La Niña D(1)JF(2) -> possible weaker recharge for the first-year La Niña from wind stress curl in MAMJJA(2) -> more second-year La Niña. Due to this complexity, the authors should provide a schematic for this chaining argument.

We have included a schematic in Supplementary Fig. 10 (please also see Fig. R3-6 below).

Fig. R3-6 (Supplementary Fig. 10) | Schematic depicting the mechanism for increased frequency of consecutive La Niña events under greenhouse warming. Besides heat discharge in the tropical Pacific, an El Niño warm anomaly in the equatorial eastern Pacific can induce a negative NPM-like SST and wind anomaly (black arrows) in the subtropical northeastern Pacific through an atmospheric Gill-type response (red arrow). The negative NPM-like pattern that features increased extratropical trade winds then grows in boreal spring-summer under the WES feedback, favoring development of a meridionally broad La Niña in the following winter by initiating an SST cooling from the subtropics. The meridionally wide pattern of SST and easterly wind anomalies of the La Niña is accompanied by a weak anticyclonic (AC) wind stress curl at more-extratropical latitudes, slowing the heat recharge of the equatorial Pacific. Therefore, the cold SST anomalies can persist through the decaying phase of the first-year La Niña into spring, and with a seasonal strengthened air-sea coupling in summer and autumn, the cold SST anomalies re-intensify and develop into another La Niña event in the next winter. Under transient greenhouse warming, a faster mean-state warming in the subtropical northeastern Pacific favors a northward broadened easterly wind anomaly by enhancing the WES feedback more to the north, boosting a stronger and more sensitive negative NPM-like response to an equatorial eastern Pacific warm anomaly, which is further strengthened by a faster mean-state warming in the equatorial eastern Pacific. The consequence

of the northward broadened easterlies is a slower heat recharge in the upper equatorial Pacific (indicated by thick gray arrow) because of a weakened negative wind stress curl (blue circular arrow). This leaves a colder upper ocean conducive for the cold anomalies of the first-year La Niña to persist into the second year with additional help of a warming-induced increase in air-sea coupling in the tropical Pacific.

2. The argument in this manuscript can be considered as a follow-up study of the Park et al. (2020), as the mechanism for Strong El Niño to multi-year La Niña are mostly following the paper. The authors found the mechanism is strengthened under greenhouse warming and leading to more multi-year La Niña; however, throughout the manuscript, I did not see enough discussion on the Park et al. (2020) paper, which not only making the manuscript too complicate, but also not giving enough credit for the Park et al. (2020).

We have added more discussion about Park et al. (2020) in the revision.

3. How does the strong El Niño induced wind in spring lead to a broadened La Niña in winter? The authors should also either show or describe the persistence of such negative PMM from spring to winter, especially under greenhouse warming.

To reflect more on the role of a negative NPMM, we have included a complete evolution of the first-year La Niña in Extended Data Fig. 4.

4. Why using the MAMJJA(2) in Extended Figure 12 while Park et al. (2020) use the winter to early spring. Also, what is the SST and wind pattern in these months?

While Park et al. (2020) showed wind patterns both in OND0 and FMA1 (see their Fig.6), they focused on FMA1, the decay season of the first-year La Niña, to illustrate the importance of the meridional broadened easterly winds in sustaining a multi-year La Niña event compared to a single-year La Niña in observations.

For a similar reason, we meant to focus on the decay phase of the first-year La Niña to diagnose the impact of weakened recharge on the persistence of cold anomalies in CMIP6 models under greenhouse warming. Boreal spring to early summer is the time when the first-year La Niña starts to decay after its peak in boreal winter but before the background seasonal air-sea coupling strengthen in late boreal summer-autumn.

To be consistent with other figures, we have now opted to use MAMJJ(2) but results remain essentially the same with those of MAMJJA(2). The SST and wind patterns during these months have been added to Extended Data Fig. 4.

5. As mentioned from the reviewer 1, how do the authors explain the large number of multi-year La Niña under current climate but the mean state change (more cooling in the eastern equatorial Pacific) is different to the change under greenhouse warming projections.

The current observational record is subject to much uncertainty especially in the first half of

the 20th century. Therefore, the observed change itself may not be relied upon without any additional corroborating evidence. Although models tend to underestimate the observed SST warming trend between the western and eastern equatorial Pacific (**Fig. R3-6a**; please also note large inter-product differences in reanalysis datasets), there are models that are able to simulate both an increase in multi-year La Niña frequency and a faster warming in the western than eastern equatorial Pacific in the last 40 years (1981-2020) than 40 years before (1941-1980) (**Fig. R3-6a, the upper right quadrant**), suggesting an influence from natural variability on recent multi-year La Niña change.

Further, a sensitivity test indicates that, even if data quality were not an issue, detecting observed change may depend on the time period in which the frequency of multi-year La Niña is diagnosed. For example, when comparing 1945-1979 with 1980-2014, there is **no change** in the frequency of multi-year La Niña events in observations (**Fig. R3-6b**), though there is still a strengthened west-minus-east SST gradient in the equatorial Pacific. Therefore, the observed multi-year La Niña change is subject to uncertainty and may be influenced by natural variability.

In the revised version, we have added a new section ‘Model bias and recent multi-year La Niña change’ in Methods to discuss about model bias and observed multi-year La Niña changes (Lines 136-139).

Fig. R3-6 | Multi-year La Niña frequency and mean equatorial Pacific zonal SST gradient during recent decades. a, Relationship between change (1981-2020 minus 1941-1980) in multi-year La Niña numbers and trend in mean equatorial Pacific zonal SST gradient (°C per decade) during 1941-2020. Shown are results from the selected CMIP6 models and three observational reanalysis datasets. The zonal SST gradient is defined the SST difference between the western Pacific (150°E-180°, 5°N-5°S) and the central-eastern Pacific (150°W-

120°W, 5°N-5°S), so that a positive trend in the gradient represents a faster warming in the western than eastern equatorial Pacific. **b**, As in **a**, but for the period of 1945-2014.

6. If the broadened first-year La Niña is the main reason of the onset of the second year La Niña, the authors should focus on this relation to make the story less complicated, and discuss the increased of strong El Niño as an additional reason of the increase of multi-year La Niña. Otherwise, the strong El Niño largely impede the readability of this manuscript since the authors show the strong El Niño affect the first La Niña by only its frequency but not with its “discharge” process. And then the first-year La Niña impact the second-year La Niña through its weakened “recharge” process.

It is commonly conceived that strong El Niño (and associated strong equatorial upper ocean heat discharge) is the main reason for the occurrence of multi-year La Niña. That is why we choose to start from strong El Niño and focus on the change of first-year La Niña, which we think is reasonable because otherwise readers might assume that the change in the frequency of multi-year La Niña may simply follow the change in frequency of strong El Niño (which is not the case). Apart from this, our logic is essentially similar to yours.

7. If the broaden first-year La Niña is important, can this explain why there are less third-year La Niña under greenhouse warming? For example, since the second-year La Niña is not broaden enough, the third-year La Niña cannot be generated?

Triple La Niña is rare in the historical record and its mechanism is still uncertain. As for model simulations, our analysis indicates that there is no significant change in the meridional structure of the second-year La Niña between 1900-1999 and 2000-2099 (**Fig. R3-7**), which may explain why there is no inter-model consensus on the change in triple La Niña frequency under greenhouse warming (Lines 620-627).

Fig. R3-7 | Composite maps of SST ($^{\circ}\text{C}$; shadings) and surface wind stress (N m^{-2} ; vectors) anomalies for simulated multi-year La Niña events during D(2)JF(2) in (a) 1900-1999, (b) 2000-2099, and (c) their difference (2000-2099 minus 1900-1999) in the selected models. Only values that are statistically significant above the 95% confidence level based on a two-tailed Student's t-test are shown in c.

Replied by Shih-Wei Fang

Reviewer Reports on the First Revision:

Referees' comments:

Referee #1 (Remarks to the Author):

I appreciate the authors' detailed response and revision to my previous comments. I agree with the authors that the observed multi-year La Niña change is subject to uncertainty and natural variability based on a single short realization. This reminds me that the one ensemble member used for each model may also be subject to large internal variability (Lee et al. 2021; Maher et al. 2023), which is also pointed out by reviewer #2. The changes in ENSO might be time-varying, and many features might be missed by simply taking the averages of the 20th and 21st centuries (Maher et al. 2023). The authors should at least include some discussions of these influences on their results. Other than this comment, I would recommend accepting this manuscript for publication in Nature.

Lee, J., Planton, Y. Y., Gleckler, P. J., Sperber, K. R., Guilyardi, E., Wittenberg, A. T., et al. (2021). Robust evaluation of ENSO in climate models: How many ensemble members are needed? *Geophysical Research Letters*, 48, e2021GL095041. <https://doi.org/10.1029/2021GL095041>

Maher, N., Wills, R. C. J., DiNezio, P., Klavans, J., Milinski, S., Sanchez, S. C., Stevenson, S., Stuecker, M. F., and Wu, X.: The future of the El Niño-Southern Oscillation: Using large ensembles to illuminate time-varying responses and inter-model differences, *Earth Syst. Dynam. Discuss.* [preprint], <https://doi.org/10.5194/esd-2022-26>, in review, 2022.

Referee #2 (Remarks to the Author):

March 2023,

Review of the revised manuscript titled "Increased occurrences of consecutive La Niña events under global warming" by T Geng et al.

In the original submission, I had three major concerns about originality, presentation, and accuracy. The authors addressed most of my previous concerns. As for originality, the present manuscript described the advance from recent works (Jia et al. 2021 and Fan et al. 2022) in Lines 301-303; Two-way interaction and multi-year La Nina change are key for this work to be novel. The presentation is improved as the comparison among the SSP scenarios is now explicitly shown in the main figures. The numbers in the first paragraph are supported by the main figures in the revised manuscript. I also confirmed that methods for estimating the statistical significance and error bars in Figures 1-5 are properly explained in the figure captions (one could be improved; see the specific comment 6). It is also nice that their key codes are now provided in a Zenodo archive. However, the accuracy of the results could be discussed more in the text. Please see the specific comments 1-5 below.

Specific comments:

1. The definition of climatology needs to be paid more attention to.

In Lines 541-543, the authors added that "calculating anomalies separately for 1900-1999 and 2000-2099 with reference to the respective monthly climatology does not alter our conclusions" and showed me a figure (Fig. R2-1) in their reply. Line 541-543 is not enough to describe the sensitivity of results on the basic state definition. The results are actually changed quantitatively in some models (NESM3, MIROC6, CIESM, CanESM5, MRIOC-ES2L, etc., at a glance). Although the multi-model ensemble average is less sensitive to the mean state definition, a more appropriate

definition of the basic state is desired. It is rather natural that the anomalies and selected La Nina events can be sensitive to the climatologies as the warming and its seasonal cycle are added onto the historical climatology. It is easy to excuse by saying like 'this method has been often used in previous works...', but it would be better to add the results in Figure R2-1 to emphasize the robustness of the present results, and also the text should discuss the potential issue of how to define the basic state as a caveat.

2. The model selection method in Lines 82-93 could be justified more in terms of the simulated ENSO dynamics.

In Lines 134-135, the authors mentioned that "... which possibly related to the overly weak nonlinear ENSO dynamics (Fig. 1a)". However, Figure 1a only showed the SST skewness, which is statistics. The weakness of nonlinear ENSO dynamics is not obvious from the figure. I wonder if these selected models having positive skewness could simulate the ENSO dynamics better as well. At least, two recent papers, Bayr and Latif (2022) and Hayashi et al. (2020), showed that models simulating the positive ENSO SST skewness tend to have better ENSO dynamics in terms of atmospheric feedback and ocean nonlinear dynamic heating, respectively. Adding this information may provide a positive reason for this model selection method.

Bayr, T., Latif, M. ENSO atmospheric feedbacks under global warming and their relation to mean-state changes. *Clim Dyn* (2022). <https://doi.org/10.1007/s00382-022-06454-3>

Hayashi, M., Jin, FF. & Stuecker, M.F. Dynamics for El Niño-La Niña asymmetry constrain equatorial-Pacific warming pattern. *Nat Commun* 11, 4230 (2020).

<https://doi.org/10.1038/s41467-020-17983-y>

3. Concatinating piControl and historical with 60-year smoothing in Fig. 1b is inappropriate. As the authors mentioned in the method section (Lines 545-548), two independent time series are still connected. For the present purpose, the authors do not need to concatenate the piControl and historical time series in the 60-year window analysis (Fig. 1b). The internal variabilities are not connected between the two in time, and thus the 60-year smoothing between the two is physically misleading. The authors could analyze these two (blue and gray) time series separately and, for instance, create combined two panels in Fig. 1b.

4. The data is seemingly inconsistent between Figure 3 and Extended Data Figure 8.

Why is the slope in 2000-2099 different between Fig. 3a and Extended Data Fig. 8h? The slope in 1900-1999 looks close to each other (~ 0.5) but that in 2000-2099 is about 0.7 in Fig. 3a while 0.6 in the Extended Data Fig. 8h. Is this because of the averaging method difference? Showing a consistent result may eliminate this confusion.

5. The results without scaling in Figure 3b and Extended Data Figure 7 could be shown to confirm if "Using raw data without scaling does not qualitatively alter our results" in the reply would be true. Extended Data Figure 7 was used for denying the other potential mechanisms, so the results should be confirmed by changing the method a bit. This must be easy for the authors.

6. The observed slope in the caption of Fig. 4 could have a 95% confidence interval to compare with the 1900-1999 slope in Fig. 4d.

Typos:

Line 407: Replace "pacific" with "Pacific".

Referee #3 (Remarks to the Author):

Comments on "Increased occurrences of consecutive La Niña events under greenhouse warming" by Geng et al.

Overall comment:

In this manuscript, the authors analyzed CMIP6 models to quantify and explain the projected increased number of multi-year La Niña under global warming. They stated that larger mean state warming in the subtropical northeastern Pacific could lead to both a faster warming in the eastern Pacific (i.e., more strong El Niños) and a weaker recharge process during La Niña, leading to more numbers of multi-year La Niña under global warming.

The topic is crucial and interesting, especially since there was a three-year La Niña event last year. I appreciate the authors did a lot of effort to analyze those results and the revised manuscript has a better readability and more accurate reference and evidence. And indeed, the authors propose a possible explanation for "parts of" the increased number of multi-year La Niña under greenhouse warming. That said, the scientific story of why multi-year La Niña is projected to increase is partially convincing but several critical comments still need to be addressed. I thus cannot recommend the publication of the current manuscript in Nature. My detailed comments are as follows.

Major comments:

1. The confusion between multi-year La Niña and the consecutive ENSO events of strong El Niño to La Niña to La Niña

The authors explain the increase of multi-year La Niña by the increase of consecutive ENSO events of strong El Niño to La Niña to La Niña. However, this explanation only explains part of the increase in multi-year La Niña but the authors do not clearly reflect this difference in the title and the main texts. As a result, this manuscript is still difficult to follow and the content is complicated. For example, in the abstract, after the increased frequency of multi-year La Niña, the authors started explaining warming leading to more strong El Niño, which is not intuitive since the reader expects an explanation of why multi-year La Niña increases. Similar confusion can be found throughout the manuscript, which reduces the readability.

2. Lack of discussing the tripole (or more) La Niña event

As we have experienced another tripole La Niña event, the authors should also discuss the entire tripole La Niña event, but not consider only the first two years of them. Since La Niña is mostly either preceded by another La Niña or an El Niño event, by excluding the second-to-third year of tripole La Niña, the manuscript only discusses the El Niño-to-multi-year La Niña transition. The authors should discuss how La Niña-to-multi-year La Niña transition is changed and whether there is a meridionally broader La Niña when a La Niña happens before. In fact, excluding the second-to-third year of tripole La Niña enhances the statistical significance in this manuscript, such as Fig. 1b, 1d, and the increased frequency of multi-year La Niña in Fig. 2. This is the main reason that the tripole (or more) La Niña needs to be discussed.

To be noticed the authors reply that they already considered the tripole La Niña but, in fact, they only discussed the first two years of the tripole La Niña, which is not the main concern from previous comments. Instead, the tripole (or more) La Niña should be considered as two (or more) two-year La Niña events, and the changes in the statistics shown in Fig. 2 need to be provided in the text and supplementary.

Detailed comments:

1. Still, the title should be narrowed/weakened as the manuscript is mainly about the strong El Niño to two-year La Niña, but not the entire multi-year La Niña or consecutive La Niña.

2. Lines 18-19, to my knowledge, the meridionally broader first La Niña is mainly found when strong El Niño precedes. Please either provide the reference for meridionally broader first La Niña when a strong El Niño does not precede or revise the sentence to specifically the strong El Niño to multi-year La Niña transition.

3. Lines 26-31, after showing the increase of multi-year La Niña, the authors focus only on El Niño, which confuses the reader, as people will expect an explanation of why multi-year La Niña increase. The authors should guide the reader to the increase of the El Niño-to multi-year La Niña.

4. Lines 43-44, citations 17, 19, 20, 21, and 22 are not really for multi-year La Niña impacts, even though the sentence is for comparing single to multi-year La Niña impacts.

5. Lines 72-73, the cooling left from the first La Niña increases the “probability” of creating the second year La Niña, which needs to be included.

6. Lines 76-78, this

7. Lines 79-80, “from strong El Niño” or a similar description needs to be mentioned.

8. Line 89, “do”?

9. Line 100-101, this is a surprising result, as previous studies show most CMIP models' difficulty in simulating observed multi-year La Niña, such as Fang and Yu (2020). Is it because of a different classification method? The reason should be mentioned in the manuscript.

Fang, S. W., & Yu, J. Y. (2020). Contrasting transition complexity between El Niño and La Niña: Observations and CMIP5/6 models. *Geophysical Research Letters*, 47(16), e2020GL088926.

10. Lines 96-97, the authors still excluded the three-year (or more) multi-year La Niña from classification since only the first two years of the tripole event are considered. That is, the authors exclude the multi-year La Niña coming from La Niña, which is the reason why the composite of multi-year La Niña has a strong El Niño signal than the single-year La Niña event (Fig. 1b). And why there is a meridionally broader first-year La Niña (fig. 1d). Suggestion is provided in major comment 2.

11. Line 118, the multi-year La Niña considered in this study, can only come from El Niño or Neutral states since the authors do not consider the tripole La Niña as two two-year La Niña events. As a result, the “multi-year La Niña” and “all La Niña” are misleading. The authors should mention the changes in statistics when considering tripole La Niña as two events.

12. Line 160, the piControl is not ‘unforced’.

13. Extended Data Fig. 10, there is no d in the figure but in the caption.

14. Extended Data Fig. 10c, how does the figure create? Since it is the regression between temperature of D(1)JFMAMJJAS(2) and wind stress of MAMJJ(2). Does it mean the regression is only performed on multi-year La Niña events? If so, how many events are there for each experiment? It may not be representative if the events are limited.

15. Lines 248-250, “the persisting cold SST anomalies of the first La Niña are then amplified by the stronger Bjerknes positive feedback in late boreal summer-autumn, and develop into a second La Niña in the following winter”. Please provide corresponding evidence for this argument.

Author Rebuttals to First Revision:

Response to Referee #1

Thank you for your positive comments!

I appreciate the authors' detailed response and revision to my previous comments. I agree with the authors that the observed multi-year La Niña change is subject to uncertainty and natural variability based on a single short realization. This reminds me that the one ensemble member used for each model may also be subject to large internal variability (Lee et al. 2021; Maher et al. 2023), which is also pointed out by reviewer #2. The changes in ENSO might be time-varying, and many features might be missed by simply taking the averages of the 20th and 21st centuries (Maher et al. 2023). The authors should at least include some discussions of these influences on their results. Other than this comment, I would recommend accepting this manuscript for publication in Nature.

We have added discussion in the revision. Please see Lines 554-567.

Lee, J., Planton, Y. Y., Gleckler, P. J., Sperber, K. R., Guilyardi, E., Wittenberg, A. T., et al. (2021). Robust evaluation of ENSO in climate models: How many ensemble members are needed? *Geophysical Research Letters*, 48, e2021GL095041. <https://doi.org/10.1029/2021GL095041>

Maher, N., Wills, R. C. J., DiNezio, P., Klavans, J., Milinski, S., Sanchez, S. C., Stevenson, S., Stuecker, M. F., and Wu, X.: The future of the El Niño-Southern Oscillation: Using large ensembles to illuminate time-varying responses and inter-model differences, *Earth Syst. Dynam. Discuss.* [preprint], <https://doi.org/10.5194/esd-2022-26>, in review, 2022.

Response to Referee #2

Thank you for your helpful comments!

Specific comments:

1. The definition of climatology needs to be paid more attention to.

In Lines 541-543, the authors added that "calculating anomalies separately for 1900-1999 and 2000-2099 with reference to the respective monthly climatology does not alter our conclusions" and showed me a figure (Fig. R2-1) in their reply. Line 541-543 is not enough to describe the sensitivity of results on the basic state definition. The results are actually changed quantitatively in some models (NESM3, MIROC6, CIESM, CanESM5, MRIOC-ES2L, etc., at a glance). Although the multi-model ensemble average is less sensitive to the mean state definition, a more appropriate definition of the basic state is desired. It is rather natural that the anomalies and selected La Nina events can be sensitive to the climatologies as the warming and its seasonal cycle are added onto the historical climatology. It is easy to excuse by saying like 'this method has been often used in previous works...', but it would be better to add the results in Figure R2-1 to emphasize the robustness of the present results, and also the text should discuss the potential issue of how to define the basic state as a caveat.

We have included Fig. R2-1 as new Supplementary Fig.12 and added more discussions in this regard. Please see the following and Lines 548-552.

“As expected, definition of events and the choice of period over which climatology is calculated could affect our results. As a test, we calculate anomalies separately for 1900-1999 and 2000-2099 with reference to the respective monthly climatology. Although changes in individual models occur, the multi-model ensemble results hold (Supplementary Fig.12).”

2. The model selection method in Lines 82-93 could be justified more in terms of the simulated ENSO dynamics.

In Lines 134-135, the authors mentioned that "... which possibly related to the overly weak nonlinear ENSO dynamics (Fig. 1a)". However, Figure 1a only showed the SST skewness, which is statistics. The weakness of nonlinear ENSO dynamics is not obvious from the figure. I wonder if these selected models having positive skewness could simulate the ENSO dynamics better as well. At least, two recent papers, Bayr and Latif (2022) and Hayashi et al. (2020), showed that models simulating the positive ENSO SST skewness tend to have better ENSO dynamics in terms of atmospheric feedback and ocean nonlinear dynamic heating, respectively. Adding this information may provide a positive reason for this model selection method.

Bayr, T., Latif, M. ENSO atmospheric feedbacks under global warming and their relation to mean-state changes. *Clim Dyn* (2022). <https://doi.org/10.1007/s00382-022-06454-3>

Hayashi, M., Jin, FF. & Stuecker, M.F. Dynamics for El Niño-La Niña asymmetry constrain equatorial-Pacific warming pattern. *Nat Commun* 11, 4230 (2020). <https://doi.org/10.1038/s41467-020-17983-y>

Yes, better simulation of ENSO skewness is associated with more realistic simulation of ENSO dynamics, as shown by numerous studies including the two papers provided.

We have added this information in the revised manuscript (see Lines 95-96).

3. Concatinating piControl and historical with 60-year smoothing in Fig. 1b is inappropriate. As the authors mentioned in the method section (Lines 545-548), two independent time series are still connected. For the present purpose, the authors do not need to concatenate the piControl and historical time series in the 60-year window analysis (Fig. 1b). The internal variabilities are not connected between the two in time, and thus the 60-year smoothing between the two is physically misleading. The authors could analyze these two (blue and gray) time series separately and, for instance, create combined two panels in Fig. 1b.

We take it as that you are referring to Fig. 2b.

You are right that there is no need to concatenate the two (piControl and historical) experiments together. We have taken your advice, analysed the two time series separately and generated two combined panels in Fig.2b. The corresponding text has also been revised (Lines 153-164).

4. The data is seemingly inconsistent between Figure 3 and Extended Data Figure 8.

Why is the slope in 2000-2099 different between Fig. 3a and Extended Data Fig. 8h? The slope in 1900-1999 looks close to each other (~0.5) but that in 2000-2099 is about 0.7 in Fig. 3a while 0.6 in the Extended Data Fig. 8h. Is this because of the averaging method difference? Showing a consistent result may eliminate this confusion.

There is no inconsistency. Fig. 3a shows the total response, which is the product of regression coefficient (i.e., slope in Extended Data Fig.8g,h) and the standard deviation of SST expansion coefficient time series. This information is given in Lines 222-225.

5.The results without scaling in Figure 3b and Extended Data Figure 7 could be shown to confirm if "Using raw data without scaling does not qualitatively alter our results" in the reply would be true. Extended Data Figure 7 was used for denying the other potential mechanisms, so the results should be confirmed by changing the method a bit. This must be easy for the authors.

The results should be scaled by global warming for Fig.3b, so as to exclude the possibility that the relationship is due to difference in climate sensitivity, because even without scaling, the relationship is significant above the 95% confidence level ($p=0.026$), with a correlation of $r=0.51$. Thus, even after removing the influence from the difference in climate sensitivity, the relationship holds. For Extended Data Fig. 7, with or without scaling, there is no significant relationship (without scaling, correlations are -0.15, 0.11, 0.02, and 0.12, for panels a, b, c, and d, respectively).

6. The observed slope in the caption of Fig. 4 could have a 95% confidence interval to compare with the 1900-1999 slope in Fig. 4d.

Added.

Typos:

Line 407: Replace "pacific" with "Pacific".

Corrected. Thank you for picking up.

Response to Referee #3

Thank you for your time and effort in evaluating our paper.

Major comments

1. The framing is confusing. The relationship between multi-year La Niña and the consecutive ENSO events of strong El Niño to La Niña to La Niña...

We believe that our framing is clear and that some of the reviewer's statements are inaccurate. For example, contrary to the referee's statement, our first paragraph does not include wording about "warming leading to more strong El Niño."

2. The paper should discuss tripole La Niña events...

We believe our previous argumentation of the issue sufficient and that a comprehensive exploration is out of scope, given that we focus on two-year La Niña.

Other comments:

1. Still, the title should be narrowed/weakened as the manuscript is mainly about the strong El Niño to two-year La Niña, but not the entire multi-year La Niña or consecutive La Niña.

We believe that the current title is appropriate.

2. Lines 18-19, to my knowledge, the meridionally broader first La Niña is mainly found when strong El Niño precedes. Please either provide the reference for meridionally broader first La Niña when a strong El Niño does not precede or revise the sentence to specifically the strong El Niño to multi-year La Niña transition.

Our sentence is accurate and provides an appropriate context; the relation with strong El Niño is discussed elsewhere.

3. Lines 26-31, after showing the increase of multi-year La Niña, the authors focus only on El Niño, which confuses the reader, as people will expect an explanation of why multi-year La Niña increase. The authors should guide the reader to the increase of the El Niño-to multi-year La Niña.

We believe the current structure is appropriate. Text has been modified to the effect that the paragraph conveys the message that there is an increasing La Niña's sensitivity to El Niño.

4. Lines 43-44, citations 17, 19, 20, 21, and 22 are not really for multi-year La Niña impacts, even though the sentence is for comparing single to multi-year La Niña impacts.

This comment seems to be taken from the last round of review. It is out of context because Lines 43-44 do not contain citations 17 and 19. We believe that our citations here are appropriate.

5. Lines 72-73, the cooling left from the first La Niña increases the "probability" of creating

the second year La Niña, which needs to be included.

Revised. Thank you.

6. Lines 76-78, this

This seems to be an incomplete comment.

7. Lines 79-80, “from strong El Niño” or a similar description needs to be mentioned.

Revised.

8. Line 89, “do”?

Yes.

9. Line 100-101, this is a surprising result, as previous studies show most CMIP models' difficulty in simulating observed multi-year La Niña, such as Fang and Yu (2020). Is it because of a different classification method? The reason should be mentioned in the manuscript.

This is due to selection of models. We have added this information to the text (Line 102).

10. Lines 96-97, the authors still excluded the three-year (or more) multi-year La Niña from classification since only the first two years of the tripole event are considered. That is, the authors exclude the multi-year La Niña coming from La Niña, which is the reason why the composite of multi-year La Niña has a strong El Niño signal than the single-year La Niña event (Fig. 1b). And why there is a meridionally broader first-year La Niña (fig. 1d). Suggestion is provided in major comment 2.

We don't think considering a three-year La Niña event as two double-year La Niña events is appropriate.

11. Line 118, the multi-year La Niña considered in this study, can only come from El Niño or Neutral states since the authors do not consider the tripole La Niña as two two-year La Niña events. As a result, the “multi-year La Niña” and “all La Niña” are misleading. The authors should mention the changes in statistics when considering tripole La Niña as two events.

Again, we don't think considering a three-year La Niña event as two double-year La Niña events is appropriate. In the current framework, the distinction between “multi-year La Niña” and “all La Niña (which is single-year La Niña plus multi-year La Niña)” is clear.

12. Line 160, the piControl is not ‘unforced’.

Revised.

13. Extended Data Fig. 10, there is no d in the figure but in the caption.

Corrected.

14. Extended Data Fig. 10c, how does the figure create? Since it is the regression between temperature of D(1)JFMAMJJAS(2) and wind stress of MAMJJ(2). Does it mean the regression is only performed on multi-year La Niña events? If so, how many events are there for each experiment? It may not be representative if the events are limited.

Yes, the figure shows inter-model regression of changes (2000-2099 minus 1900-1999) in D(1)JFMAMJJAS(2) equatorial (5°S-5°N average) temperature tendency onto changes in mean intensity of MAMJJ(2) wind stress curl anomalies. Both temperature tendency and wind stress curl anomalies are each a composite of the first La Niña of multi-year La Niña events in each model before performing the inter-model regression. The number of events in each model is shown in Fig. 2a.

We have clarified this in figure caption.

15. Lines 248-250, “the persisting cold SST anomalies of the first La Niña are then amplified by the stronger Bjerknes positive feedback in late boreal summer-autumn, and develop into a second La Niña in the following winter”. Please provide corresponding evidence for this argument.

The stronger Bjerknes positive feedback has been thoroughly discussed in Cai et al. 2018 (Citation no. 37). Another representation of the intensified feedback is provided in new Supplementary Fig. 7.

Reviewer Reports on the Second Revision:

Referees' comments:

Referee #2 (Remarks to the Author):

April 2023,

A review of the resubmitted manuscript, "Increased occurrences of consecutive La Niña events under global warming" by Geng et al.

The authors have addressed my previous concerns satisfactorily except for the third point, which is about the concatenation of piControl and historical-ssp585 time series. I also found a minor typo. Meanwhile, a new supplementary Figure 12 and related text in Lines 548-552 are reasonably provided to indicate a caveat of the climatology definition method which is important but does not seriously alter the main results based on the multi-model ensemble. I also think the new Supplementary Fig. 7, which is about the seasonal ENSO feedback changes, in response to a comment from reviewer 3, is reasonable and good supporting information for Lines 250-253. I would like to ask the authors to reconsider the following two points before publishing this manuscript in the journal Nature.

1. Figure 2b and Lines 153-161: The time series in new Fig. 2b are separated between the piControl and historical-ssp585 to avoid concatenating these two that are actually disconnected in time in terms of internal variability (old Fig. 2b). However, at least to my eyes, it seems that the piControl time series is unchanged from the old figure, which is unexpected. Indeed, the end of the piControl time series is still very smoothly connected to the beginning of the historical time series. This probably means that, even in the new one, the first 59 years of the historical time series have been still concatenated to the end of the piControl 500-year time series when creating the left panel of Fig. 2b. If this is the case, the piControl time series still needs to be revised by removing the last 59 years of the 500 years in the present Fig. 2b or by using longer than 559-year piControl data to calculate a new 500-year time series.

2. Typo in Line 89. "Bjerknes" instead of "Bjkernes".

Author Rebuttals to Second Revision:

Response to Referee #2

The authors have addressed my previous concerns satisfactorily except for the third point, which is about the concatenation of piControl and historical-ssp585 time series. I also found a minor typo. Meanwhile, a new supplementary Figure 12 and related text in Lines 548-552 are reasonably provided to indicate a caveat of the climatology definition method which is important but does not seriously alter the main results based on the multi-model ensemble. I also think the new Supplementary Fig. 7, which is about the seasonal ENSO feedback changes, in response to a comment from reviewer 3, is reasonable and good supporting information for Lines 250-253. I would like to ask the authors to reconsider the following two points before publishing this manuscript in the journal Nature.

Thank you for your further comments.

1. Figure 2b and Lines 153-161: The time series in new Fig. 2b are separated between the piControl and historical-ssp585 to avoid concatenating these two that are actually disconnected in time in terms of internal variability (old Fig. 2b). However, at least to my eyes, it seems that the piControl time series is unchanged from the old figure, which is unexpected. Indeed, the end of the piControl time series is still very smoothly connected to the beginning of the historical time series. This probably means that, even in the new one, the first 59 years of the historical time series have been still concatenated to the end of the piControl 500-year time series when creating the left panel of Fig. 2b. If this is the case, the piControl time series still needs to be revised by removing the last 59 years of the 500 years in the present Fig. 2b or by using longer than 559-year piControl data to calculate a new 500-year time series.

We have removed the last 59 years of the 500 years (black time series) in Fig.2b. Corresponding figure caption has also been revised.

2. Typo in Line 89. "Bjerknes" instead of "Bjkernes".

Corrected.

Thank you.